# $G^2$Retro as a two-step graph generative models for retrosynthesis prediction

Ziqi Chen[1], Oluwatosin R. Ayinde[2], James R. Fuchs [2], Huan Sun[1,3] & Xia Ning [1,3,4 ✉]

Retrosynthesis is a procedure where a target molecule is transformed into potential reactants and thus the synthesis routes can be identified. Recently, computational approaches have been developed to accelerate the design of synthesis routes. In this paper,we develop a generative framework $G^2$Retro for one-step retrosynthesis prediction. $G^2$Retro imitates the reversed logic of synthetic reactions. It first predicts the reaction centers in the target molecules (products), identifies the synthons needed to assemble the products, and transforms these synthons into reactants. $G^2$Retro defines a comprehensive set of reaction center types, and learns from the molecular graphs of the products to predict potential reaction centers. To complete synthons into reactants, $G^2$Retro considers all the involved synthon structures and the product structures to identify the optimal completion paths, and accordingly attaches small substructures sequentially to the synthons. Here we show that $G^2$Retro is able to better predict the reactants for given products in the benchmark dataset than the state-of-the-art methods.

[1] Computer Science and Engineering, The Ohio State University, Columbus, OH 43210, USA. [2] Medicinal Chemistry and Pharmacognosy, College of Pharmacy, The Ohio State University, Columbus, OH 43210, USA. [3] Translational Data Analytics Institute, The Ohio State University, Columbus, OH 43210, USA. [4] Biomedical Informatics, The Ohio State University, Columbus, OH 43210, USA. ✉email: ning.104@osu.edu

Retrosynthesis is a procedure where a target molecule is transformed into potential reactants and thus the synthesis routes can be identified. One-step retrosynthesis, which transforms a molecule into the possible direct reactants that can be used to synthesize the molecule, serves as the foundation of multi-step synthesis planning[1,2] that identifies a full synthesis route in which the target molecule can be made through a series of one-step synthesis reactions. In drug discovery, identifying feasible synthesis routes for drug-like molecules remains a factor that substantially challenges medicinal chemists in making the desired molecules experimentally[3]. An extensive, diverse library of high-quality synthesis routes for a given molecule has the potential to enable more feasible reaction solutions starting from commercially available, chemical building blocks, and to provide more options for operationally simple, high-yielding transformations using widely accessible reactants.

Current retrosynthesis planning is primarily conducted by synthetic and medicinal chemists based on their knowledge and experience. It has been long known that there exists substantial disagreement among chemists in assessing synthesisbilty and designing synthesis routes[4–7]. In addition, an ever-increasing number of new chemical reactions makes it highly challenging for a chemist to keep up to date. Therefore, a data-driven model that predicts synthetic reactions could provide a useful complement to chemist evaluations, and could provide a large pool of potential reactions that the chemists can consider. There exist proprietary synthesis reaction databases manually curated from the literature, including Reaxys[8] and SciFinder[9]. Unfortunately, the high prices of these databases act to limit their accessibility in some academic and small biotech settings. Open-sourced synthesis reaction databases such as the Open Reaction Database[10] are limited in the reactions they cover (e.g., majorities are United States Patent and Trademark Office (USPTO) public reactions[11]) and their search functionalities (e.g., via SMILES strings). Even with the aid of these databases, the development of new reactions and synthetic pathways for the preparation of challenging molecules remains non-trivial. In addition, database searches can be time-consuming with low throughput, particularly when without extensive domain knowledge to guide the process. Recent in silico retrosynthesis prediction methods using deep learning[12–32] have enabled alternative computationally generative processes to accelerate the conventional paradigm. These deep-learning methods learn from string-based representations (SMILES) or graph representations of given molecules, and generate possible reactant structures that can be used to synthesize these molecules, leveraging the advancement of natural language processing[33], graph neural networks[34], variational auto-encoders[35] and other techniques in deep learning. They have demonstrated strong potential to substantially accelerate and advance retrosynthesis analysis[36]. In this manuscript, we focus on the one-step retrosynthesis prediction, which predicts the possible direct reactants for the synthesis of the target molecules, and acts as the foundation of multi-step retrosynthesis analysis[1].

We develop a semi-template-based method via deep learning for one-step retrosynthesis prediction, denoted as G$^2$Retro. G$^2$Retro imitates the reversed logic of synthetic reactions: it first predicts the reaction centers in the target molecules, identifies the synthons needed to assemble the final products, and transforms these synthons into reactants. Therefore, G$^2$Retro follows the semi-template-based frame, as in the previous methods[27–30]. To predict reaction centers, G$^2$Retro learns from the molecular graphs of the products via a customized graph representation learning[37] and embedding approach (in "Molecule Representation Learning" Section), and uses the graph structures to predict potential reaction centers. G$^2$Retro defines a comprehensive set of reaction center types, and for each reaction center type, uses the graph structures that are most relevant to that reaction center type (in "Reaction Center Identification" Section). G$^2$Retro-B integrates information of synthetically accessible fragments in its molecule graph representation learning (in Supplementary Note 1).

The predicted reaction centers by G$^2$Retro split the products into synthons. To complete synthons into reactants, G$^2$Retro considers all the involved synthon structures and the product structures to identify the optimal completion paths (in "Attachment Continuity Prediction (AACP)" Section), and accordingly attaches small substructures (i.e., bonds or rings) sequentially to the synthons until the extended synthon structures are predicted as possible reactants (in "Attachment Type Prediction (AATP)" Section). All the involved predictions in G$^2$Retro and G$^2$Retro-B are done via tailored neural networks. Note that G$^2$Retro and G$^2$Retro-B allow multiple reaction centers and multiple completion paths for each product to increase diversity in its predicted reactions. That is, the top predicted reaction centers (according to predicted likelihoods) are all tested in synthon completion to produce different reactions. Meanwhile, to avoid the exhaustive generation of all possible reactions from the top reaction centers, G$^2$Retro prioritizes the most possible completion paths via a new beam search strategy (in "Inference" Section). An ensemble of G$^2$Retro was also developed, denoted as G$^2$Retro-ens, an ensemble of G$^2$Retro, increases the pool of generated reactions by combining multiple G$^2$Retro models and their predictions. Figure 1 presents an overview of G$^2$Retro. A comprehensive review of existing retrosynthesis prediction methods and related fragment-based molecular generation methods is available in "Related Work" Section.

As a summary, G$^2$Retro has the following advantages:

- G$^2$Retro follows a semi-template-based framework, predicts reaction centers of different types in products first, and then transforms the resulting synthons into reactants by adding substructures to the synthons. This process imitates the reversed logic of synthetic reactions and enables necessary interpretability as to which reaction centers are predicted by G$^2$Retro, which reactants are generated from the reaction centers and the corresponding step-by-step generation process.

- G$^2$Retro defines a comprehensive set of reaction center types, covering 97.5% of the test data and conforming to synthetic chemistry knowledge. New customized neural networks are developed to predict each type of the reaction centers as well as their associated atom changes. Multiple reaction center candidates are considered for each product to enable diverse reactions generated from different reaction centers in the predicted reactions.

- G$^2$Retro develops a new fragment-based generation strategy compared to the previous semi-template-based methods[27–30], to complete synthons into reactants by sequentially attaching substructures (i.e., bonds and rings) starting from the predicted reaction centers (in "Synthon Completion" Section). The prediction of these substructure attachments utilizes a holistic view of the most updated structures of the synthon to be completed, and the structures of the final product and other synthons.

- G$^2$Retro employs a new, effective beam search strategy compared to the previous semi-template-based methods[27–30], that prioritizes the most possible reactants and the corresponding completion actions along the synthon completion paths. The beam search also allows multiple different reaction centers, enabling diversity in the completed reactants.

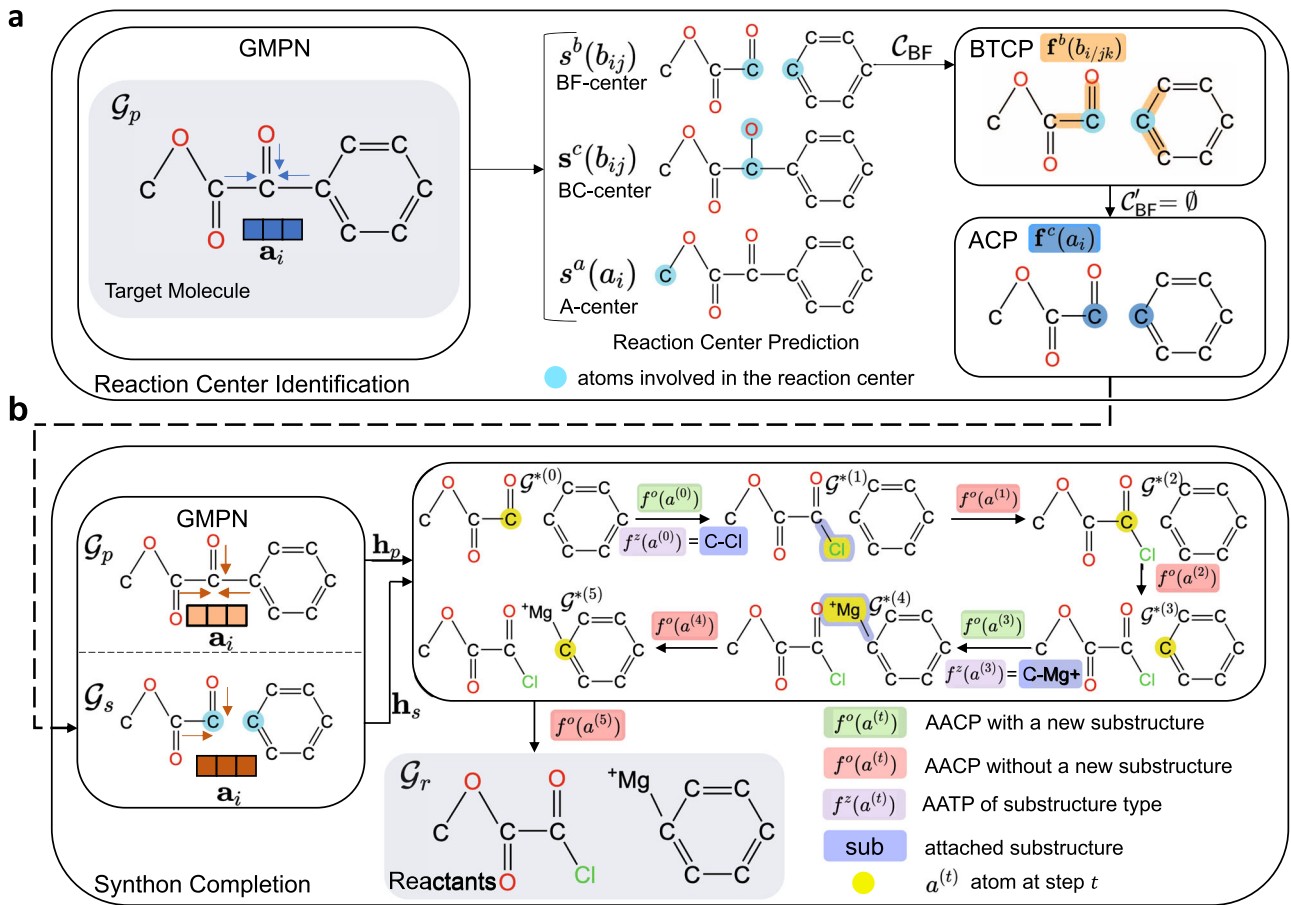

**Fig. 1 G²Retro retrosynthesis prediction process. a** G²Retro reaction center identification. G²Retro uses a graph message passing network (GMPN); G²Retro predicts three types of reaction centers: newly formed bonds (**BF-center**), bonds with type changes (**BC-center**), and atoms with leaving fragments (**A-center**); for BF-center, G²Retro also predicts bonds that have type changes induced by the newly formed bonds (**BTCP**); for all the reaction center types, G²Retro predicts atoms with charge changes (**ACP**). **b** G²Retro synthon completion. G²Retro uses **GMPN** to represent both the products and the synthons; G²Retro sequentially predicts whether a new substructure should be attached (**AACP**) and the type of the attachment (**AATP**); G²Retro adds predicted substructures until **AACP** predicts 'stop'.

- G²Retro and G²Retro-B are compared with nineteen baseline methods and demonstrate the state-of-the-art performance over the benchmark data (in "Overall Comparison" Section). Case studies show that G²Retro could propose diverse and reasonable synthesis routes with high predicted likelihoods that are not included in the benchmark data (in "Case Study" Section).
- G²Retro-ens is an ensemble of G²Retro models and demonstrates strong performance on the benchmark data compared to two baseline methods with data augmentation (in "Performance of Ensemble-based Methods" Section).

## Results

**Overall comparison**. Table 1 presents the overall comparison between G²Retro, G²Retro-B and the baseline methods on one-step retrosynthesis under two conditions, following the standard protocol in literature[12–14,19,24,26–30]: (1) when the reaction type is given a priori for both model training and inference (i.e., "Reaction type known"); and (2) when the reaction type is always unknown (i.e., "Reaction type unknown"). When the reaction type is known, G²Retro uses a one-hot encoder as an additional feature for each atom in product molecules indicating the reaction type. Particularly, for Semi-TB methods, the performance in Table 1 corresponds to the predictions out of the two steps, that is, the synthon completion is done according to a reaction center

that is *predicted* from the reaction center prediction step. Following the prior work[12,14,28,29], we used the top-$k$ ($k = 1,3,5,10$) accuracy to evaluate the overall performance of all the methods. Top-$k$ accuracy is the ratio of test products that have their ground truth correctly predicted among their top-$k$ predictions. Higher top-$k$ accuracy indicates better performance. Note that ground truth reactions are those included in the benchmark data. While there is always one ground-truth reaction for each product in the benchmark data, there may exist actually numerous feasible reactions for each product that are not included in the benchmark data. Therefore, reactants that are considered incorrect based on the benchmark data might still be plausible and included in other larger databases. Also, note that the top-$k$ accuracies of all the baseline methods are the reported results in their original papers (issues related to the comparison among methods are discussed later). Details of baseline methods are available in "Baselines" Section.

*Comparison with semi-template-based (Semi-TB) methods*. When the reaction type is known, compared to other Semi-TB methods, G²Retro achieves the best performance on top-3 (84.2%), top-5 (88.5%) and top-10 (91.7%) accuracies, corresponding to 3.2%, 2.9%, and 3.4% improvement over those from the best baselines (81.6% for RetroPrime[30] on top-3, 86.0% and 88.7% for G2G[28] on top-5 and top-10) on these three metrics. In terms of top-1 accuracy, G²Retro-B achieves the third-best performance

**Table 1 Overall comparison on retrosynthesis prediction in top-*k* accuracy (%).**

| Method type | Method | Coverage(%) | Reaction type known | | | | Reaction type unknown | | | |
|---|---|---|---|---|---|---|---|---|---|---|
| | | | 1 | 3 | 5 | 10 | 1 | 3 | 5 | 10 |
| TB | Retrosim[12] | 100.0 | 52.9 | 73.8 | 81.2 | 88.1 | 37.3 | 54.7 | 63.3 | 74.1 |
| | Neuralsym[13] | 100.0 | 55.3 | 76.0 | 81.4 | 85.1 | 44.4 | 65.3 | 72.4 | 78.9 |
| | GLN[14] | 93.3 | **64.2** | 79.1 | 85.2 | 90.0 | 52.5 | 69.0 | 75.6 | 83.7 |
| | MHNreact[15] | 100.0 | - | - | - | - | 50.5 | 73.9 | 81.0 | 87.9 |
| | LocalRetro[16] | 98.1 | 63.9 | **86.8** | **92.4** | **96.3** | 53.4 | **77.5** | **85.9** | **92.4** |
| TF | SCROP[17] | 100.0 | 59.0 | 74.8 | 78.1 | 81.1 | 43.7 | 60.0 | 65.2 | 68.7 |
| | LV-Trans[18] | 100.0 | - | - | - | - | 40.5 | 65.1 | 72.8 | 79.4 |
| | GET[19] | 100.0 | 57.4 | 71.3 | 74.8 | 77.4 | 44.9 | 58.8 | 62.4 | 65.9 |
| | Chemformer[20] | 100.0 | - | - | - | - | **54.3** | - | 62.3 | 63.0 |
| | Graph2SMILES[21] | 100.0 | - | - | - | - | 51.2 | 66.3 | 70.4 | 73.9 |
| | TiedTransformer[22] | 100.0 | - | - | - | - | 47.1 | 67.1 | 73.1 | 76.3 |
| | GTA[23] | 100.0 | - | - | - | - | 51.1 | 67.6 | 74.8 | 81.6 |
| | Dual[24] | 100.0 | **65.7** | 81.9 | 84.7 | 85.9 | 53.6 | 70.7 | 74.6 | 77.0 |
| | Retroformer[25] | 100.0 | 64.0 | **82.5** | 86.7 | 90.2 | 53.2 | **71.1** | 76.6 | 82.1 |
| | MEGAN[26] | 100.0 | 60.7 | 82.0 | **87.5** | **91.6** | 48.1 | 70.7 | **78.4** | **86.1** |
| Semi-TB | RetroXpert[27] | 100.0 | 62.1 | 75.8 | 78.5 | 80.9 | 50.4 | 61.1 | 62.3 | 63.4 |
| | G2G[28] | 97.9 | 61.0 | 81.3 | 86.0 | 88.7 | 48.9 | 67.6 | 72.5 | 75.5 |
| | GraphRetro[29] | 95.0 | 63.9 | 81.5 | 85.2 | 88.1 | 53.7 | 68.3 | 72.2 | 75.5 |
| | RetroPrime[30] | 100.0 | **64.8** | 81.6 | 85.0 | 86.9 | 51.4 | 70.8 | 74.0 | 76.1 |
| | G$^2$Retro | 97.5 | 63.1 | **84.2** | **88.5** | **91.7** | <u>53.9</u> | **74.6** | <u>80.7</u> | <u>86.6</u> |
| | G$^2$Retro-B | 97.5 | 63.6 | <u>83.6</u> | <u>88.4</u> | 91.5 | **54.1** | <u>74.1</u> | **81.2** | **86.7** |

Columns with 1, 3, 5 and 10 present top-1, top-3, top-5 and top-10 accuracies, respectively. Column "Coverage(%)" represents the percentage of test reactions that the methods can be applied to. Best top-*k* accuracy values among the methods of each type are in **bold**. Top-*k* accuracy values of G$^2$Retro and G$^2$Retro-B are <u>underlined</u> if they are not the best but still better than all the baselines of the respective type. All the baseline results are reported in their original papers, where "-" represents that the corresponding results are not reported.

(63.6%) compared to those of RetroPrime (64.8%) and Graph-Retro (63.9%) on this metric. While G$^2$Retro underperforms RetroPrime on one metric, it is substantially better than RetroPrime on all the other metrics: G$^2$Retro outperforms RetroPrime on top-3 accuracy at 3.2%, on top-5 accuracy at 4.1%, and on top-10 accuracy at 5.6%.

When the reaction type is unknown, a similar trend is observed: G$^2$Retro-B outperforms all the Semi-TB baseline methods on all the top accuracy metrics, with 0.7% improvement over the best baseline GraphRetro on top-1 accuracy, and 4.7%, 9.7% and 13.9% improvement over those from the best baseline RetroPrime on top-3, top-5 and top-10 accuracies. G$^2$Retro has a performance similar to that of G$^2$Retro-B, with an even better top-3 performance 74.6% that is 5.4% improvement from that of RetroPrime.

Compared with the performance with known reaction types, all the methods including G$^2$Retro and G$^2$Retro-B have worse performance when the reaction types are unknown. It is well-known in synthetic chemistry that there are several well-characterized reaction types. These types have distinct patterns in their reactions and reaction centers. For example, acylation reactions are very common approaches to creating amide and sulfonamide linkages. They are known for their efficiency and high yields, especially when they involve acyl/sulfonyl halides[38]. The improved performance with known reaction types integrated into retrosynthesis model training demonstrates that leveraging a priori reaction type information could benefit retrosynthesis prediction in general. However, in real applications, reaction types are typically not available in retrosynthesis when only the target molecule is presented. The superior performance of G$^2$Retro and G$^2$Retro-B in "reaction type unknown" condition demonstrates their great utility in real applications.

As Table 1 shows, G$^2$Retro and G$^2$Retro-B can cover (i.e., can be applied to) 97.5% of the test reactions, which determines the upper bound of accuracy values, due to the definition of reaction centers (the rest 2.5% correspond to reactions with multiple newly formed or changed bonds). Among other Semi-TB methods, G2G and GraphRetro[29] also have limited coverage on test set (97.9% for G2G and 95.0% for GraphRetro). RetroXpert[27] has 100% coverage because its reactant SMILES generation from synthons recovers all possible reaction centers. RetroPrime[30] also has 100% coverage due to its very comprehensive set of reaction centers. Although G$^2$Retro and G$^2$Retro-B cannot cover all possible cases in the test set, they still outperform other Semi-TB methods, measured over the entire test set. More discussion on the coverage of the two steps in Semi-TB methods is available in the Section "Individual Module Performance".

GraphRetro and RetroPrime are two strong baselines. GraphRetro has good top-1 accuracies but much worse results on other top accuracy metrics. According to its authors[29], GraphRetro tends to bias its beam search to the most possible reaction center. Thus, it may prioritize the most possible reactants from the most possible reaction center at the very top of its predictions. However, if the most possible reaction centers are not the ground truth, GraphRetro would totally miss the ground truth in its beam search, resulting in poor performance on other top accuracy metrics. In addition, such focused beam search limits the diversity of identified synthons, and thus the completed reactants. RetroPrime achieves the best top-1 accuracy with reaction type known. It uses augmented SMILES strings (i.e., each product has multiple, equivalent, non-canonical SMILES strings) in training the two sequence-to-sequence transformers. It is likely that top results in RetroPrime correspond to the ground truth but in different, augmented SMILES strings, and thus high top-1 accuracy but low and similar other top accuracies. These three Semi-TB baseline methods only perform well on one certain metric (in one certain condition), but do not show consistent optimality across many metrics or across the two conditions.

Compared to these baselines, G$^2$Retro always achieves the best performance on all the top accuracy metrics (except on top-1 accuracy when reaction types are known). High top-*k* accuracies

at all different $k$ are desired as they indicate the holistically high ranking positions of the ground truth in the predicted reactions, and thus the capability of models in recovering knowledge from data. High top-$k$ accuracies with $k > 1$ may signify plausible reactions not included in the dataset, as will be examined later in Section "Case Study". This is because high top-$k$ ($k > 1$) accuracy implies that there might be a few reactions different from the ground truth but are very possible and thus are ranked on top. Such results may enable the exploration of multiple synthesis routes and may be of synthetic value if specific coupling methods fail or if specific starting materials are unavailable. From the above two aspects, over all the metrics, G²Retro and G²Retro-B achieve the overall best performance compared to the three strong Semi-TB methods.

G²Retro-B performs slightly better than G²Retro when the reaction types are unknown, but worse than G²Retro when the reaction types are known. G²Retro-B integrates synthetically accessible fragments in atom embeddings (Eq. S3 in Supplementary Note 1). When the reaction types are unknown, the fragment information provides additional local contexts to atoms, which could facilitate better decisions on reaction center prediction and synthon completion. When the reaction types are known, atom embeddings directly integrate the reaction type information in G²Retro, which may outweigh the contextual information provided by the fragments, and thus G²Retro-B does not achieve additional performance improvement from G²Retro.

*Comparison with template-free (TF) methods.* G²Retro and G²Retro-B also demonstrate superior or competitive performance compared to TF methods on all the top accuracies. With reaction types known, G²Retro is the best on top-3, top-5 top-10 accuracies compared to all the template-free methods; with reaction types unknown, G²Retro-B is the best on top-3, top-5 and top-10 accuracies, and is the second best one on top-1 accuracy. For example, G²Retro is 4.9% better than the best TF method on top-3 accuracy (i.e., Retroformer[25]) with the reaction types unknown. Most TF methods such as Dual[24] and Chemformer[20] have the competitive performance on top-1 accuracy but relatively worse results on other top accuracy metrics. This could be due to that these TF methods with SMILES representations may fail to generate diverse or even many valid reactants with beam search[39], leading to limited variation in their predicted results, and thus low and similar top-3, top-5 and top-10 accuracies. This lack of diversity and richness in the predictions, in addition to the lack of interpretability during the chemical sequence transformation process, could hinder the application of TF methods in retrosynthesis prediction. However, the prediction diversity and richness in G²Retro is enabled by the multiple possible reaction centers predicted by G²Retro and the corresponding completed reactants.

In terms of the coverage on the test set, all the SMILES-based TF methods can cover the entire test set, because all the reactions can be represented as SMILES string transformation. The graph-based TF method MEGAN[26] also covers the entire test set due to its comprehensive set of graph edit actions. Compared to these TF methods, though without the full coverage on the test set, G²Retro and G²Retro-B model reactions through a two-step process of reaction center identification and synthon completion, allowing for the interpretability of reaction centers in the predicted reactants. Overall, G²Retro and G²Retro-B achieve even better performance than the methods with full coverage, measured on the entire test set.

*Comparison with template-based (TB) methods.* G²Retro and G²Retro-B achieve competitive performance with that from the TB methods. With reaction types known, G²Retro achieves either the second or the third on all the top accuracies; with reaction types unknown, G²Retro-B achieves the best performance on top-1 (54.1%), and either the second or the third on all the other top accuracies. For example, with reaction types unknown, G²Retro-B is the second best on top-3 accuracy, with 3.8% difference from the best performance of LocalRetro[16]; G²Retro-B slightly underperforms the second-best baseline MHNreact[15] on top-10 (86.7% compared to 87.9% from MHNreact), but outperforms MHNreact on all the other metrics. LocalRetro is a very strong TB method. It extracted 731 templates from the benchmark training data, whereas other TB methods have much more templates (11,647 for GLN and 9162 for MHNreact). Therefore, LocalRetro could achieve better template selection over a small template set compared to others over much larger template sets. However, LocalRetro may suffer from scalability issues on large datasets because it scores all the reaction templates on all the potential reaction centers (i.e., all atoms and all bonds) in the product molecules. In general, all TB methods may not generalize well to reactions that are not covered by the templates[29]. In terms of coverage on the test set, Table 1 shows that the templates used in Retrosim, Neuralsym and MHNreact can cover the entire test set, while the templates used in GLN[14] and LocalRetro cannot (93.3% for GLN and 98.1% for LocalRetro). Unlike TB methods, G²Retro does not use reaction templates, and only scores all the bonds and atoms once for reaction center identification, and thus is much more scalable in inference. It learns the patterns from training data and thus has a better chance to discover new patterns from the training data that are not covered by templates.

**Individual module performance.** Following the typical evaluation for Semi-TB methods as in literature[29], Table 2 presents the individual performance of the two modules—reaction center identification and synthon completion in Semi-TB methods. In Table 2, for the reaction center identification module, the top-$k$ accuracy measures the ratio of test products that have the ground-truth reaction center correctly predicted among the top-$k$ predictions. In the synthon completion module, the synthon completion is done according to the *ground-truth* reaction center, not the *predicted* reaction center; the top-$k$ accuracy measures the ratio of test products that have the ground-truth reactants correctly predicted among the top-$k$ predictions. Please note that here "ground-truth" reaction center means the reaction center as appears in the benchmark data per our reaction center definition.

*Comparison on reaction center identification.* Among all the Semi-TB methods, the definitions of reaction centers vary. In G2G, reaction centers are referred to as the only one newly formed bond during the reaction, and reaction center identification predicts whether there is such a new bond (and its location) or not in the products as in a classification problem. This reaction center definition and classification can cover 97.9% of the test data (the rest 2.1% correspond to multiple newly formed bonds). GraphRetro defines the reaction center as the newly formed bond (BF-center as defined in Section "Reaction Centers with New Bond Formation" but without induced bond changes), the changed bond (BC-center as in "Reaction Centers with Bond Type Change") and the single atom with changed hydrogen count (A-center as in "Reaction Centers with Single Atoms"), which in total covers 95.0% of the reactions in the test set. RetroPrime aims to identify all the atoms involved in the reactions as reaction centers, which covers all the reactions in the test set. G²Retro extends the definition of the reaction center in GraphRetro with induced bond type change and atom charge changes, covering 97.5% of the test set.

**Table 2 Module performance comparison on reaction center identification and synthon completion in top-k accuracy (%).**

| Module | Method | Coverage (%) | Reaction type known | | | | Reaction type unknown | | | |
|---|---|---|---|---|---|---|---|---|---|---|
| | | | 1 | 2 | 3 | 5 | 1 | 2 | 3 | 5 |
| Reaction center identification | G2G | 97.9 | 90.2 (92.1) | 94.5 (96.5) | 94.9 (96.9) | 95.0 (97.0) | 75.8 (77.4) | 83.9 (85.7) | 85.3 (87.1) | 85.6 (87.4) |
| | GraphRetro | 95.0 | 84.6 (89.1) | 92.2 (97.1) | 93.7 (98.6) | 94.5 (99.5) | 70.8 (74.5) | 85.1 (89.6) | 89.5 (94.2) | 92.7 (97.6) |
| | RetroPrime | 100.0 | 84.6 (84.6) | 94.0 (94.0) | 96.7 (96.7) | 97.9 (97.9) | 65.6 (65.6) | 81.3 (81.3) | 87.7 (87.7) | 92.0 (92.0) |
| | G²Retro | 97.5 | 84.3 (86.5) | 94.6 (97.0) | 96.5 (99.0) | 97.0 (99.5) | 69.5 (71.3) | 85.6 (87.8) | 90.8 (93.1) | 94.8 (97.2) |
| | G²Retro-B | 97.5 | 85.0 (87.2) | 94.1 (96.5) | 96.2 (98.7) | 97.3 (99.8) | 69.3 (71.1) | 85.4 (87.6) | 91.1 (93.4) | 94.7 (97.1) |
| Synthon completion | G2G | 100.0 | 66.8 | - | 87.2 | 91.5 | 61.1 | - | 81.5 | 86.7 |
| | GraphRetro | 99.7 | 77.4 (77.6) | 89.5 (89.8) | 94.2 (94.5) | 97.6 (97.9) | 75.6 (75.8) | 87.4 (87.7) | 92.5 (92.8) | 96.1 (96.4) |
| | RetroPrime | 100.0 | 75.0 | - | 88.9 | 90.6 | 73.4 | - | 87.9 | 89.8 |
| | G²Retro | 100.0 | 72.8 | 85.6 | 90.2 | 93.0 | 73.3 | 84.6 | 89.6 | 92.8 |

Columns with 1, 3, 5 and 10 present top-1, top-3, top-5 and top-10 accuracies, respectively. Column "Coverage(%)" represents the percentage of test reactions that the modules of methods can be applied to. "( . )": the accuracy within the covered reactions. All the baseline results are reported in their original papers, where "-" represents that the corresponding results are not reported.

Due to the data leakage issue as revealed by Yan et al.[27] (i.e., reaction center is given in both the training and test data), the reported G2G reaction center identification performance as cited in Table 2 is overestimated, but the updated results have not been provided in their Github. GraphRetro uses two functions, one for bonds and one for atoms, to predict reaction centers. While these functions are able to predict well when such bonds and atoms are truly reaction centers (i.e., performance in parentheses in Table 2), GraphRetro's reaction center definition covers the least (95%) of the test set compared to the other methods, resulting in still low accuracies (i.e., performance outside parentheses) over the test set. RetroPrime has a very generic definition of reaction centers—any atoms involved in the reactions, and uses one unified model to predict these atoms. However, as these atoms may experience different changes (e.g., connected to or disconnected from other atoms), a unified model not customized to specific changes may not suffice, leading to overall relatively low accuracies compared to other methods, particularly when reaction types are unknown. G²Retro and G²Retro-B have the most comprehensive definition of reaction centers (Section "Reaction Center Identification") with high coverage (97.5%) on the test set. In addition, G²Retro and G²Retro-B use a specific predictor for each of the reaction center types. Therefore, they achieve the best overall accuracy among the entire test set, as well as good performance over the reactions covered by its reaction center definition.

*Comparison on synthon completion.* To compare synthon completion performance, all the ground-truth reaction centers defined by different methods are given and used to start the completion processes. G2G predicts only bond establishment in its reaction center identification and thus has to deal with any associated changes such as bond type change in its synthon completion process, which complicates the synthon completion prediction. Therefore, its performance on synthon completion is the worst among all the methods.

GraphRetro formulates the synthon completion as a classification problem over all the subgraphs that can realize the difference between the synthons and reactants. Therefore, its synthon completion is not guaranteed to work for all possible products (e.g., 99.7% coverage over the test set), particularly if the needed subgraph is not included in the pre-defined vocabulary. Among all the products that GraphRetro can handle, its synthon completion performance is the best, due to that classification can be much easier than generation as all the other methods do. However, since GraphRetro does not do well in reaction center identification, overall, it does not outperform other methods in retrosynthesis prediction as Table 1 demonstrates. In addition, the synthon completion module of GraphRetro may fail to accurately estimate the likelihoods of leaving groups, due to the ignorance of overall structures of predicted reactants. Such inaccurate likelihood estimation may aggravate the bias of beam search and reduce the diversity of predicted reactants as discussed in GraphRetro[29].

RetroPrime transforms the synthons to reactants using a Transformer, but similarly to G2G, also needs to deal with additional predictions such as bond type change. RetroPrime's synthon completion performs reasonably well on top-1 accuracies. Together with its good top-1 accuracy on reaction center identification, RetroPrime achieves the best top-1 accuracy with reaction type known as demonstrated in Table 1. RetroPrime uses a rule to enumerate predicted reactants from the top-3 reaction centers, limiting the potential diversity of predicted reactants. On average, RetroPrime underperforms G²Retro, particularly on top-3 and top-5 accuracies in synthon completion.

**Table 3 G²Retro performance on different reaction types.**

| Type name | Percentage (%) | Reaction type known | | | | Reaction type unknown | | | |
|---|---|---|---|---|---|---|---|---|---|
| | | 1 | 3 | 5 | 10 | 1 | 3 | 5 | 10 |
| Heteroatom alkylation and arylation | 30.3 | 62.3 | 84.1 | 90.2 | 94.4 | 56.1 | 77.2 | 84.4 | 91.3 |
| Acylation and related processes | 23.8 | 76.1 | 93.9 | 96.7 | 97.6 | 67.0 | 87.3 | 92.3 | 95.4 |
| Deprotections | 16.5 | 58.3 | 87.2 | 91.5 | 93.9 | 51.8 | 76.5 | 82.7 | 87.9 |
| C-C bond formation | 11.3 | 48.1 | 68.1 | 75.7 | 82.4 | 37.2 | 56.6 | 67.9 | 75.7 |
| Reductions | 9.2 | 72.5 | 87.9 | 91.8 | 95.0 | 52.7 | 69.8 | 78.1 | 84.6 |
| Functional group interconversion | 3.7 | 50.5 | 69.0 | 75.5 | 81.0 | 42.4 | 52.7 | 60.9 | 67.9 |
| Heterocycle formation | 1.8 | - | - | - | - | - | - | - | - |
| Oxidations | 1.6 | 86.6 | 91.5 | 92.7 | 95.1 | 62.2 | 80.5 | 85.4 | 91.5 |
| Protections | 1.4 | 85.3 | 89.7 | 89.7 | 89.7 | 48.5 | 67.6 | 85.3 | 86.8 |
| Functional group addition | 0.5 | 95.7 | 95.7 | 95.7 | 95.7 | 78.3 | 82.6 | 87.0 | 87.0 |

Columns with 1, 3, 5 and 10 present top-1, top-3, top-5 and top-10 accuracies, respectively. Column "Percentage(%)" represents the percentage of reactions in the test set belonging to the specific reaction type. "-" represents that the corresponding results are not available due to the lack of coverage.

G²Retro does not use BRICS fragments in synthon completion because the fragment information is not available for the substructures that will be attached to synthons. Compared to GraphRetro, G²Retro leverages a generative process to add substructures to synthons in synthon completion, which is inherently more difficult than classification as in GraphRetro but could be generalizable to new products and reactants. Meanwhile, G²Retro does not limit the number of reaction centers within the top-10 predicted reactants, and thus increases the diversity of predicted reactants.

Although G²Retro does not outperform GraphRetro in the synthon completion module alone, its generative process allows G²Retro to consider all the intermediate molecular structures and more accurately estimate the likelihood of each completion action, conditioned on the reaction centers and the corresponding synthons from its reaction center identification module (i.e., not the ground-truth reaction centers). Consequently, despite employing a beam search strategy similar to that of GraphRetro, the generative process of G²Retro could alleviate the bias of beam search on most possible reaction centers by accurately estimating the likelihood of the completed reactants. In contrast, GraphRetro may not generalize well, particularly given that GraphRetro's reaction center identification does not perform well with respect to the ground-truth reaction centers (i.e., in the top panel of Table 2), but its synthon completion module is trained using the ground-truth reaction centers (i.e., in the bottom panel of Table 2).

**Performance on different reaction types**. Table 3 presents the top-$k$ accuracy ($k = 1,3,5,10$) of the reactions of different types. This method appears to predict certain reaction types more accurately than others as shown in Table 3. This is likely due to the relative structural diversity among potential reactants, particularly for substrates that can all provide the same products. For example, in the case of oxidations, only a very limited set of substrates can be utilized to generate a ketone, most commonly the oxidation of an alcohol, although ketones can certainly be accessed through other types of reactions as well. This leads to the relatively higher accuracies of G²Retro on the reactions of oxidations (e.g., 62.2% top-1 accuracy with reaction type unknown). In terms of reductions, however, numerous substrates could be utilized to generate an amine, including reductions of amides, nitro groups, and nitriles to name a few. In addition, there are numerous methods to access the same amines through various structurally unique deprotection reactions. The number of methods available to access a specific functional group, therefore,

may make it more difficult to accurately predict which method has been used for a specific molecule, leading to the lower accuracies on reactions of deprotections (e.g., 58.3% top-1 accuracy with reaction type known). This would certainly be the case in carbon-carbon bond forming reactions as well, which can be assembled in a number of ways from various substrates, potentially leading to a somewhat lower prediction success rate (e.g., 37.2% top-1 accuracy with reaction type unknown). In addition, as shown in our case studies, in molecules containing more than one functional group, there are often multiple ways in which that molecule can be assembled by targeting each individual functional group as the reaction center. This means that there are multiple valid reaction pathways which could be considered by synthetic chemists in order to most efficiently construct a molecule. Please note that G²Retro is designed to predict reactions that involve three types of reaction centers: (1) a single newly formed bond with induced changes in bond types; (2) a single changed bond; (3) a single atom with a fragment removed. As a result, G²Retro could not fully cover reaction types such as rearrangement, isomerization, cyclization and click reactions, which involve multiple changes in bond formation or atom detachment. This illustrates G²Retro's limitation in handling all possible reaction types. It is worth noting that other semi-template-based methods such as G2G and GraphRetro, also share this limitation. Therefore, developing an effective semi-template-based method that overcomes this limitation could be an interesting future research direction.

**Performance of ensemble-based methods**. We also compared the performance of an ensemble of G²Retro, referred to as G²Retro-ens, with AT[31] and R − SMILES[32], both of which test each target product multiple times and are strong baselines. AT and R − SMILES represent each target molecule using multiple non-canonical but equivalent SMILES strings, and use the multiple SMILES strings during model training and testing. By combining the predictions from the multiple SMILES strings of the same target product, these methods have the choice to explore a larger reaction subspace seeded by the SMILES strings, and thus achieve better prediction performance. Compared to the SMILES strings, G²Retro uses molecular graph representations, and thus each molecule can only have a unique representation. Instead of augmenting molecule representations but still being able to explore a larger reaction subspace as AT and R − SMILES do, G²Retro-ens tests each molecule multiple times using multiple G²Retro models. Details of G²Retro-ens are available in the supplementary Note 2.

**Table 4 Overall comparison on retrosynthesis prediction between G²Retro-ens and baselineswith test set augmentation in top-$k$ accuracy (%).**

| Dataset | Method type | Method | Reaction type unknown | | | |
|---|---|---|---|---|---|---|
| | | | 1 | 3 | 5 | 10 |
| All reactions | TF | AT[31] | 52.7 | 73.4 | 79.1 | 83.7 |
| | | R − SMILES[32] | **56.5** | **79.4** | **86.0** | **91.0** |
| | Semi-TB | G²Retro-ens | 56.4 | 78.8 | 85.2 | 90.5 |
| Reactions covered by G²Retro | TF | AT[31] | 54.1 | 75.5 | 81.4 | 85.8 |
| | | R − SMILES[32] | 56.8 | 79.7 | 86.2 | 91.3 |
| | Semi-TB | G²Retro-ens | **57.8** | **80.7** | **87.3** | **92.7** |

Columns with 1, 3, 5 and 10 present top-1, top-3, top-5 and top-10 accuracies, respectively. Best top-$k$ accuracy values among the methods of each type are in bold.

Table 4 presents the comparison among G²Retro-ens, AT and R − SMILES on top-$k$ accuracy ($k = 1,3,5,10$) over all the reactions and the reactions covered by G²Retro, both with the reaction type unknown. Please note that the performance of AT and R − SMILES on reactions with known types is not available in the respective papers[31,32], and the methods also cannot be easily extended to handle known reaction types. In Table 4, all the methods test each molecule 20 times, that is, AT and R − SMILES augment each target molecule with 20 SMILES strings, and G²Retro-ens uses an ensemble of 20 models to test each molecule. The results of AT and R − SMILES are calculated using the source code and data available from the respective papers. Table 4 shows that G²Retro-ens achieves competitive performance with the best baseline R − SMILES. Over all the reactions, G²Retro-ens achieves almost the best performance on top-1 (56.4%, compared to 56.5% for R − SMILES), and only slightly underperforms the best baseline R − SMILES on top-3, top-5 and top 10 (78.8% vs 79.2% on top-3; 85.2% vs 86.2% on top-5; 90.5% vs 91.0% on top-10). Over the reactions covered by G²Retro, G²Retro-ens outperforms the baseline R − SMILES on top-1 accuracy at 1.76%, on top-3 accuracy at 1.25%, on top-5 accuracy at 1.28%, and on top-10 accuracy at 1.53%. Compared to R − SMILES, which is an end-to-end black-box that directly transfers product SMILES string to reactant SMILES strings, G²Retro provides certain interpretability of the predicted reaction centers, and what reactants are generated from them. More details about the comparison on different reaction types and on reactions covered by G²Retro-ens are available in the supplementary Note 2.

**Case study**. G²Retro can predict multiple reactions for each product due to multiple predicted reaction centers. This variability could be useful for chemical synthesis in order to consider all possible reaction strategies. In order to illustrate the predictive power of G²Retro, we have highlighted the top-10 predicted reactants by G²Retro with reaction types unknown for four newly approved drug molecules in 2022, including Mitapivat, Tapinorf, Mavacamten, and Oteseconazole[40]. Among them, the predicted reactants for Mitapivat and Tapinorf are presented in Fig. 2aa and ba which will be discussed later; the results and discussions for Mavacamten and Oteseconazole are available in Supplementary Figure 1, Supplementary Figure 2 and Supplementary Note 3. Note that these drugs are not included in our training, validation, or testing data. Therefore, how G²Retro works on these drugs truly indicates its predictive power for new molecules.

Mitapivat as in Fig. 2aa is a drug approved for hereditary hemolytic anemias in 2022[41]. The synthetic route within the patent[42] reporting the discovery of Mitapivat utilizes an amide coupling reaction to form the C2-N23 bond (Fig. 2ab). This is

correctly predicted by G²Retro as the top-1 reaction (Fig. 2ac). As indicated by the top-5 reaction (Fig. 2ag), G²Retro also predicts that the amide coupling reaction could be performed with the carboxylate salt of one of the reactants, a useful reactant under the right pH conditions. G²Retro also predicts that the acyl chloride as the substrate in this transformation would also react with the amine group and produce the desired molecule (Fig. 2aj), In addition, G²Retro identifies the N7-S8 bond of sulfonamide linkage as the reaction center (e.g., Fig. 2ad, ae, af, ak, al). Most impressively, G²Retro predicts various S8 sulfonyl groups reacting with the N7 amine group, such as sulfonyl chloride (Fig. 2ad), sulfonyl fluoride (Fig. 2ae) and sulfonic acid (Fig. 2af), which are theoretically feasible for the formation of the N7-S8 bond. G²Retro also predicts that the N26-C27 bond could be the reaction center and formed by the N26 amine group reacting through a reductive amination with ketone in Fig. 2ah or through a nucleophilic substitution with the chloride in Fig. 2ai.

Tapinarof as in Fig. 2ba is a drug approved for plaque psoriasis and atopic dermatits[43]. The reported synthesis in patent[44] constructs this drug by removing the protecting groups on O5 and O10 (Fig. 2bb). G²Retro correctly predicts the deprotection of the methyl groups on O5 (Fig. 2bc) or O10 (Fig. 2bd), which would work to produce the desired molecule, although the ground truth failed to be predicted due to the limitation of reaction centers. Similarly, G²Retro generates possible reactants that contain different types of protected alcohols, as seen with the methoxymethyl groups on O5 and O10 in Fig. 2bf and bi and the benzyl-protected O5 in Fig. 2bj. Most impressively, G²Retro also identifies the alkene linkage between C11 and C12 (Fig. 2be and bl) and the C-C bond between C7 and C11 (Fig. 2bg, bh, ad bk) as reaction centers with various coupling reactions. These coupling reactions include McMurry coupling[45] (Fig. 2be), Wittig coupling[46] (Fig. 2bl) and Suzuki coupling[47] (Fig. 2bg and bh).

In addition, we also highlighted two molecules in the test set and their predicted reactions by G²Retro with reaction types unknown in Fig. 3a and b, respectively. The product in Fig. 3aa contains amide linkages and was assembled in the patent literature utilizing amide coupling reactions (ground truth in Fig. 3ab). G²Retro correctly predicted this coupling as the top-1 reaction for the construction of this molecule (Fig. 3ac). The other reactions predicted, however, are also very instructive into the strengths and limitations of G²Retro. In Fig. 3aa, the product has two amide groups in the side chain of the molecule. G²Retro identified both of these linkages as potential reaction centers (e.g., in Fig. 3ac between N5 and C6; in Fig. 3ag between N1 and C2). Typically, chemists would disconnect the molecule at the C6 amide carbonyl rather than C2 so that a fully elaborated side chain can be introduced to complete the molecule. This approach would generally be considered more efficient since its reaction introduces more complexity into the molecule in a single step and would therefore be predicted to limit the total number of steps

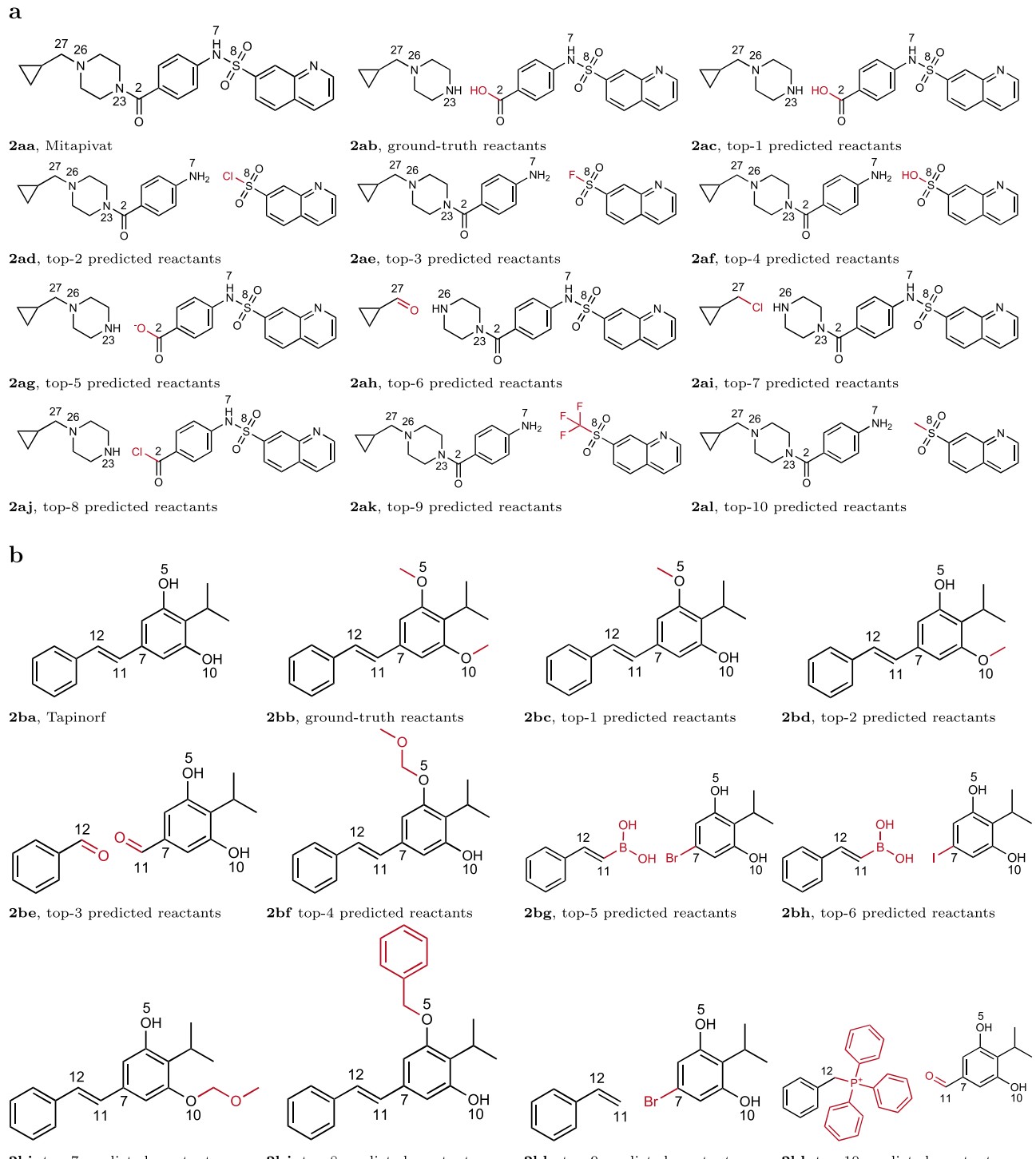

**Fig. 2 Predicted reactions by** G²Retro **for two newly approved drug molecules. a** Predicted reactions by G²Retro for Mitapivat; **b** Predicted reactions by G²Retro for Tapinorf. Numbers next to each atom are the indices of the atoms. Atoms with same indices in different subfigures are corresponding to each other. Atoms and bonds colored in red are leaving groups for synthon completion. Molecules with labels ending in (**a**) are product/target molecules; molecules with labels ending in (**b**) are the reactants reported in patents; molecules with labels ending in (**c-l**) are the top predicted reactants.

necessary to construct the molecule. In some limited cases, however, it may be necessary to introduce the nitrogen at N1 last (e.g., in Fig. 3af–ah), so this should also be considered a feasible reaction. In addition to the typical amide coupling strategy, which takes place between an amine and a carboxylic acid, G²Retro also correctly identifies the reaction of the amine with an acid chloride to make the same bond (Fig. 3ad). Although this was not the strategy utilized in the ground-truth study, this strategy would

certainly be expected to work in this case for construction of this molecule. The other common reaction that was predicted for this example was the nucleophilic addition of the N5 (or N1) amine into the C6 (or C2) carbonyl of an ester (N5-C6 - Fig. 3ae, ai, aj, ak, al and N1-C2 - Fig. 3ag and ah). This type of reaction, which is essentially a transamidation reaction, should also work to provide the product. Interestingly, however, G²Retro predicts several different esters as substrates for this transformation

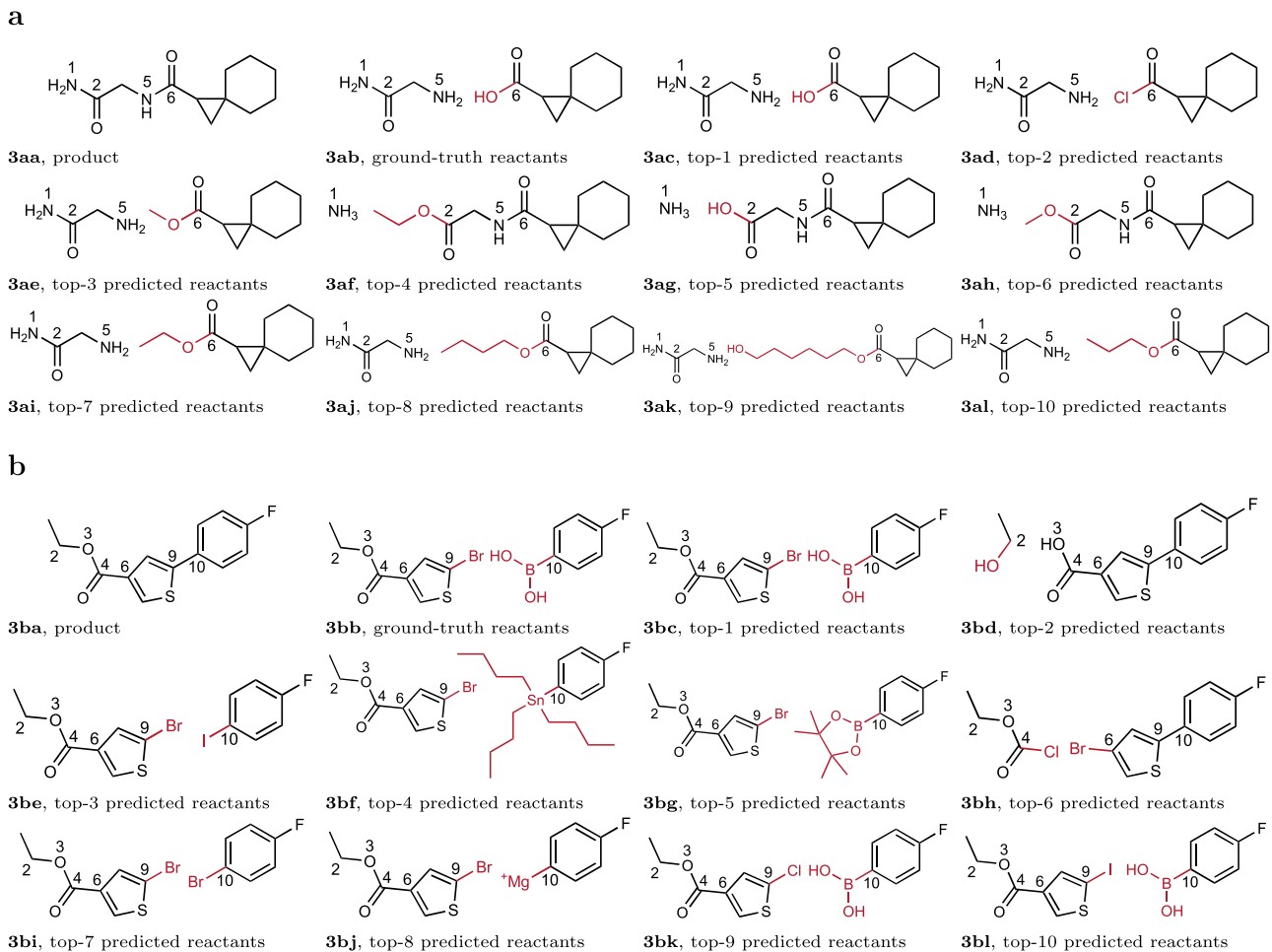

**Fig. 3 Predicted reactions by G²Retro for two test molecules in USPTO-50K. a** Predicted reactions by G²Retro for product "NC(=O)CNC(=O) C1CC12CCCCC2''; **b** Predicted reactions by G²Retro for product "CCOC(=O)c1csc(-c2ccc(F)cc2)c1''. Numbers next to each atom are the indices of the atoms. Atoms with same indices in different subfigures are corresponding to each other. Atoms and bonds colored in red are leaving groups for synthon completion. Molecules with labels ending in (**a**) are product/target molecules; molecules with labels ending in (**b**) are the ground-truth reactants in USPTO-50K; molecules with labels ending in (**c–l**) are the top predicted reactants.

(Fig. 3ac, ae, ai, aj, ak and al). While these are different substrates, the variation of the ester side chain in these cases would not typically be considered as greatly different by a synthetic chemist unless steric or electronic contributions affect the reactivity/ electrophilicity of the ester carbonyl.

Retrosynthesis of the product in Fig. 3b involves a C-C bond forming reaction between C9 and C10 (Fig. 3ba). The disconnection of the carbon-carbon bond between the two aromatic rings, a heteroaromatic thiophene and a benzene ring in this case, represents the most obvious disconnection in the molecule. In this case, the top-1 reaction (Fig. 3bc) predicted by G²Retro for this transformation is a Suzuki coupling[47], a common metal-mediated coupling between a boronic acid reactant and a corresponding aryl halide. This common transformation is the same reaction observed in the ground truth (Fig. 3bb). Interestingly, G²Retro also identifies additional permutations of this Suzuki reaction through changing the nature of the aryl halide (Fig. 3bk and bl). Traditionally, aryl chlorides (Fig. 3bk) are less reactive than aryl bromides or iodides (Fig. 3bc and bl) for coupling reactions and in the past were considered unreactive in these reactions. Newer methods[48] using specially designed ligands, however, have made the use of such chlorides possible. The other difference observed in the predicted Suzuki couplings is the use of a boronic ester (Fig. 3bg) vs a boronic acid (Fig. 3bc). Both boronic acids and boronic esters are common

reagents for these transformations, with many being readily available from commercial sources. G²Retro also predicts that an esterification reaction at the C4 carboxylic acid would also work to produce the desired molecule (Fig. 3bd). While this is potentially not as synthetically useful for building the molecule, it is a reasonable transformation. Most impressively, G²Retro also predicts other coupling reactions[49] for the biaryl coupling reaction. These other methods include an Ullmann-type coupling[50] (Fig. 3be and bi) a Stille coupling[51] (Fig. 3bf), and a Kumada coupling[48,52] (Fig. 3bj). This versatility predicted in the top-10 reactions may be of synthetic value for substrates if specific coupling methods fail or if the functionality necessary for one type of coupling reaction is not able to be easily prepared.

The above examples indicate that the predicted reactions from G²Retro rather than the ground truth could be still possible and synthetically useful. Therefore, a more comprehensive evaluation strategy is needed not to miss those possible and potentially novel synthesis reactions.

**Diversity on predicted reactions.** Diversity in predicted reactions is always desired, as it has the potential to enable the exploration of multiple synthesis routes. G²Retro has the mechanisms to facilitate diverse predictions: The beam search strategy in G²Retro allows multiple reaction centers and multiple different

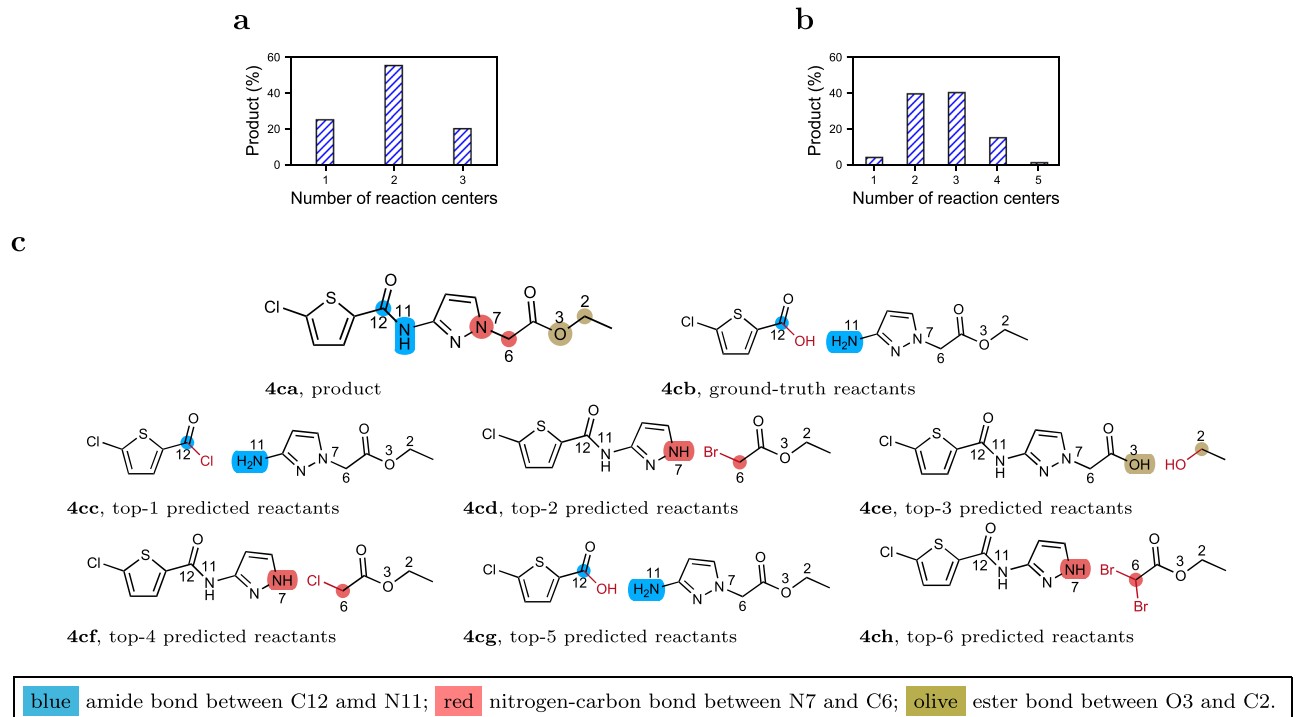

**Fig. 4 Reaction center analysis in predicted reactions and a representative example. a** Percentage of products (Product (%)) with the different number of predicted reaction centers and the third predicted reaction as the ground-truth reaction (i.e., hits at 3); **b** Percentage of products (Product (%)) with the different number of predicted reaction centers and the fifth predicted reaction as the ground-truth reaction (i.e., hits at 5); **c** Predicted reactions by G²Retro for product "CCOC(=O)Cn1ccc(NC(=O)c2ccc(Cl)s2)n". Numbers next to each atom are the indices of the atoms. Atoms with the same indices in different subfigures correspond to each other. Different reaction centers are highlighted in different colors (blue, red and olive). Atoms and bonds colored in red are leaving groups for synthon completion. Molecules with labels ending in (**a**) are product/target molecules; molecules with labels ending in (**b**) are the ground-truth reactants in USPTO-50K; molecules with labels ending in (**c–h**) are the top predicted reactants.

attachments, and therefore potentially different scaffolds and structures in the predicted reactants.

To analyze the diversity of G²Retro results, we analyzed the reaction centers among the top-predicted reactions. We identified a set of products such that their third or fifth predicted reactions are the ground truth, referred to as having a hit at 3 or 5, respectively. Please note each predicted reaction was scored using the sum of the log-likelihoods of all the predictions along the transformation paths from the product to its reactants (please refer to Section "Inference"), and then ranked based on the score. Thus, the predicted reactions ranked above the ground truth have a higher likelihood than the ground truth. Given that G²Retro has demonstrated strong performance as in Table 1 in scoring and prioritizing the ground-truth reactions, we assume that its likelihood calculation is reliable and therefore, the reactions ranked above the ground truth might also be likely to occur.

Figure 4a and b presents the distribution of products with hits at 3 or 5 over the number of reaction centers among predicted reactions ranked above the ground truth. Figure 4a shows that more than 50% of the products with a hit at 3 have their top-3 reactions from two different reaction centers; about 20% of the products have their top-3 reactions from three different reaction centers. Figure 4b shows that for products with a hit at 5, almost 40% have two reaction centers, and another 40% have three reaction centers, among their top-5 predicted reactions; more than 10% have four reaction centers. Thus, Fig. 4a and b clearly demonstrate that the top predicted reactions were diverse, demonstrated by the different reaction centers they were derived from. Meanwhile, we acknowledge that the diverse, top predictions may still be errors and thus, more reliable wet-lab experimental validation is needed.

Figure 4c presents an example of very diverse reactions with diverse reaction centers predicted by G²Retro. For the product in Fig. 4ca, G²Retro predicts three different reaction centers: an amide bond (between C12 and N11), a nitrogen-carbon bond (between N7 and C6) and ester (between O3 and C2). The patent reported that the target molecule was synthesized from a carboxylic acid derivative and an amine using amide coupling with a widely-used coupling reagent, EDC (Fig. 4cb). G²Retro predicted an acyl chloride-amine reactant pair as the top-1 result (Fig. 4cc), a potentially viable and even high yielding synthetic approach. It also predicts three reactant pairs from the other two reaction centers as possible routes within the top 4 (Fig. 4cd and cf at which involve alkylation reactions to form the C6-N7 bond; Fig. 4ce at which forms the ester linkage between O3 and C2).

We also analyzed the reaction diversity by comparing the number of reaction centers in products with high reaction diversity and low reaction diversity. For each product, the diversity of its predicted reactions is represented by the distribution of all pairwise similarities of its predicted reactions, that is, lower reaction similarities indicate higher reaction diversity. Please note that the reaction similarity is only applicable to two reactions that share the same product. Therefore, the product is not considered in the similarity calculation. Formally, for reaction $R_1$: $M_1 + M_2 \rightarrow M_p$ and reaction $R_2$: $M_3 + M_4 \rightarrow M_p$, the similarity between $R_1$ and $R_2$ was calculated as follows,

$$\text{sim}(R_1, R_2) = \frac{1}{2}\max(\text{sim}_m(M_1, M_3) + \text{sim}_m(M_2, M_4),$$
$$\text{sim}_m(M_1, M_4) + \text{sim}_m(M_2, M_3)),$$

(1)

where $\text{sim}_m()$ is a similarity function over molecules, calculated using Tanimoto coefficient over 2,048-bit Morgan fingerprints of the

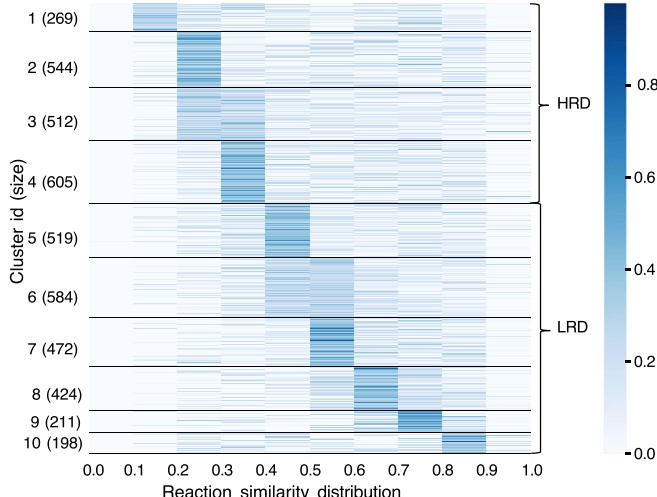

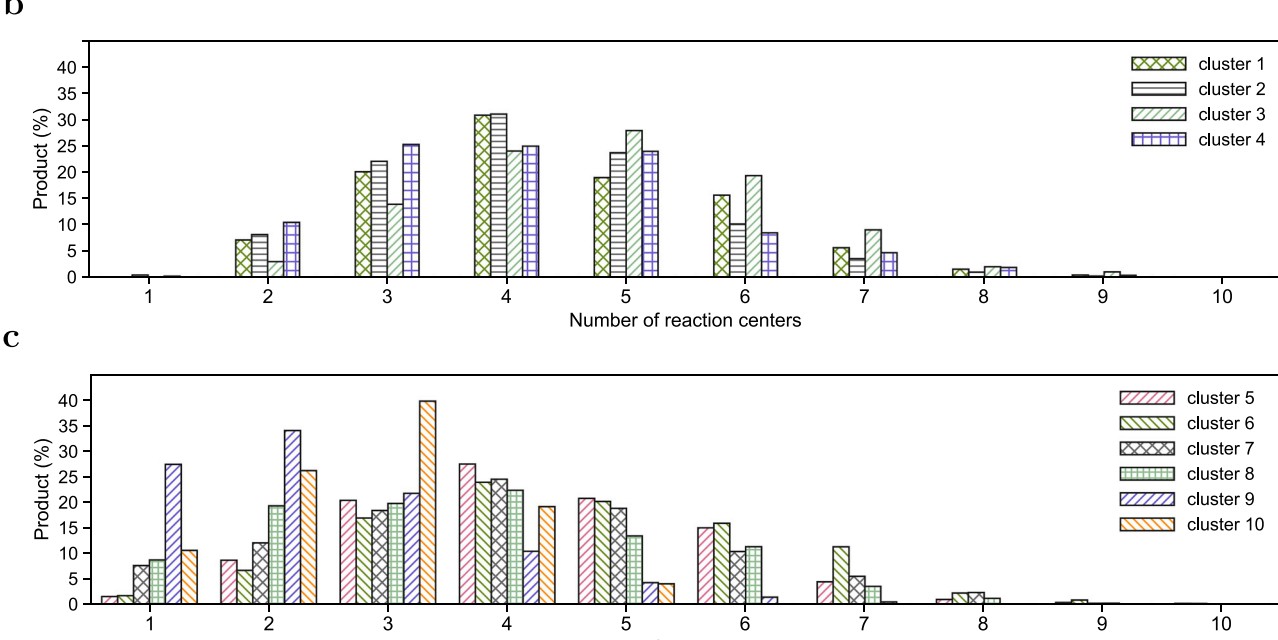

**Fig. 5 Cluster analysis on test products based on similarities of their predicted reactions. a** Clustering on test products based on similarities of their predicted reactions. The *x*-axis indicates the range of reaction similarities (e.g., the column between 0.1 and 0.2 indicates the range (0.1, 0.2]); the *y*-axis shows the cluster ID and the cluster size. Each row in the heatmap corresponds to the reaction similarity distribution of a product belonging to a specific cluster; each block in the row corresponds to the frequency of reaction similarities within each similarity range, and the block color represents the scale of the frequency (e.g., a darker color indicates a higher frequency value). The clusters are labeled as 'HRD' for high-reaction-diversity clusters with average low reaction similarities, and 'LRD' for low-reaction-diversity clusters with average high reaction similarities. **b** Test product distributions over the number of reaction centers of HRD products. **c** Test product distributions over the number of reaction centers of LRD products.

molecules. For reaction $R_1$: $M_1 \rightarrow M_p$ and reaction $R_2$: $M_2 + M_3 \rightarrow M_p$, the similarity between them was calculated as follows,

$$\text{sim}(R_1, R_2) = \text{sim}_m(M_1, M_2 + M_3), \quad (2)$$

where $M_2 + M_3$ denotes the composite molecule consisting of two disconnected components $M_2$ and $M_3$. For reaction $R_1$: $M_1 \rightarrow M_p$ and reaction $R_2$: $M_2 \rightarrow M_p$, the similarity between them was calculated as follows,

$$\text{sim}(R_1, R_2) = \text{sim}_m(M_1, M_2). \quad (3)$$

We clustered the products according to their reaction similarity distributions using the K-means clustering algorithm in Euclidean distances. The clustering algorithm is presented in Supplementary

Algorithm 1 in Supplementary Note 4. Figure 5a presents the clustering results for products that have their ground-truth reaction correctly predicted among the top-10 predictions. In Fig. 5a, the first four clusters have on average lower reaction similarities (on average 0.46 among the four clusters; 0.41, 0.45, 0.45, 0.49 in each of the clusters, respectively), and thus are referred to as high-reaction-diversity clusters (HRD); the other six clusters, referred to as low-reaction-diversity clusters (LRD), have relatively higher reaction similarities (on average 0.58 for among the six clusters; 0.52, 0.53, 0.58, 0.62, 0.67, 0.67 in each of the clusters, respectively).

Figure 5b and c present the distributions of the number of reaction centers in the products of these two clusters. Comparing Fig. 5b and c, HRD products tend to have more reaction centers

in their predicted reactions than those in LRD products, and the number of reaction centers correlates well with reaction diversity (-0.8486 between the average reaction similarities and the number of reaction centers). Particularly, the first cluster (in HRD), which has the highest reaction diversity (lowest reaction similarity), has on average 4.41 reaction centers in the top-10 predicted reactions of each product, compared to the average 3.92 reaction centers in the top-10 predicted reactions of each product in LRD clusters. The ninth and tenth clusters, which have the lowest reaction diversity, have on average 2.57 reaction centers. These results clearly show the diversity of G$^2$Retro predictions.

## Discussion

**Comparison among template-based, template-free and semi-template-based methods**. Template-based methods were first developed for retrosynthesis prediction. They match products into pre-defined templates that are extracted from training data or hand-crafted based on knowledge. A notable advantage of templates is that they can enable strong interpretability (e.g., each template may correspond to a certain reaction type, a chemical scaffold, or a reactivity pattern) and thus result in reactions that better conform to domain knowledge. They can also well fit the data if the templates are extracted from the data. However, they suffer from a lack of strong learning capabilities and a lack of generalizability, if the templates do not cover and cannot automatically discover novel reaction patterns.

Template-free methods largely leverage the technological advancement in Natural Language Processing (NLP), including large-scale language models such as Transformer and BART[53], and also many pre-training techniques. Most of them formulate a reaction as a SMILES string translation problem. Rather than enumerating pre-defined patterns (i.e., templates) as template-based methods do, template-free methods are equipped with much stronger learning capabilities from SMILES strings and can represent latent reaction transformation patterns in an operable manner. However, template-free methods sacrifice their interpretability as it is non-retrieval to decipher why an atom (analogous to a token in NLP) is generated next along the SMILES strings, or what chemical knowledge the actions correspond to. In addition, as SMILES strings are a 'flattened' representation of molecular graphs according to the atom orderings from a graph traversal, template-free methods using SMILES strings only cannot fully leverage molecular structures, which ultimately determine molecule synthesizability and reaction types. To mitigate this issue, some template-free methods either enrich the product SMILES representation with molecular graph information[19,23,25] or decode reactant SMILES strings from product molecular graphs[21], which, however, require additional learning of the mapping from molecular graphs to SMILES and thus increase the learning complexity.

Semi-template-based methods, typically over molecular graphs, represent the most recent and also in general the best performing retrosynthesis prediction methods. They utilize the powerful graph representation learning paradigm to better capture molecule structures. They also take advantage of graph (variational) auto-encoder frameworks or sequential predictions to empower the models with generative ability. More importantly, semi-template-based methods have the mechanism to enable diversity among predicted reactions, by allowing multiple samplings from the latent space. Meanwhile, semi-template-based methods have two steps: (1) reaction center identification, and (2) synthon completion, better complying with how chemical reactions are understood and enabling certain interpretability of predicted reaction centers and derived reactants. G$^2$Retro is a semi-template-based method and achieves superior performance

to other methods, demonstrating it as a state-of-the-art method for retrosynthesis prediction.

**Comparison issues among existing methods**. In our study of the baseline methods, several issues were identified among existing methods that make comparison across different methods hard. In Table 1, RetroXpert's results are from its updated GitHub[54], as their results originally reported in their manuscript had a data leakage issue (all the reaction centers were implicitly given) and thus were overestimated[27]. G2G may also suffer from the data leakage issue as discussed in its github[55], but G2G's results were only available from its original paper, though likely overestimated. In addition, there have been some reproducibility issues with G2G[56], as we also observed in our study. All the methods except Neuralsym, LV-Trans, Dual and Retroformer published their code and datasets. Among these methods, most template-free methods including SCROP, GET, Chemformer, TiedTransformer, GTA and AT used the same data split, which is, however, different from the benchmark data split used in the other methods. For example, the training set of these template-free methods has 40,029 reactions, while the training set of the other methods including G$^2$Retro has 40,008 reactions. Even though all the methods adopted the same ratio (i.e., 80%/10%/10% for training/validation/test set) to split the benchmark dataset, their splits, particularly their test sets, are not identical, making it hard to compare these methods. In this manuscript, we adopted the data split used by the previous semi-template-based methods; for the template-free methods with different data splits, we still used the results reported by their authors. We believe reproducibility and unbiased comparison (e.g., on the same benchmark data and same splits, generating the same amount of results to compare) among all the retrosynthesis prediction methods are critical to moving this research forward. They require dedicated research, implementation and regulatory effort from the entire research community, for example, by following the Open Science Policy from the European Union[57] and the Data Sharing Policy from the United States National Institute of Health[58]. Unfortunately, it is out of the scope of this manuscript.

**Conclusions**. G$^2$Retro predicts reactions of given target molecules by predicting their reaction centers, and then completing the resulting synthons by attaching small substructures. Based on a comparison against twenty baseline methods over a benchmark dataset, G$^2$Retro achieves the state-of-the-art performance under most metrics. The case studies show that G$^2$Retro also enables diverse predictions. However, G$^2$Retro still has several limitations. First, the three types of reaction centers in G$^2$Retro still cannot cover all possible reaction center types (e.g., the reactions with multiple newly formed bonds). Therefore, a more comprehensive definition of reaction center types is still needed. G$^2$Retro cannot cover bonds or rings that are attached at the reaction centers but do not appear in the training data either, as the substructures that G$^2$Retro employs to complete synthons are extracted only from training data. In addition, the atom-mapping between products and reactants that is required by G$^2$Retro (and required by many existing methods) to complete synthons is not always available or of high quality (it is available in USPTO-50K). To identify such mappings, it requires to calculate graph isomorphism, which is an NP-hard problem. Moreover, the sum of log-likelihoods of all the involved predictions (i.e., reaction center prediction, attached atom type prediction) that G$^2$Retro uses to prioritize reactions, is not necessarily the same as the likelihood of the reactions, which could affect the quality of the prioritized reactions. We are also investigating a systemic evaluation and in vitro validation protocol, in addition to using top-$k$ accuracy,

as we discussed earlier. Multiple-step retrosynthesis could be possible by applying G2Retro multiple times iteratively, each time on a reactant as the target molecule. Connected after the deep generative models that have been developed to optimize small molecule structures and properties[59,60] for lead optimization, G2Retro has a great potential to generate synthetic reactions for these in silico generated drug-like molecules, and thus substantially speed up the drug development process.

## Methods

G2Retro is developed for the one-step retrosynthesis prediction problem, that is, given the target molecule (i.e., product), G2Retro identifies a set of reactants that can be used to synthesize the molecule through one synthetic reaction. Following the prior semi-template-based methods[28,29], G2Retro generates reactants from products in two steps. In the first step, G2Retro identifies the reaction center from the target molecule using the center identification module. G2Retro defines the reaction centers as the single bond that is either newly formed or has the bond type changed, or the single atom with changed hydrogen count during the reaction. G2Retro also incorporates into the reaction center the bonds neighboring the reaction centers that have type changes induced by the newly formed bond, and the atoms with charge changes within the target molecule (more details in "Reaction Center Identification" Section). Given the reaction center, G2Retro converts the target molecule into a set of intermediate molecular structures referred to as synthons, which are incomplete molecules and will be completed into reactants. In the second step, G2Retro completes synthons into reactants by sequentially attaching bonds or rings in the synthon completion module. The intermediate molecular structures before being completed to reactants are referred to as updated synthons. Figure 1 presents the overall model architecture of G2Retro. All the algorithms are presented in Supplementary Note 5.

**Molecule representations and notations**. Supplementary Table 3 in Supplementary Note 6 presents the key notations used in this manuscript. A synthetic reaction involves a set of reactants $\{M_r\}$ and a product molecule $M_p$ that is synthesized from the reactants. Please note that we do not consider reagents or catalysts in this study. Each reactant $M_r$ has a corresponding synthon $M_s$, representing the substructures of $M_r$ that appear in $M_p$. We represent the product molecule $M_p$ using a molecular graph $\mathcal{G}_p^M$, denoted as $\mathcal{G}_p^M = (\mathcal{A}, \mathcal{B})$, where $\mathcal{A}$ is the set of atoms $\{a_i\}$ in $M_p$, and $\mathcal{B}$ is the set of corresponding bonds $\{b_{ij}\}$, where $b_{ij}$ connects atoms $a_i$ and $a_j$. We also represent the set of the reactants $\{M_r\}$ or the set of synthons $\{M_s\}$ of $M_p$ using only one molecular graph $\mathcal{G}_r^M$ or $\mathcal{G}_s^M$, respectively. Here, $\mathcal{G}_r^M$ and $\mathcal{G}_s^M$ could be disconnected with each connected component representing one reactant or one synthon.

For synthon completion, we define a substructure $z$ as a bond (i.e., $z = b_{ij}$) or a ring structure (i.e., $z = \{b_{ij}|a_i, a_j \in \text{a single or polycyclic ring}\}$) that is used to complete synthons to reactants. We construct a substructure vocabulary $\mathcal{Z} = \{z\}$ by comparing $\mathcal{G}_r$'s and their corresponding $\mathcal{G}_s$'s in the training data, and extracting all the possible substructures from their differences. In total, G2Retro extracted 83 substructures, covering all the reactions in the test data. Details about these substructures are available in Supplementary Fig. 3 and Supplementary Fig. 4 in Supplementary Note 7. Note that different from templates used in TB methods, the substructures G2Retro used are only bonds and rings, and multiple bonds and rings can be attached to complete a synthon. For simplicity, when no ambiguity arises, we omit the super/sub-scripts and use $\mathcal{G}$ to represent $\mathcal{G}^M$.

**Molecule representation learning**. G2Retro learns the atom representations over the molecular graph $\mathcal{G}$ using the same message passing networks (MPN) as in Chen et al.[59] (Supplementary Algorithm 4 in Supplementary Note 5).

G2Retro first learns atom embeddings to capture the atom types and their local neighborhood structures by passing the messages along the bonds in the molecular graphs. Each bond $b_{ij}$ is associated with two message vectors $\mathbf{m}_{ij}$ and $\mathbf{m}_{ji}$. The message $\mathbf{m}_{ij}^{(t)}$ at $t$-th iteration encodes the messages passing from $a_i$ to $a_j$, and is updated as follows,

$$\mathbf{m}_{ij}^{(t)} = W_1^a \text{ReLU}\left(W_2^a \mathbf{x}_i + W_3^a \mathbf{x}_{ij} + W_4^a \sum_{a_k \in \mathcal{N}(a_i)\setminus\{a_j\}} \mathbf{m}_{ki}^{(t-1)}\right), \quad (4)$$

where $\mathbf{x}_i$ is the atom feature vector, including the atom type, valence, charge, the number of hydrogens, whether the atom is included in a ring and whether the ring is aromatic; $\mathbf{x}_{ij}$ is the bond feature vector, including the bond type, whether the bond is conjugated or aromatic, and whether the bond is in a ring; $W_i^a$'s ($i = 1,2,3,4$) are the learnable parameter matrices; $\mathbf{m}_{ij}^{(0)}$ is initialized with the zero vector; $\mathcal{N}(a_i)$ is the set with all the neighbors of $a_i$ (i.e., atoms connected with $a_i$); and ReLU is the activation function. The message $\mathbf{m}_{ij}^{(t)}$ captures the structure of $t$-hop neighbors passing through the bond $b_{ij}$ to $a_j$, by iteratively aggregating the neighborhood messages $\mathbf{m}_{ki}^{(t-1)}$. With the maximum $t_a$ iterations, G2Retro derives

the atom embedding $\mathbf{a}_i$ as follows,

$$\mathbf{a}_i = U_1^a \text{ReLU}\left(U_2^a \mathbf{x}_i + U_3^a \sum_{a_k \in \mathcal{N}(a_i)} \mathbf{m}_{ki}^{(1\cdots t_a)}\right), \quad (5)$$

where $\mathbf{m}_{ki}^{(1\cdots t_a)}$ denotes the concatenation of $\{\mathbf{m}_{ki}^{(t)}|t \in [1 : t_a]\}$; $U_i^a$'s ($i = 1,2,3$) are the learnable parameter matrices. The embedding of the molecular graph $\mathcal{G}$ is calculated by summing over all the atom embeddings as follows,

$$\mathbf{h} = \sum_{a_i \in \mathcal{G}} \mathbf{a}_i. \quad (6)$$

For $M_p$ and $M_s$, their embeddings calculated from their molecular graphs as above are denoted as $\mathbf{h}_p$ and $\mathbf{h}_s$, respectively.

**Reaction center identification**. Given a product $M_p$, G2Retro defines three types of reaction centers in $M_p$ (Supplementary Algorithm 3 in Supplementary Note 5).

1. a new bond $b_{ij}$, referred to as bond formation center (BF-center), that is formed across the reactants during the reaction but does not exist in any of the reactants;
2. an existing bond $b_{ij}$ in a reactant, referred to as bond type change center (BC-center), whose type changes during the reaction due to the gain or loss of hydrogens, while no other changes (e.g., new bond formation) happen; and
3. an atom in a reactant, referred to as atom reaction center (A-center), from which a fragment is removed during the reaction, without new bond formation or bond type changes.

The above three types of reaction centers cover 97.7% of the training set. The remaining 2.3% of the reactions in the training data involve multiple new bond formations or bond type changes, and will be left for future research. Note that with a single atom as the reaction center, the synthon is the product itself. We refer to all the transformations needed to change a product to synthons as product-synthon transformations, denoted as p2s-T (Supplementary Algorithm 5 in Supplementary Note 5).

*Reaction centers with new bond formation (BF-center)*. Following Somnath et al.[29], G2Retro derives the bond representations as follows,

$$\mathbf{b}_{ij} = U_1^b \text{ReLU}(U_2^b \mathbf{x}_{ij} + U_3^b(\mathbf{a}_i + \mathbf{a}_j) + U_4^b \text{Abs}(\mathbf{a}_i - \mathbf{a}_j)), \quad (7)$$

where Abs($\cdot$) represents the absolute difference; $U_i^b$'s ($i = 1,2,3,4$) are the learnable parameter matrices. G2Retro uses the sum and the absolute difference of embeddings of the connected atoms to capture the local neighborhood structure of bond $b_{ij}$. Meanwhile, the two terms are both permutation-invariant to the order of $\mathbf{a}_i$ and $\mathbf{a}_j$, and together can differentiate the information in $\mathbf{a}_i$ and $\mathbf{a}_j$. With the bond representation, G2Retro calculates a score for each bond $b_{ij}$ as follows,

$$s^b(b_{ij}) = \mathbf{q}^b \text{ReLU}(Q_1^b \mathbf{b}_{ij} + Q_2^b \mathbf{h}_p), \quad (8)$$

where $\mathbf{h}_p$ is the representation of the product graph $\mathcal{G}_p$ calculated as in Eq. (6); $\mathbf{q}^b$ is a learnable parameter vector and $Q_1^b$ and $Q_2^b$ are the learnable parameter matrices. G2Retro measures how likely bond $b_{ij}$ is a BF-center using $s^b(b_{ij})$ by looking at the bond itself (i.e., $\mathbf{b}_{ij}$) and the structure of the entire product graph (i.e., $\mathbf{h}_p$). G2Retro scores each bond in $M_p$ and selects the most possible BF-center candidates $\{b_{ij}\}$ with the highest scores. G2Retro breaks each product at each possible BF-center into synthons, and thus can generate multiple possible reactions.

In synthetic reactions, the formation of new bonds could induce the changes of neighbor bonds. Therefore, G2Retro also predicts whether the types of bonds neighboring the BF-center are changed during the reaction, referred to as the BF-center induced bond type change prediction (BTCP). Given the BF-center $b_{ij}$, the set of the bonds neighboring $b_{ij}$ is referred to as the BF-center neighbor bonds, denoted as $\mathcal{C}_{BF}$, that is:

$$\mathcal{C}_{BF}(b_{ij}) = \{b_{ik}|a_k \in \mathcal{N}(a_i)\setminus\{a_j\}\} \cup \{b_{jk}|a_k \in \mathcal{N}(a_j)\setminus\{a_i\}\}. \quad (9)$$

Thus, G2Retro predicts a probability distribution $\mathbf{f}^b \in \mathbb{R}^{1\times 4}$ for each neighboring bond in $\mathcal{C}_{BF}$, denoted as $b_{i/jk} \in \mathcal{C}_{BF}$, as follows,

$$\mathbf{f}^b(b_{i/jk}) = \text{softmax}(V_1^b \mathbf{b}_{i/jk} + V_2^b \mathbf{b}_{ij} + V_3^b \mathbf{h}_p), \quad (10)$$

where $V_i^b$'s ($i = 1,2,3$) are the learnable parameter matrices. The first element $\mathbf{f}_1^b$ in $\mathbf{f}^b$ represents how likely the $b_{i/jk}$ type is changed during the reaction (It is determined as type change if $\mathbf{f}_1^b$ is not the maximum in $\mathbf{f}^b$), and the other three represent how likely the original $b_{i/jk}$ in the reactant is single, double or triple bond, respectively (these three elements are reset to 0 if $b_{i/jk}$ type is predicted unchanged). Here, G2Retro measures neighbor bond type change by looking at the neighbor bond itself (i.e., $\mathbf{b}_{i/jk}$), the BF-center (i.e., $\mathbf{b}_{ij}$) and the overall product (i.e., $\mathbf{h}_p$). G2Retro updates the synthons $\mathcal{G}_s$ by changing the neighboring bonds of the BF-center to their predicted original types. The predicted changed neighbor bonds are denoted as $\mathcal{C}_{BF}'$.

*Reaction centers with bond type change* (BC-center). If a reaction center is due to a bond type change without new bond formations, G²Retro calculates a score vector $\mathbf{s}^c \in \mathbb{R}^{1 \times 3}$ for each bond $b_{ij}$ in $M_p$ as follows,

$$\mathbf{s}^c(b_{ij}) = Q_1^c \, \text{ReLU}\,(Q_2^c \mathbf{b}_{ij} + Q_3^c \mathbf{h}_p), \tag{11}$$

where $Q_i^c$'s ($i = 1,2,3$) are the learnable parameter matrices. Each element in $\mathbf{s}^c(b_{ij})$, denoted as $s_k^c(b_{ij})$ ($k = 1, 2, 3$), represents, if $b_{ij}$ is the BC-center, the score of $b_{ij}$'s original type in $\mathcal{G}_r$ being single, double, and triple bond, respectively. The element in $\mathbf{s}^c$ corresponding to $b_{ij}$'s type in $\mathcal{G}_p$ is reset to 0 (i.e., $b_{ij}$'s type has to be different in $\mathcal{G}_r$ compared to that in $\mathcal{G}_p$). Thus, the most possible BC-center candidates $\{b_{ij}\}$ and their possible original bond types scored by $\mathbf{s}^c(\cdot)$ are selected. G²Retro then changes the corresponding bond type to construct the synthons.

*Reaction centers with single atoms* (A-center). If a reaction center is only at a single atom with a fragment removed, G²Retro predicts a center score for each atom $a_i$ in $M_p$ as follows,

$$s^a(a_i) = \mathbf{q}^a \, \text{ReLU}\,(Q_1^a \mathbf{a}_i + Q_2^a \mathbf{h}_p), \tag{12}$$

where $\mathbf{q}^a$ is a learnable parameter vector and $Q_1^a$ and $Q_2^a$ are the learnable parameter matrices. G²Retro selects the atoms $\{a_i\}$ in $M_p$ with the highest scores as potential A-center's. In synthon completion, new fragments will be attached at the atom reaction centers.

*Atom charge prediction* (ACP). For all the atoms $a_i$ involved in the reaction center or BF-center changed neighbor bonds $\mathcal{C}_{BF}'$, G²Retro also predicts whether the charge of $a_i$ remains unchanged in reactants. G²Retro uses an embedding $\mathbf{c}$ to represent all the involved bond formations and changes in $p2s$-T. If the reaction center is predicted as a BF-center at $b_{ij}$, G²Retro calculates the embedding $\mathbf{c}$ as follows,

$$\mathbf{c} = \sum_{b_{kl} \in \mathcal{C}_{BF}'(b_{ij}) \cup \{b_{ij}\}} W_1^c \, \text{ReLU}\,(W_2^c \mathbf{x}_{kl}' + W_3^c \mathbf{b}_{kl}), \tag{13}$$

where $\mathcal{C}_{BF}'$ is a subset of $\mathcal{C}_{BF}$ with all the bonds that changed types; $\mathbf{x}_{kl}'$ is a $1 \times 4$ one-hot vector, in which $\mathbf{x}_{kl}'(0) = 1$ if bond $b_{kl}$ is the bond formation center (i.e., $b_{kl} = b_{ij}$), or $\mathbf{x}_{kl}'(i) = 1$ ($i = 1, 2, 3$) if $b_{kl}$ type is changed from single, double or triple bond in reactants, respectively, during the reaction (i.e., $b_{kl}$ is in $\mathcal{C}_{BF}'(b_{ij})$); $W_i^c$'s ($i = 1, 2, 3$) are the learnable parameter matrices.

If the reaction center is predicted as a BC-center at $b_{ij}$, $\mathbf{c}$ is calculated as follows,

$$\mathbf{c} = W_1^c \, \text{ReLU}\,(W_2^c \mathbf{x}_{ij}' + W_3^c \mathbf{b}_{ij}), \tag{14}$$

where $\mathbf{x}_{ij}'(0) = 0$ and $\mathbf{x}_{ij}'(i) = 1$ ($i = 1, 2, 3$) if $b_{kl}$ type is changed from single, double or triple bond in reactants, respectively, during the reaction. If the reaction center is an A-center, no $p2s$-T are needed and thus $\mathbf{c} = \mathbf{0}$.

With the embedding $\mathbf{c}$ for $p2s$-T, G²Retro calculates the probabilities that $a_i$ will have charge changes during the reaction as follows,

$$\mathbf{f}^c(a_i) = \text{softmax}\,(V_1^c \mathbf{a}_i + V_2^c \mathbf{c}), \tag{15}$$

where $V_1^c$ and $V_2^c$ are the learnable parameter matrices; $\mathbf{f}^c \in \mathbb{R}^{1 \times 3}$ is a vector representing the probabilities of accepting one electron, donating one electron or no electron change during the reaction. The option corresponding to the maximum value in $\mathbf{f}^c$ is selected and will be applied to update synthon charges accordingly. G²Retro considers at most one electron change since this is the case for all the reactions in the benchmark data.

*Reaction center identification module training*. With the scores for three types of reaction centers, G²Retro minimizes the following cross entropy loss to learn the above scoring functions (i.e., Eqs. (8), (11) and (12)),

$$\mathcal{L}^s = - \sum_{b_{ij} \in \mathcal{B}} \left( y_{ij}^b l^b(b_{ij}) + \sum_{k=1}^{3} \mathbb{I}_k(y_{ij}^c) l_k^c(b_{ij}) \right) - \sum_{a_i \in \mathcal{A}} y_i^a l^a(a_i), \tag{16}$$

where $y^*$ ($x = a, b, c$) is the label indicating whether the corresponding candidate is the ground-truth reaction center of type $*$ ($y^* = 1$) or not ($y^* = 0$); $\mathbb{I}_k(x)$ is an indicator function ($\mathbb{I}_k(x) = 1$ if $x = k$, 0 otherwise), and thus $\mathbb{I}_k(y_{ij}^c)$ indicates whether the ground-truth bond type of $b_{ij}$ is $k$ or not ($k = 1, 2, 3$ indicating single, double or triple bond); and $l^*(\cdot)$ ($* = a, b$)/($l_k^c(\cdot)$) is the probability calculated by normalizing the score $s^*(\cdot)/s_k^c$, that is, $l^*(x) = \exp(s^*(x))/\Delta$, where $\Delta = \sum_{b_{ij} \in \mathcal{B}}(\exp(s^b(b_{ij})) + \sum_{k=1}^{3} \exp(s_k^c(b_{ij}))) + \sum_{a_i \in \mathcal{A}} \exp(s^a(a_i))$ ($l_k^c(x) = \exp(s_k^c(x))/\Delta$). Similarly, G²Retro also learns the predictor $\mathbf{f}^b(\cdot)$ for neighbor bond changes (Eq. (10)) and $\mathbf{f}^c(\cdot)$ for atom charge changes (Eq. (15)) by minimizing their respective cross entropy loss $\mathcal{L}^b$ and $\mathcal{L}^c$. Therefore, the center identification module learns the predictors by solving the following optimization problem:

$$\min_{\Theta} \mathcal{L}^s + \mathcal{L}^b + \mathcal{L}^c, \tag{17}$$

where $\Theta$ is the set of all the parameters in the prediction functions. We used Adam

algorithm to solve the optimization problem and do the same for the other training objectives.

**Synthon completion**. Once the reaction centers are identified and all the product-synthon transformations ($p2s$-T) are conducted to generate synthons from products, G²Retro completes the synthons into the reactants by sequentially attaching substructures (Supplementary Algorithm 6 in Supplementary Note 5). All the actions involved in this process are referred to as synthon-reactant transformations. During the completion process, any intermediate molecules $\{M^*\}$ are represented as molecular graph $\{\mathcal{G}^*\}$. At step $t$, we denote the atom in the intermediate molecular graph $\mathcal{G}^{*(t)}$ ($\mathcal{G}^{*(0)} = \mathcal{G}_s$) that new substructures will be attached to as $a^{(t)}$, and denote the substructure attached to $a^{(t)}$ as $z^{(t)}$, resulting in $\mathcal{G}^{*(t+1)}$.

*Atom attachment prediction*. The algorithm for atom attachment prediction is presented in Supplementary Algorithm 8 in Supplementary Note 5. G²Retro first predicts whether further attachment should be added to $a^{(t)}$ or should stop at $a^{(t)}$, referred to as the atom attachment continuity prediction (AACP), with the probability calculated as follows,

$$f^o(a^{(t)}) = \sigma(V_1^o \mathbf{a}^{(t)} + V_2^o \mathbf{h}_s + V_3^o \mathbf{h}_p), \tag{18}$$

where

$$\mathbf{h}_s = \sum_{a_i \in \mathcal{G}_s} \mathbf{a}_i. \tag{19}$$

In Eq. (18), $\mathbf{a}^{(t)}$ is the embedding of $a^{(t)}$ calculated over the graph $\mathcal{G}^{*(t)}$ (Eq. (5)); $\mathbf{h}_s$ is the representation for all the synthons as in Eq. (19); $V_i^o$'s ($i = 1,2,3$) are the learnable parameter matrices; $\sigma$ is the sigmoid function. In Eq. (19), G²Retro calculates the representations by applying MPN over the graph $\mathcal{G}_s$ that could be disconnected, and the resulted representation is equivalent to applying MPN over each $\mathcal{G}_s$'s connected component independently and then summing over their representations. G²Retro intuitively measures "how likely" the atom has a new substructure attached to it by looking at the atom itself (i.e., $\mathbf{a}^{(t)}$), all the synthons (i.e., $\mathbf{h}_s$), and the product (i.e., $\mathbf{h}_p$). Note that in Eq. (18), BRICS fragment information (i.e., $\mathbf{a}'$ as in Eq. S3 in Supplementary Note 1) is not used because the fragments for the substructures that will be attached to $a^{(t)}$ will not be available until the substructures are determined.

If $a^{(t)}$ is predicted to attach with a new substructure, G²Retro predicts the type of the new substructure, referred to as the atom attachment type prediction (AATP), with the probabilities of all the substructure types in the vocabulary $\mathcal{Z}$, calculated as follows,

$$\mathbf{f}^z(a^{(t)}) = \text{softmax}\,(V_1^z \mathbf{a}^{(t)} + V_2^z \mathbf{h}_s + V_3^z \mathbf{h}_p), \tag{20}$$

where $V_i^z$'s ($i = 1,2,3$) are the learnable parameter matrices. Higher probability for a substructure type $z$ indicates that $z$ is more likely to be selected as $z^{(t)}$. The atoms $a \in z^{(t)}$ in the attached substructure are stored for further attachment, that is, they, together with any newly added atoms along the iterative process, will become $a^{(T)}$ ($T = t + 1, t + 2, \cdots$) in a depth-first order in the retrospective reactant graphs. G²Retro stops the entire synthon completion process after all the atoms in the reaction centers and the newly added atoms are predicted to have no more substructures to be attached.

*Synthon completion model training*. G²Retro trains the synthon completion module using the teacher forcing strategy, and attaches the ground-truth fragments instead of the prediction results to the intermediate molecules during training. G²Retro learns the predictors $f^o(\cdot)$ (Eq. (18)) and $\mathbf{f}^c(\cdot)$ (Eq. (20)) by minimizing their cross entropy losses $\mathcal{L}^o$ and $\mathcal{L}^z$ as follows:

$$\min_{\Phi} \mathcal{L}^o + \mathcal{L}^z, \tag{21}$$

where $\Phi$ is the set of parameters.

**Inference**. The algorithm for G²Retro inference is presented in Supplementary Algorithm 2 in Supplementary Note 5.

*Top-K reaction center selection*. During the inference, G²Retro generates a ranked list of candidate reactant graphs $\{\mathcal{G}_r\}$ (note that each reactant graph can be disconnected with multiple connected components each representing a reactant). With a beam size $K$, for each product, G²Retro first selects the top-$K$ most possible reaction centers from each reaction center type (BF-center, BC-center and A-center), and then selects the top-$K$ most possible reaction centers from all the $3K$ candidates based on their corresponding scores (i.e., $s^b$ as in Eq. (8) for BF-center, $\mathbf{s}^c$ as in Eq. (11) for BC-center, and $s^a$ in Equation (12) for A-center). Then G²Retro converts the product graph $\mathcal{G}_p$ into the top-$K$ synthon graphs $\{\mathcal{G}_{s,i}\}_{i=1}^{K}$ accordingly. Different reaction centers lead to diverse synthons. For these synthon graphs, neighbor bond type change is predicted when necessary; atom charge change is predicted for all the atoms involved in reaction centers and their neighboring bonds $\mathcal{C}_{BF}$ for BF-center's. All the bond type changes and atom charge changes are predicted as those with the highest probabilities as in Eq. (10) and Eq. (15), respectively.

*Top-N reactant graph generation*. Once the top-$K$ reaction centers for each product are selected and their synthon graphs are generated, G$^2$Retro completes the synthon graphs $\{\mathcal{G}_{s,i}\}_{i=1}^{K}$ into reactant graphs. During the completion, G$^2$Retro scores each possible reactant graph and uses their final scores to select the top-$N$ reactant graphs, and thus top-$N$ most possible synthetic reactions, for each product. Since during synthon completion, the attachment substructure type prediction (Eq. (20)) gives a distribution of all possible attachment substructures; by using top possible substructures, each synthon and its intermediate graphs can be extended to multiple different intermediate graphs, leading to exponentially many reactant graphs and diversity in the predicted reactions. The intermediate graphs are denoted as $\{\mathcal{G}^{*(t)}_{ij}\}_{i=1}^{K}$, where $\mathcal{G}^{*(t)}_{ij}$ is for the $j$-th possible intermediate graph of the $i$-th synthon graph $\mathcal{G}_{s,i}$ at step $t$. However, to fully generate all the possible completed reactant graphs, excessive computation is demanded. Instead, G$^2$Retro applies a greedy beam search strategy (Supplementary Algorithm 7 in Supplementary Note 5) to only explore the most possible top reactant graph completion paths.

In the beam search strategy, G$^2$Retro scores each intermediate graph $\mathcal{G}^{*(t)}_{ij}$ using a score $s^{(t)}_{ij}$, which is calculated as the sum over all the log-likelihoods of all the predictions along the completion path from $\mathcal{G}_s$ up to $\mathcal{G}^{*(t)}_{ij}$; $s^{(0)}_{ij}$ is initialized as the sum of the log-likelihoods of all the predictions from $\mathcal{G}_p$ to $\mathcal{G}_s$. At each step $t$ ($t \leq 30$), each intermediate graph $\mathcal{G}^{*(t)}_{ij}$ is extended to at most $N+1$ intermediate graph candidates. These $N+1$ candidates include the one that is predicted to stop at the atom that new substructures could be attached to (i.e., as $a^{(t)}$ in Eq. (18); this intermediate graph could be further completed at other atoms) in this step, and at most $N$ candidates with the top-$N$ predicted substructures attached (Eq. (20)). Among all the candidates generated from all the intermediate graphs at step $t$, the top-$N$ scored ones will be further forwarded into the next completion step $t+1$. In case some of the top-$N$ graphs are fully completed, the remaining will go through the next steps. This process will be ended until the number of all the completed reactant graphs at different steps reaches or goes above $N$. Then, among all the incomplete graphs at the last step, the intermediate graphs with log-likelihood values higher than the $N$-th largest score in all the completed ones will continue to complete as above. The entire process will end until no more intermediate graphs are qualified to further completion. Among all the completed graphs, the top-$N$ graphs are selected as the generated reactants.

**Related work**. Deep-learning-based retrosynthesis prediction methods are typically categorized into three classes: template based (TB), template free (TF) and semi-template based (Semi-TB).

*Template-based methods*. Template-based methods formulate the retrosynthesis problem as a selection problem over a set of reaction templates. These templates can be either hand-crafted by experts[61] or automatically extracted from known reactions in databases[12–16]. Szymkuc et al.[61] provided a review on using reaction templates coded by human experts for synthetic planning. However, these rules may not cover a large set of reactions due to the limitation of human annotation capacity. Recent template-based methods extract reaction templates automatically from databases. With the reaction templates available, Coley et al.[12] (Retrosim) selected the reaction templates that the corresponding reactions in the database have the products most similar with the target molecules, in order to synthesize the target molecules. Dai et al.[14] learned the joint probabilities of templates matched in the product molecules and all its possible reactants using two energy functions, one for reaction template scoring and the other for reactant scoring conditioned on templates. Seidl et al.[15] (MHNreact) learned to associate the target molecule with the relevant reaction templates using a modern Hopfield network. Chen et al.[16] (LocalRetro) scored the suitability of all the reaction templates at all the potential reaction centers (atoms and bonds) in the target molecule. The use of templates provides interpretability toward the reasoning behind the generated reactions. However, these templates also limit the template-based methods to the reactions only covered by the templates.

*Template-free methods*. Template-free methods directly learn to transform the product into the reactants without using the reaction templates[17–22,24,25,31]. Most template-free methods utilize the sequence representations of molecules (SMILES) and formulate the transformation between the product and its corresponding reactants as a sequence-to-sequence problem. Many SMILES-based methods use Transformer[33], a language model with attention mechanisms to model the relationship across tokens. Transformer follows the encoder-decoder architecture, which encodes the product SMILES string into a latent vector and then decodes the vector into the reactant SMILES strings. For example, Kim et al.[22] (TiedTransformer) learned the transformation from a product to its reactants using two coupled Transformers with shared parameters, one for the forward product prediction (synthesis) and the other for the backward reactant prediction (retrosynthesis). During the inference, they leveraged both the forward and backward models to find the best reactions. Sun et al.[24] (Dual) transformed a product to its reactants using an energy-based framework. They also leveraged the duality of the forward and backward models by training them together and selected the best reactions with the highest energy value from the two models. Tetko et al.[31]

(AT) learned to transform a product into its reactants using a Transformer trained on a dataset augmented with various non-canonical SMILES representations of each molecule. In AT, each target molecule was tested multiple times using different SMILES string representations. Zhong et al.[32] (R – SMILES) aligned the product and reactant SMILES strings to minimize their edit distance, and trained a transformer to decode the reactant SMILES strings from the products. They also augmented the training dataset and tested each target molecule multiple times as in AT. In addition to SMILES-based template-free methods, Sacha et al.[26] (MEGAN) formulated retrosynthesis as a graph editing process from a product to its reactants. These graph edits include the change in the atom properties or the bond types, or the addition of the new atoms or the benzene rings into the synthons. These template-free methods are independent of reaction templates, and thus they may have better generalizability to unknown reactions compared to template-based methods. However, template-free methods lack interpretability toward the reasoning behind their end-to-end predictions. SMILES-based template-free methods also suffer from the validity issue that the generated sequences may fail to follow the grammar of SMILES strings or violate chemical rules[17].

*Semi-template-based methods*. Semi-template-based methods[26–30] do not use reaction templates, or they do not directly transform a product into its reactants. Instead, semi-template-based methods follow a two-step workflow utilizing atom-mappings: (1) they first identify the reaction centers and transform the product into synthons (intermediate molecules) using the reaction centers; and then (2) they complete the synthons into the reactants. Shi et al.[28] (G2G) first predicted reaction centers as bonds that can be used to split the product into the synthons, and then utilized a variational autoencoder[35] to complete synthons into reactants by sequentially adding new bonds or new atoms. Somnath et al.[29] (GraphRetro) predicted the bonds with changed bond types or the atoms with changed hydrogen count as the reaction centers, and then completed the synthons by selecting the pre-extracted subgraphs that realize the difference between synthons and reactants. Wang et al.[30] (RetroPrime) formulated the reaction center identification and synthon completion problems as two sequence-to-sequence problems (i.e., product to synthon, and synthon to reactant), and trained two Transformers for these problems, respectively. The prediction of reaction centers first in the above methods allows better interpretability toward the reasoning behind the generation process. The two-step workflow also empowers these methods to diversify their generated reactants by allowing multiple different reaction center predictions forwarded into their synthon completion step.

G$^2$Retro also identifies the reaction centers and then completes the synthons into the reactants in a sequential way as G2G does. However, G$^2$Retro is different from G2G. G$^2$Retro can cover multiple types of reaction centers while G2G takes only the newly formed bonds as the reaction center, which leads to lower coverage of G2G on the dataset. During synthon completion, G$^2$Retro attaches substructures (e.g., rings and bonds) instead of single atoms as in G2G, into synthons to simplify the completion process. In addition and more importantly, G$^2$Retro uses other synthons of the same reaction and also the product to complete a synthon, and thus the synthon completion is more contextualized for the product, while G2G does not consider other synthons.

*Fragment-based molecule generation*. Following the idea of fragment-based drug design[62,63], fragment-based molecule generation methods have been developed. For example, Jin et al.[64] first decomposed a molecular graph into a junction tree of chemical substructures, and then used a variational autoencoder over the junction trees and its chemical substructures to generate and assemble new molecules (JT-VAE). Podda et al.[65] encoded and decoded a sequence of fragments via a variational autoencoder, and generated new molecules by connecting fragments generated from the autoencoder. Chen et al.[59] optimized a molecule by removing and attaching substructures in a starting molecule. G$^2$Retro generates reactants from synthons also by attaching new substructures. However, the generation strategy in G$^2$Retro is fundamentally different from that in the previous fragment-based molecule generation methods. During synthon completion, G$^2$Retro does not encode the synthons using their substructures as what JT-VAE and Modof do. It does not either encode or decode the substructures that are to be attached to the synthons. Instead, G$^2$Retro attaches the substructures to a specific, identified atom in the molecular graph of the synthons. Therefore, G$^2$Retro can directly attach a substructure to the predicted reaction centers.

**Data preprocessing and experimental settings**. We used the benchmark dataset provided by Yan et al.[27]. This dataset, also referred to as USPTO-50K, contains 50K chemical reactions that are randomly sampled from a large dataset collected by Lowe[11] from US patents published between 1976 and September 2016. Each reaction in the large dataset is atom-mapped so that each atom in the product is uniquely mapped to an atom in the reactants. The 50K reactions in USPTO-50K are classified into 10 reaction types by Schneider et al.[66]. To avoid the information leakage issue[27] (e.g., reaction center is given in both the training and test data), all the product SMILES strings in USPTO-50K are canonicalized. We used exactly the same training/validation/test data splits of USPTO-50K as in the previous methods[12,27], which contain 40K/5K/5K reactions, respectively. Table 5 presents the data statistics. We trained G$^2$Retro models on the 40K training data, with parameters tuned on the 5K validation data, and tested on the 5K test data. For

**Table 5 USPTO-50K data statistics.**

| Dataset | | Statistics |
|---|---|---|
| # Training reactions | | 40,008 |
| # Validation reactions | | 5001 |
| # Test reactions | | 5007 |
| Training reactions | Average size of products | 26.0 |
| | Average size of larger reactants | 21.9 |
| | Average size of smaller reactants | 9.0 |
| | Average number of reactants | 1.7 |
| Validation reactions | Average size of products | 25.9 |
| | Average size of larger reactants | 21.8 |
| | Average size of smaller reactants | 9.1 |
| | Average number of reactants | 1.7 |
| Test reactions | Average size of products | 25.9 |
| | Average size of larger reactants | 21.7 |
| | Average size of smaller reactants | 9.2 |
| | Average number of reactants | 1.7 |

reproducibility purposes, details about model training and parameter tuning are provided in Supplementary Note 8.

**Baselines.** We compared G$^2$Retro with the state-of-the-art baseline methods for the one-step retrosynthesis problem, including five template-based (TB) methods, ten template-free (TF) methods and five semi-template-based (Semi-TB) methods. Inspired by the recent success of using fragments in other tasks[67], we further extended G$^2$Retro into G$^2$Retro-B by incorporating the fragments generated from the breaking retrosynthetically interesting chemical substructures (BRICS) fragmentation algorithm[68]. Details of G$^2$Retro-B are available in Supplementary Note 1. The experimental setting for G$^2$Retro-B is identical to that of G$^2$Retro.

**Template-based baseline methods** The five TB baseline methods include Retrosim, Neuralsym, GLN, MHNreact and LocalRetro. These methods first mine reaction templates from training data and apply only these templates to construct reactants from the target molecule.

- Retrosim[12] selects the templates of reactions that produce molecules most similar to the target molecule.
- Neuralsym[13] predicts suitable templates using product fingerprints through a multi-layer perceptron.
- GLN[14] predicts reactions using two energy functions, one for template scoring and the other for reactant scoring conditioned on templates.
- MHNreact[15] learns the associations between molecules and reaction templates using modern Hopfield networks, and selects templates based on the associations.
- LocalRetro[16] selects templates against each atom and each bond using classifiers.

**Template-free baseline methods** The ten TF baseline methods all use Transformer over SMILES string representations of products and/or reactants.

- SCROP[17] maps the SMILES strings of products to the SMILES strings of reactants using a Transformer, and then corrects syntax errors (e.g., mismatch of parentheses in SMILES strings) to ensure valid reactant SMILES strings.
- LV-Trans[18] pre-trains a vanilla Transformer using reactions generated from templates, and then fine-tunes the Transformer with a multinomial latent variable representing reaction types.
- GET[19] trains standard Transformer encoders and decoders using the combined atom representations learned from molecular graphs and from SMILES strings.
- Chemformer[20] translates product SMILES strings into reactant SMILES strings using Transformer, which is pre-trained on an independent dataset to recover masked SMILES strings (i.e., with some atoms masked out) or to normalize augmented SMILES strings (i.e., multiple, equivalent non-canonical SMILES strings for each SMILES string).
- Graph2SMILES[21] encodes molecular graphs using graph neural networks with attention mechanisms, and decodes the reactant SMILES strings from the graph representations using a Transformer decoder.
- TiedTransformer[22] uses two Transformers with shared parameters to learn the transformation from products to reactants and vice versa, respectively, and selects the best reactions using the likelihood values from these two Transformers.
- GTA[23] enhances a Transformer with truncated attention connections regulated by molecular graph structures.
- Dual[24] uses an energy-based model with two Transformers to learn the transformation from product SMILES strings to reactants' SMILES strings and vice versa, and selects the best reactions using the energy.

- Retroformer[25] predicts the reaction center region using a reaction center detection module, and uses the embedding of predicted centers as a condition to transform via Transformer the product into the reactants in SMILES. Although Retroformer predicts the reaction center, it does not split products into synthons using the reaction center, and thus does not follow a two-step, semi-template-based framework.
- MEGAN[26] transforms the product molecular graphs into the corresponding reactant graphs using a sequence of graph edits (e.g., change atom charges, add a new bond) that are learned from products and their reactants in the training set.

**Semi-template-based methods**
The five Semi-TB baseline methods all use molecular graph representations. Most of them explicitly predict reaction centers first.

- RetroPrime[30] trains two Transformers independently to predict the transformation from the product to its synthons and from the synthons to the reactants, respectively.
- RetroXpert[27] predicts reaction centers on molecular graphs via a graph attention network, and transforms resulting synthons to reactants using a Transformer.
- G2G[28] predicts reaction centers on molecular graphs via a graph neural network, and completes synthons into reactants through sequential additions of new atoms or bonds using the latent variables sampled from the latent space of a variational graph autoencoder.
- GraphRetro[29] predicts reaction centers via a message passing neural network over molecular graphs, and completes synthons by selecting the subgraphs in a vocabulary that realize the difference between the synthons and reactants.

## Data availability
The data used in this paper are available publicly[69] at the link https://doi.org/10.5281/zenodo.7839013 and the link https://github.com/ninglab/G2Retro.

## Code availability
The code for G$^2$Retro, G$^2$Retro-B and G$^2$Retro-ens is available publicly[69] at the link https://doi.org/10.5281/zenodo.7839013 and the link https://github.com/ninglab/G2Retro. A web portal for G$^2$Retro is available at the link http://go.osu.edu/G2Retro.

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

## Acknowledgements

This project was made possible, in part, by support from the National Science Foundation grant nos. IIS-2133650 (X.N.), and The Ohio State University President's Research Excellence program (X.N., H.S.). Any opinions, findings and conclusions or recommendations expressed in this paper are those of the authors and do not necessarily reflect the views of the funding agency. We thank Dr. Michael A. Walters for his constructive comments.

## Author contributions

X.N. conceived the research. X.N. and H.S. obtained funding for the research. Z.C. and X.N. designed the research. Z.C. and X.N. conducted the research, including data curation, formal analysis, methodology design and implementation, result analysis and visualization. Z.C. and X.N. drafted the original paper. O.R.A. and J.R.F. provided comments on case studies. H.S. provided comments on the original paper. Z.C. and X.N. conducted the paper editing and revision. All authors reviewed the final paper.

## Competing interests

The authors declare no competing interests.
