## [Peer review file · Communications Chemistry]

G²Retro as a two-step graph generative model for retrosynthesis predictionEditorial Note: Parts of this Peer Review File have been redacted as indicated to maintain the confidentiality of proprietary data.

Reviewers' comments:

Reviewer #1 (Remarks to the Author):

The manuscript describes a machine learning architecture to propose a pair of reactants able to transform into a target product. The manuscript suffers from its awkward introduction, but remains in general clear and pleasant to read. The two main comments are the following.

- The manuscript reports a model (G2Retro) that is an improvement on existing methods (see page 3, "Semi-template-based methods"). It is also limited to a specific retro-synthesis sub-task ("propose possible direct reactants for the synthesis of the target molecules"); as a consequence, the audience is limited. In my opinion, the contribution would be more suited for a publication such as Journal of Cheminformatics.

- As I understood from "Data processing and experimental settings", the contribution used existing USPTO data as prepared from former publications. Therefore, the manuscript does not bring new data. It does not bring new curation of existing data. It does not bring new understanding of existing data.

- There are no experimental proofs. This is not a problem for me, but it can be a reason to reject the contribution in publications such as JCIM.

Below are some major revisions I request.

General remark 1: the manuscript tone makes me feel uneasy because it sometimes sounds more like an advertisement than a scientific publication.

General remark 2: The G2Retro-B must be removed from the contribution since it is insufficiently described, validated and illustrated. It only blurs the picture. It can be proposed as supplementary information though. Figures must be adapted.

General remark 3: The G2Retro is not illustrated on rearrangement, isomerisation, cyclisation or click reactions that are less obvious than those chosen in the manuscript. But it may be out of scope considering that the reaction center definition used is "the single bond that is either newly formed or has the bond type changed".

General remark 4: In the introduction specifically, there are many sentences whose formulation makes them hard to understand. Below are two examples:

-- "In predicting reaction centers, G2Retro learns from the molecular graphs of the products and encodes their molecule structures most indicative of multiple reaction center types." -> As I understand, the authors mention the training of machine learning on a dataset of chemical reactions. The model outputs are hypothetical reaction centers, as substructures where bond and atoms are changing types, during the reaction.

-- "G2Retro leverages a semi-template-based approach, predicts multiple types of reaction centers in product first, and converts the resulted synthons into reactants, best imitating the reversed logic of synthetic reactions, and enabling step-wise interpretability" -> As I understand it, the model implements a so-called "semi-templated" approach. This approach is based on the detection of reaction centers of various types (such as?). From the reaction center, the model deduces substructures (synthons) that are used as seeds to generate molecular structures that are interpreted as reactants.

Reading the abstract and the introduction I had to guess many times what was the meaning of the authors. The manuscript must be carefully proof-read.

General remark 5: on some occasions the term "significant" is used as an emphasis and can be

confused with the result of a rigorous statistical test. For instance page 5: "(...) G2Retro (...) is significantly better than RetroPrime on all other metrics".

page 1: "Current retrosynthesis analysis is (...) conducted by (...) chemists (...) which could be (...) susceptible to human error". This statement is not only flat; it also brings the expectation that the proposed G2Retro system performs better than human and this is not proven in the manuscript. It is needed to remove or reformulate this statement.

page 1: the authors mention only Reaxys and SciFinder as sources of knowledge on chemical reaction and insist on their price. This is a weak argument for two reasons: there is an open source of chemical reaction (see 10.1021/jacs.1c09820) even though it is more modest; SciFinder is an expression of the ACS which is a not-for-profit organisation (see the financial overview of the ACS). I perceived this financial argument as an attempt to gain adhesion of the reader based on emotion rather than reason. Therefore, the paragraph must be changed.

page 3: "Template-free methods are independent of reaction templates and thus have better generalizability". This seems a personal opinion of the authors. If it is not supported by facts or by literature, it must be emphasized as a personal opinion.

page 4: "The reactants are considered correct if (...) they are considered reasonable based on domain knowledge". I did not find where this "domain knowledge" is defined and, to me, it seems to allow an arbitrary definition of what is a "correct" reactant. This ambiguity must be solved in the manuscript.

Table 3 mentions the coverage and not the table 2. It seems inconsistent and table 2 should also mention the coverage.

It is not clear why the results reported in table 3 and table 2 differ. I understand that table 3 performances are measured combining the reaction center and the synthon completion. But these precisions are missing from the captions of table 2 and 3. The captions of these two tables are insufficient and must be updated.

Figure 3a and Figure 3e are overlapping.

In the case study, it seems that the G2Retro uses reaction type labeling. The description of the case study must be clear if it uses "a one-hot encoder as an additional feature (...) indicating the reaction type".

page 10 "The above examples (...) demonstrate that G2Retro performance is underestimated". I cannot agree with this statement because there are no experimental evidence that the reactions proposed occurred in comparable conditions to those reported in the US patents. Therefore, the question if the performances are under- or overestimated is not decided.

page 10 "To analyze the diversity of G2Retro results we identified products such that their third or fifth predicted reactions are ground truth (...) it is likely that their top-3 or top-5 predicted reactions are also possible". The unknown predicted reactions cannot be "likely possible". It is impossible to assess without reaction conditions. They can be however termed as "working hypothesis". Therefore, the diversity may also be illustrative of "errors" of the model if these hypothesis are not useful.

Page 11 the clustering description is not clear. The authors seem to have clustered reactants from diverse products. In that case, I don't understand what conclusions can be made.

The discussion is a bit weak because the authors bring criticisms on practices that they do not help to solve themselves. For instance, they mention that experimental validation should be done, but they did not perform it themselves. However, I don't request modifications.

For the above reasons, I believe that the contribution is better suited to a more specialized journal. However, if the edit wish so, it is needed to bring to this manuscript some major modifications.

Reviewer #2 (Remarks to the Author):

Key Results

G2Retro presents a method for one-step retrosynthesis prediction consisting of two modules: a reaction center selection module proposing multiple candidates as well as a synthon completion block outputting the final reactants. The method falls into the category of semi-template-based approaches. The method shows strong performance and compares against current state-of-the-art methods.

Validity

G2Retro follows previous methods by dividing the one-step retrosynthesis prediction into a two-step procedure. The method is solid, and well introduced in the supplementing Method section. The paper is well written. The literature and related work section are complete and the authors are well informed on current developments as well as state-of-the-art methods. The chosen datasets are recognized benchmarks and the chosen comparison to other methods is fair. In particular, the authors point out current flaws in existing pipelines (i.e., RetroPrime and G2G) which suffer from an existing bias in the dataset. While one could argue that the method could be tested in a more large-scale setting, these are the usually chosen settings and a well-suited test environment to evaluate the method's performance.

My main criticism involves the last sections of the Experimental Results. As the proposed method is a semi-template-based method, it is always a stretch to assume diverse reactants. It would be great if the authors could emphasize and reiterate why they assume G2Retro to increase the diversity of reactants over previous methods.

While investigating the top-3 or top-5 predicted reactants if the ground truth is not contained, the analysis conducted is not clear. In particular, Figure 4 ("Product (%)") is not clear to me. Could the authors clarify and potentially improve the description of that section? It is not clear how the conclusions the authors make can be drawn from the conducted analysis.

Further, the purpose of the section on "In vitro validation" is unclear to me. Why did the authors not attempt further analysis to overcome the limitations of the top-k accuracy analysis? Would it be possible for the authors, to conduct further analysis in this direction? If not, why?

In the first part of the Experimental Results section, when comparing to previous methods, it is necessary to also cite each method (despite being done before). This improves reading. Similarly, the table is easier to pass if citations are added here as well.

Minor comment: You state that neither Neuralsym, LV-Trans, Dual, and Retroformer published their code. G2G should be added to this list as well, otherwise, I do not understand why the authors did not rerun G2G on the debiased dataset.

Originality and Significance

Building upon previous results, G2Retro

While the authors conduct a very thorough comparison to existing work, it might be worth pointing out that previous methods as well have divided retrosynthesis prediction into two modules (reaction center identification and synthon completion), i.e., as done in 26 and 7.

Data & Methodology

The presentation of the method is clear. Standard datasets are chosen. The hyperparameters are published in the Supplement. Additional algorithms are provided to improve the general understanding of the method. Code is released.

Conclusions and Suggested Improvements

The presented method is sound and, after addressing the concerns raised above, I recommend acceptance.

Reviewer #3 (Remarks to the Author):

Chen and colleagues present a machine learning approach to one-step retrosynthesis prediction, named G2Retro. The method builds upon previous semi-template-based methods, in particular G2G and GraphRetro, and uses a two-step approach in which, first, multiple reaction centers are identified to break the product molecule into synthons, and then a generative process selects substructures that complete the synthons to yield the reactants. The authors compare their approach to several other computer-aided retrosynthesis methods, showing competitive performance. Overall, the work appears technically sound and the manuscript is fairly well written. However, I felt that the novelty and impact of the work was often overstated, and one key method comparison appeared to be missing from the discussion. My main suggestion would be to revise the manuscript to better and more fairly contextualize the work and reduce speculation.

Here below I summarize the major issues I believe should be addressed before publication, followed by more minor suggestions.

(1) This work mostly builds on top of G2Gs and GraphRetro, but the main novelty seems to be a more crafted definition of reaction centers which can cover a greater portion of test molecules. The author claimed the deep neural networks are "novel, customized" (page 2) but the GMPN and FMPN seem to be simple versions of MPNNs (e.g., unlike RetroXpert which considers edge update, or GraphRetro which uses gated and directed updates). The reactant recovery stage makes use of a similar graph generation process as G2Gs, albeit with fragments. However, also the novelty of this aspect is not too clear (see next comment). The work might be publishable even if the technical novelty is limited, but what the novel aspects are should be stated clearly and without exaggeration.

(2) The contextualization of fragment-based molecular generation methods is missing. In particular, the synthon-to-reactants graph is formulated as sequential attachment of fragments starting from the synthons. This approach seems very similar to that of junction-tree VAEs (<https://proceedings.mlr.press/v80/jin18a>). However, this and related generative methods are not mentioned in the text, which makes it hard to understand whether the approach used by the authors is per se novel or whether it is an adaptation/application of existing techniques.

(3) The paper "Root-aligned SMILES: a tight representation for chemical reaction prediction" (Zhong et al., Chem. Sci. 2022), previously published on arXiv (<https://arxiv.org/abs/2203.11444>) reports a template-free method that seems to provide higher performance than G2Retro on USPTO-50K across the board with reaction type unknown. The results from this study should be included and discussed too, and the text adapted to reflect how G2Retro is not necessarily the top performing approach.

(4) The fact the model proposes non-ground truth reactants with known transformation is claimed across the text as evidence that the approach can discover novel reactions. This claim seems unwarranted and highly speculative without any prospective validation, and as such I would remove it from the manuscript. Perhaps, the claim may be rephrased as to say that the method can propose novel synthetic routes for known compounds, which is different from discovering new reactions.

(5) Table 3 and comparison with GraphRetro: it appears that for synthon completion, GraphRetro still outperforms G2Retro. The reason for G2Retro's overall better performance was mentioned as being that "GraphRetro does not do well in reaction center identification". But if that's the case, what's the advantage of G2Retro's synthon completion process over that of GraphRetro since it looks more complicated, and worse?

(6) As the author stated on page 3, MEGAN "does not follow the two-step workflow", does not make use of the half-template, and is essentially template-free. It thus seems odd to categorize MEGAN as a semi-template method. Unless the authors have a strong argument for classifying MEGAN as semi-template-based, the text should probably be adjusted accordingly.

Minor issues, in rough order of importance:

(8) Some statements do not seem to be entirely correct, in particular:

- On page 2, the authors discuss template-free methods and mention that "SMILES strings cannot fully leverage molecular structures, which ultimately determine molecule synthesizability and reaction types". However, there are models that make use of graph information and the full molecular structures of the targets with their graph encoders (e.g., Retroformer, GTA, GET, Graph2SMILES). It might be better to be a bit more nuanced here.

- At page 12, the author mentioned that semi-template-based methods "utilize the most advanced graph representation learning paradigm", however, it is unclear in what way these are the most advanced. None of these works made use of the most expressive graph encoder (e.g., GIN), nor did they use pretraining/contrastive learning techniques that are standard for representation learning (e.g., GROVER, DMP). Therefore, if no additional clarification is provided, it would be better to avoid this statement.

(9) I'd suggest removing claims about how the approach would speed up drug discovery. It is unclear to what extent drug discovery is slowed down by synthetic planning, so unless these claims are backed up with some data/reference, it seems to me they are unjustified.

(10) Page 12: "In general, template-based methods underperform template-free and semi-template-based methods." Is this statement correct? Table 2 would suggest that LocalRetro is in fact very competitive.

(11) In the Conclusions: "G2Retro also enables diverse predictions and novel reactions that were accessed as very possible by synthetic chemists". In addition to the issue of claiming the prediction of novel reactions, it is also unclear who the synthetic chemists were, how many, and how the study was designed. It seems like the reactions were simply analyzed by the authors. As such, I would remove this claim as it makes it sound as if an external panel of chemists evaluated the reactions in some rigorous fashion.

(12) Table 2: G2Retro-B does not seem to be introduced when presenting the method but is encountered only in the Results. It might be good to anticipate how it is different from G2Retro before its results are presented in Table 2 and discussed at page 6.

(13) My understanding is that the performance comparison to other methods (Table 2) is based on what is reported in the literature, as opposed to being based on re-runs of all the experiments with the various methods. It might be good to make this very explicit at some point in the text.

(14) On page 1, "auto-encoders" are mentioned but the associated ref. 10 is about "variational autoencoders", which are distinct from general auto-encoders. It might be good to specify whether the authors are referring to variational auto-encoders specifically.

(15) Contextualization of Retroformer: The way it's described on page 3 (reaction center detection

followed by decoding) makes it sound like a semi-template-based method. Some elaboration is needed to explain why it's categorized as template-free.

(16) The reference for BART on Page 12 is missing.

(17) Figure 3: the header of panel "e" is on top of the molecular structure of panel "a".

(18) "Discussion and conclusions" should probably be renamed just "Discussion" since there is a section called "Conclusion" later on.

(19) The header "Materials" might not be too apt for a computational study, and could perhaps be called "Methods" (with then "Methods" becoming "Extended Methods"), or something else that does not imply wet-lab experiments, like "Dataset and baselines"?

(20) At page 11, but also earlier in the text, it is discussed how the fact that the test set used by various methods are not the exactly the same results in "unfair" comparison. "Unfair" might not be the most appropriate word, as it implies some specific methods are at a disadvantage. It might be better to replace it with something else that implies it is hard to compare methods tested in slightly different ways, but not necessarily "unfair".

Summary of major revisions:

We sincerely thank the reviewers for all the positive feedbacks and constructive comments. We believe the comments are greatly helpful for us to further improve the quality of our manuscript, and for us to design the future research and make real impacts! We have substantially incorporated the comments in this revised manuscript.

In this document, we are providing point-to-point responses to all of the reviewers' comments, and providing our corresponding revised materials in the revised manuscript: the revised materials will be quoted and highlighted in red with references removed in this document if space allows; otherwise, only the Section titles will be quoted and highlighted, and please refer to the revised main text and Supplementary Information for details.

Together with this document, we are submitting a marked manuscript in which all the revisions have been highlighted in red. In the revised submission, we have made substantial changes and improvement, summarized as follows (sections, figures and tables with references starting with 'S' indicate those in the Supplementary Information):

- (1) We substantially improved the entire manuscript and the Supplementary Information in writing, and added substantially more materials including new methods, experiments, analyses and discussions.
- (2) We included **a new ensemble approach, referred to as G^2 Retro-ens**, which allows each product molecule to be tested multiple times by multiple G^2 Retro models as the other baseline methods with testset augmentation do.
 - (i) We included a new subsection **"Performance of Ensemble-based Methods"** in the main text, in which we compared G^2 Retro-ens with baseline methods AT and R-SMILES that used data augmentation, and demonstrated that G^2 Retro-ens can achieve competitive or even better results compared to AT and R-SMILES.
 - (ii) We included a new section **Section S3** in the Supplementary Information, in which we described G^2 Retro-ens in detail and compared the performance of G^2 Retro-ens and R-SMILES on different reaction types in **Table S2**.
- (3) We extended the **"Related Work"** section in the main text. We included a new subsection **"Fragment-based Molecule Generation"** to contextualize the fragment-based molecule generation and discuss the differences between G^2 Retro and previous fragment-based molecule generation methods.
- (4) We extended the **"Diversity on predicted reactions"** section in the main text. We included more discussions on the importance of diversity for one-step retrosynthesis model and clarified the description of clustering on products according to the reaction diversity.

Comments from Reviewer 1:

Comment #1.1: “The manuscript describes a machine learning architecture to propose a pair of reactants able to transform into a target product. The manuscript suffers from its awkward introduction, but remains in general clear and pleasant to read.”

Response: We sincerely thank the reviewer for all the detailed comments and constructive suggestions. We have substantially improved the introduction according to the reviewer’s suggestions. Please refer to our response to **Comment #1.8**, **Comment #1.10** and **Comment #1.11** below.

Comment #1.2: “The manuscript report a model (G²Retro) that is an improvement on existing methods (see page 3, ‘Semi-template-based methods’). It is also limited to a specific retro-synthesis sub-task (‘(propose) possible direct reactants for the synthesis of the target molecules’); as a consequence, the audience is limited. In my opinion, the contribution would be more suited for a publication such as Journal of Cheminformatics.”

Response: We thank the reviewer for the comments. Here, we would like to clarify the contributions of our methods as well as the target audience.

Contributions and innovations

- Our method G²Retro falls within the category of semi-template-based methods, but it is **not** an improvement of any existing semi-template-based methods. Instead, it is a **new** semi-template-based method for one-step retrosynthesis prediction. Compared to the existing semi-template-based methods, G²Retro defines a comprehensive set of reaction center types, and develops new and customized functions for reaction center identification and synthon completion (e.g., graph convolution method in Section “Atom Embedding over Molecular Graphs (GMPN)”; new fragment-based generation strategy for synthon completion; embeddings from synthetically accessible fragments for enriched representations). All of these are new and have never been developed in semi-template-based methods. Please refer to our “Related Work” Section and “Overall Comparison” Section for a more detailed comparison between G²Retro and the existing methods.
- One-step retrosynthesis prediction has become a well-defined research topic, demonstrated by the 21 baseline methods compared in our manuscript, many of them being published in high-impact venues (e.g., ACS Central Science, Nature Communications, Chemical Science, and Journal of Chemical Information and Modeling). It has attracted substantial attention from both the artificial intelligence (AI) research community and the medicinal chemistry community in the last few years. Many referred works were published in the last four years, involving AI experts and medicinal chemists (e.g., Tetko, I. V., Karpov, P., Deursen, R. V. Godin, G. State-of-the-art augmented NLP transformer models for direct and single-step retrosynthesis. *Nat. Commun.* **11**, 5575 (2020)).
- One-step retrosynthesis is the **foundation** of synthesis planning. By itself, it is a **critical** and **non-trivial** problem. As for now, the problem of high-throughput, accurate one-step retrosynthesis prediction has not been fully solved using computational approaches, as discussed in our “Introduction” Section. In this revision, we included the following discussion to emphasize the importance and challenges of one-step retrosynthesis.

“One-step retrosynthesis, which transforms a molecule into the possible direct reactants that can be used to synthesize the molecule, serves as the foundation of multi-step synthesis planning that identifies a full synthesis route in which the target molecule can be made through a series of one-step synthesis reactions.”

We acknowledge the importance of multi-step retrosynthesis planning. However, to the best of our knowledge, there has not been any existing work on predictive or generative approaches (not rule-based) that address the problem of large-scale, high-throughput multi-step retrosynthesis planning or predict the full retrosynthesis paths that start from commercially available molecules. Therefore, one-step retrosynthesis is especially meaningful and important.

- G²Retro achieves competitive or superior performance compared to **21** existing methods representing the state of the art on retrosynthesis prediction. In our study, we have conducted a very comprehensive set of studies, including
 - detailed comparisons with the state of the art,

- studies on the two modules (reaction center identification, synthon completion),
- studies on different reaction types,
- **model ensembles**,
- case studies for several predicted reactions, and
- studies on prediction diversity.

In addition, we identified several issues in existing work, and proposed potential solution directions, including

- the difficulties in comparing some existing work, and
- the need for *in vitro* evaluation, its difficulties and potential future research directions for *in vitro* evaluation.

We also **published our source code**, data and all the hyper-parameters to ensure the reproducibility of our work. We believe our work can contribute new knowledge and perspectives to the research community.

Target audience We believe our work has a broad audience from medicinal chemistry, synthetic chemistry, organic chemistry and computational chemistry, all being the targeted audience of *Communications Chemistry* as described in its Aims & Scope (<https://www.nature.com/commschem/aims>). Our approach could be applied to help medicinal chemists by proposing a set of possible reactions to synthesize a drug candidate. As a matter of fact, many reactions and products involved in the benchmark data that we used are for drug or drug-like molecule synthesis. In addition, synthetic chemists and organic chemists can use the synthetic routes proposed by our approaches as a reference point, or select from these routes for their applications.

We also want to point out that *Communications Chemistry* publishes research of similar nature as our work's, for example,

- Korshunova, M., Huang, N., Capuzzi, S. *et al.* Generative and reinforcement learning approaches for the automated de novo design of bioactive compounds. *Commun Chem* **5**, 129 (2022).
- Tempke, R., Musho, T. Autonomous design of new chemical reactions using a variational autoencoder. *Commun Chem* **5**, 40 (2022).

Korshunova et al “propose[ed] several technical innovations to [...] improve the balance between exploration and exploitation modes in reinforcement learning. In a proof-of-concept study, [they] demonstrate the application of the deep generative recurrent neural network architecture enhanced by several proposed technical tricks to design inhibitors of the epidermal growth factor (EGFR).” In Tempke and Musho, “an artificial intelligence model based on a Variational AutoEncoder (VAE) has been developed and investigated to synthetically generate continuous [reaction] datasets. The approach involves sampling the latent space to generate new chemical reactions.” These published works are based on deep learning and generative models (reinforcement learning, VAE) for molecule design and reaction generation. Our work is on new deep generative model development for retrosynthesis of given products, sharing similar natures with these published works in that they are on deep generative models for drug design and synthesis. Therefore, we believe our work would share similar audience as the published, similar work in *Communication Chemistry*. Note that Tempke and Musho do not generate reactions for given targeted products, but new reactions that could involve arbitrary, unknown reactants and products.

Comment #1.3: “As I understood from ‘Data processing and experimental settings’, the contribution used existing USPTO data as prepared from former publications. Therefore, the manuscript does not bring new data. It does not bring new curation of existing data. It does not bring new understanding of existing data.”

Response: We thank the reviewer for this comment. We want to clarify that the contribution of this manuscript is not about new chemical data generation or existing data curation, but about the **development of a new computational method** that can perform one-step retrosynthesis prediction better than existing methods. We used the existing benchmark dataset to train and evaluate our method – this has become the common practice in all the existing retrosynthesis prediction methods. In this revision, we improved the description of the scope and contributions of our method. Please refer to the “**Introduction**” Section.

We also want to point out that we have published our **source code**, the data, and the hyper-parameters for the model and result reproducibility. All the above, together with the analysis results and discussions in this manuscript, should be considered as “scientific data.” This is based on multiple definitions as follows:

- The United States Office of Management and Budget Circular A-110, and the Office of Science and Technology Policy memorandum, define data in scientific research as:

“scientific data’ include the recorded factual material commonly accepted in the scientific community as of sufficient quality to validate and replicate research findings.”

Please refer to <https://www.whitehouse.gov/wp-content/uploads/2022/08/08-2022-OSTP-Public-Access-Memo.pdf> for more details.

- The United States National Institute of Health (NIH) Data Management and Sharing Policy defines scientific data as:

“The recorded factual material commonly accepted in the scientific community as of sufficient quality to validate and replicate research findings, regardless of whether the data are used to support scholarly publications. ”

Please refer to <https://grants.nih.gov/grants/guide/notice-files/NOT-OD-21-013.html> for more detail.

- The National Policy of the Republic of Cyprus for Open Access to Scientific Information defines scientific data as follows:

“Science Data is the primary information, namely the data or numbers which were collected and are considered as a basis for reflection, discussion or calculation in order to carry out a scientific research. Examples of scientific data include statistical data, results of experiments, measurements, observations resulting from field research, survey results, recordings of interviews and images, with emphasis on data available at digital form.”

This is a definition aligned with Open Science Policies in Europe.

Please refer to <https://eprints.gla.ac.uk/222719/1/222719.pdf> and https://research-and-innovation.ec.europa.eu/strategy/strategy-2020-2024/our-digital-future/open-science_en for more detail.

Since the scientific data produced in our study have not been produced by anyone else in the literature, they should be considered as “new data.” So we believe our manuscript brings new data.

Comment #1.4: “There are no experimental proof. This is not a problem for me, but it can be a reason to reject the contribution in publications such as JCIM.”

Response: We appreciate that the reviewer raised concerns about experimental proof. We appreciate the generosity of the reviewer not mandating experimental proof. We fully agree with the reviewer that experimental proof is inevitable in the end to validate the synthetic paths predicted by the computational methods. However, unfortunately, experimental validation for all possible predictions (e.g., 50,070 predicted reactions for the benchmark testing data) to fully prove our methods is going to be extremely time-consuming, costly, and not scalable. In this revision, we have included the following discussion in the “Discussions” Section on this issue:

“Only a few previous studies have validated the synthetic reactions predicted by computational methods through laboratory experiments. However, their validation was limited to a small scale of eight synthetic paths involving 51 reactions in total. To the best of our knowledge, there have not been any studies in which synthetic chemists, based on their chemistry knowledge, evaluate all the 50,070 predicted reactions for the 5,007 products in the benchmark dataset used by the current computational methods, or any *in vitro* experiments in a web-lab setting that validate all of them. An in-house estimation we conducted shows that to *in vitro* validate one reaction, it takes on average more than \$100 cost on chemical reagents, solvents, chromatography supplies and lab supplies, excluding costs on lab staff. It also takes a lab staff at least 125 years to finish the experiments, all the related analyses and documentation. Therefore, it is extremely costly and time-consuming to conduct *in vitro* validation for all the predicted reactions.”

In addition, as we pointed in our response to **Comment #1.2**, publications in *Communications Chemistry* of similar work to ours did not include full experimental proof.

Comment #1.5: “General remark 1: the manuscript tone makes me feel uneasy because it sometime sounds more an advertising than a scientific publication.”

Response: We thank the reviewer for this feedback. In this revision, we have substantially revised the manuscript and changed the manuscript tone, as summarized below.

- (1) We removed all the inaccurate statements or statements without factual support. Please refer to our response to **Comment #1.13**, **Comment #1.18**, **Comment #3.6**, **Comment #3.11** and **Comment #3.12** for examples.

- (2) We summarized our contributions in a more objective way. Please refer to our response to **Comment #3.3**.
- (3) We revised the manuscript to improve clarity. Please refer to our response to **Comment #1.8, Comment #1.9, Comment #1.14, Comment #1.15, Comment #1.17, Comment #1.19, Comment #1.20, Comment #3.13** and **Comment #3.14** for examples.
- (4) We added necessary references to support our arguments. Please refer to our response to **Comment #1.10**. for example.
- (5) We removed the statements that are not closely related to the key idea of our method. Please refer to our response to **Comment #3.10** for example.
- (6) When there were our opinions, suggestions or comments, we explicitly mentioned they were our opinions. Please refer to our response to **Comment #1.12** for example.

Comment #1.6: “General remark 2: The G2Retro-B must be removed from the contribution since it is insufficiently described, validated and illustrated. It only blurs the picture. It can be proposed as supplementary information though. Figures must be adapted.”

Response: We thank the reviewer for this comment and suggestion. In this revision, we have removed the claim on G²Retro-B as a major contribution. We have also moved the description of G²Retro-B to **Section S3 in Supplementary Information** and substantially updated the description. We have also updated **Figure 1** in the main text accordingly. Please refer to **Section S3 in Supplementary Information** for more detail.

Comment #1.7: “General remark 3: The G2Retro is not illustrated on rearrangement, isomerisation, cyclisation or click reactions that are less obvious than those chosen in the manuscript. But it may be out of scope considering that the reaction center definition used is ‘the single bond that is either newly formed or has the bond type changed’.”

Response: We sincerely appreciate the reviewer’s comment. We agree with the reviewer that rearrangement and other mentioned reaction types are important. We want to clarify that cyclisation and click reactions belong to the heterocycle formation reaction type, and we have reported in Table 4 that G²Retro cannot cover the heterocycle formation reaction type. For rearrangement and isomerisation reactions, the benchmark dataset we used to build G²Retro does not include these reaction types (included reaction types are listed in Table 4) and therefore, we do not have the data to train models to handle these reaction types.

We agree with the reviewer that the current reaction center definition has limited coverage on some reaction types. In the original manuscript, we discussed this limitation of G²Retro on specific reaction types in Section “Conclusions” as follows,

“[...]the three types of reaction centers in G²Retro still cannot cover all possible reaction center types (e.g., the reactions with multiple newly formed bonds). Therefore, a more comprehensive definition of reaction center types is still needed.”

Comment #1.8: “General remark 4: In the introduction specifically, there are many sentences which formulation makes them hard to understand. Below are two examples:

– ‘In predicting reaction centers, G2Retro learns from the molecular graphs of the products and encode their molecule structures most indicative of multiple reaction center types.’ As I understand, the authors mention the training of machine learning on a dataset of chemical reaction. The model output are hypothetical reaction centers, as substructures where bond and atoms are changing types, during the reaction.

– ‘G2Retro leverages a semi-template-based approach, predicts multiple types of reaction centers in product first, and converts the resulted synthons into reactants, best imitating the reversed logic of synthetic reactions, and enabling step-wise interpretability’ As I understand it, the model implements a so-called ‘semi-templated’ approach. This approach is based on the detection of reaction centers of various types (such as?). From the reaction center, the model deduces substructures (synthons) that are used as seeds to generate molecular structures that are interpreted as reactants.

Reading the abstract and the introduction I had to guess many times what was the meaning of the authors. The manuscript must be carefully proof-read.”

Response: We thank the reviewer for this feedback. In this revision, we have substantially revised the entire “**Introduction**” Section. We have also proof-read the revision to ensure better readability.

Especially, for the first example the reviewer mentioned, we have revised the paragraph as follows:

“**To predict reaction centers, G²Retro learns from the molecular graphs of the products via a customized graph**

representation learning and embedding approach (in 'Molecule Representation Learning' Section), and uses the graph structures to predict potential reaction centers.”

For the second example, we have revised the paragraph as follows:

“G²Retro follows a semi-template-based framework, predicts reaction centers of different types in products first, and then transforms the resulting synthons into reactants by adding substructures to the synthons. This process imitates the reversed logic of synthetic reactions and enables necessary interpretability as to which reaction centers are predicted by G²Retro, which reactants are generated from the reaction centers and the corresponding step-by-step generation process.”

Comment #1.9: “General remark 5: on some occasion the term ‘significant’ is used as an emphaser and can be confused with the result of a rigorous statistical test. For instance page 5: ‘(...) G2Retro (...) is significantly better than RetroPrime on all other metrics’.”

Response: We sincerely thank the reviewer for this comment. We agree with the reviewer that the term “significant” could be confused with statistical testing. To address this issue, we have replaced the term “significant” with “substantial” or “great”, for example,

“While G²Retro underperforms RetroPrime on one metric, it is **substantially** better than RetroPrime on all the other metrics.”

“While these are different substrates, the variation of the ester side chain in these cases would not be typically be considered as **greatly** different by a synthetic chemist unless steric or electronic contributions affect the reactivity/electrophilicity of the ester carbonyl.”

Comment #1.10: “Current retrosynthesis analysis is (...) conducted by (...) chemists (...) which could be (...) susceptible to human error’. This statement is not only flat; it also bring the expectation that the proposed G2Retro system performs better than human and this is not proven in the manuscript. It is needed to remove or reformulate this statement.”

Response: We sincerely thank the reviewer for this suggestion. In this revision, we have reformulated the statement as follows:

“It has been long known that there exists substantial disagreement among chemists in accessing synthesibility and designing synthesis routes. In addition, an ever-increasing number of new chemical reactions makes it highly challenging for a chemist to keep up to date. Therefore, a data-driven model that predicts synthetic reactions could provide a useful complement to chemist evaluations, and could provide a large pool of potential reactions that the chemists can consider. ”

Comment #1.11: “the authors mention only Reaxys and SciFinder as sources of knowledge on chemical reaction and insist on their price. This is a weak argument for two reasons: there is an open source of chemical reaction (see 10.1021/jacs.1c09820) even though it is more modest; SciFinder is an expression of the ACS which is a not-for-profit organisation (see the financial overview of the ACS). I perceived this financial argument as an attempt to gain adhesion of the reader based on emotion rather than reason. Therefore, the paragraph must be changed.”

Response: We appreciate the reviewer for this great comment. We also thank the reviewer for providing the additional material 10.1021/jacs.1c09820 (i.e., Open Reaction Database). In this revision, we included the discussion on Open Reaction Database (ORD) as follows:

“Open-sourced synthesis reaction databases such as the Open Reaction Database are limited in the reactions they cover (e.g., majorities are USPTO public reactions) and their search functionalities (e.g., via SMILES strings).”

We particularly want to point out that the majorities of ORD reactions are from USPTO (about 1.8 million ORD reactions out of all its 2.2 million reactions are from USPTO). ORD provides the infrastructure where users can upload their reactions. The ORD authors also made efforts to collect and integrate reaction data published in scientific articles. However, as of now, the scale of ORD is still very limited, and its search functions and results are also limited.

As a matter of fact, **SciFinder is not free** because they own the **proprietary reaction data**. Please refer to their pricing information here: <https://www.cas.org/products/scifinder/pricing>. As a matter of fact, it is expensive. [Redaction due to proprietary information included]. This is another reason why we were not able to do a systemic evaluation and validation of our predicted reactions beside top-*k* accuracy, even at a medium scale.

Comment #1.12: “Template-free methods are independent of reaction templates and thus have better generalizability’. This seems a personal opinion of the authors. If it is not supported by facts or by literature, it must be emphasized as a personal opinion.”

Response: We thank the reviewer for this great suggestion. This is our personal opinion. In this revision, in order not to confuse the audience, we have revised this statement as below,

“These template-free methods are independent of reaction templates, and thus they may have better generalizability to unknown reactions compared to template-based methods.”

Comment #1.13: “The reactants are considered correct if (...) they are considered reasonable based on domain knowledge’. I did not find where this ‘domain knowledge’ is defined and, to me, it seems to allow an arbitrary definition of what is a ‘correct’ reactant. This ambiguity must be solved in the manuscript.”

Response: We sincerely appreciate the reviewer for the rigorous review. In this revision, we have resolved the ambiguity in the definition of correct reactants as follows,

“The reactants are considered correct only when the reactions are reported in benchmark data.”

“Please note that while there is always one ground-truth reaction for each product in the benchmark data, there may exist actually numerous feasible reactions for each product that are not included in the benchmark data. Therefore, reactants that are considered incorrect based on the benchmark data might still be plausible and included in other larger databases.”

Comment #1.14: “Table 3 mention the coverage and not the table 2. It seems inconsistent and table 2 should also mention the coverage.”

Response: We thank the reviewer for this comment. We agree with the reviewer that it is important to compare the overall coverage among methods. In this revision, we have added the coverage to the Table 2. In the “Comparison with semi-template-based (Semi-TB) methods” Section, we have also included the following discussion to compare the overall coverage among semi-template-based methods:

“As Table 2 shows, G²Retro and G²Retro-B can cover (i.e.g, can be applied to) 97.5% of the test reactions, which determines the upper bound of accuracy values, due to the definition of reaction centers (the rest 2.5% correspond to reactions with multiple newly formed or changed bonds). Among other Semi-TB methods, G2G and GraphRetro also have limited coverage on test set (97.9% for G2G and 95.0% for GraphRetro). RetroXpert has 100% coverage because its reactant SMILES generation from synthons recovers all possible reaction centers. RetroPrime also has 100% coverage due to its very comprehensive set of reaction centers. Although G²Retro and G²Retro-B cannot cover all possible cases in the test set, they still outperform other Semi-TB methods, measured over the entire test set. More discussion on the coverage of the two steps in Semi-TB methods is available in the Section ‘Individual Module Performance’.”

In the “Comparison with template-free (TF) methods” Section, we have included the following discussion to compare the coverage between TF methods and G²Retro:

“In terms of the coverage on the test set, all the SMILES-based TF methods can cover the entire test set, because all the reactions can be represented as SMILES string transformation. The graph-based TF method MEGAN also covers the entire test set due to its comprehensive set of graph edit actions. Compared to these TF methods, though without the full coverage on the test set, G²Retro and G²Retro-B better imitate the reversed logic of synthetic reactions with two steps: reaction center identification and synthon completion. Overall, G²Retro and G²Retro-B achieve even better performance than the methods with full coverage, measured on the entire test set.”

In the “Comparison with template-based (TB) methods” Section, we have included the following discussion on the coverage of TB methods:

“In terms of coverage on the test set, Table 2 shows that the templates used in Retrosim, Neursym and MHNreact can cover the entire test set, while the templates used in GLN and LocalRetro cannot (93.3% for GLN and 98.1% for LocalRetro).”

Comment #1.15: “It is not clear why the results reported in table 3 and table 2 differ. I understand that table 3 performances are measured combining the reaction center and the synthon completion. But these precisions are missing from the captions of table 2 and 3. The captions of these two tables are insufficient and must be updated.”

Response: We thank the reviewer for this careful review. We want to clarify that Table 2 has the overall results of all the methods including template-free, template-based and semi-template-based methods. If the methods are semi-template based with two steps: reaction center prediction and synthon completion, their performance in

Table 2 corresponds to the predictions out of the two steps, that is, the synthon completion is done according to a reaction center that is *predicted* from the reaction center prediction step. In the “Overall Comparison” Section, we have included the description of Table 2 as follows:

“Particularly, for Semi-TB methods, the performance in Table 2 corresponds to the predictions out of the two steps, that is, the synthon completion is done according to a reaction center that is *predicted* from the reaction center prediction step.”

Table 3 has the results of only semi-template-based methods. The performance in Table 3 was measured on the two steps separately. That is, the synthon completion is done according to the *ground-truth* reaction center, not the *predicted* reaction center; the performance measured here is only for the synthon completion step without any effects from the reaction center prediction step entangled. The goal is to measure the two prediction steps independently. This is also the reason why the results from Table 2 and 3 are not identical. Please note that here “ground-truth” reaction center means the reaction center as appears in the benchmark data per our reaction center definition. In the “Individual Module Performance” Section, we have included the description of Table 3 as follows:

“Following the typical evaluation for Semi-TB methods as in literature, Table 3 presents the **individual** performance of the two modules - reaction center identification and synthon completion in Semi-TB methods. **In Table 3, for the reaction center identification module, the top-*k* accuracy measures the ratio of test products that have the ground-truth reaction center correctly predicted among the top-*k* predictions. In the synthon completion module, the synthon completion is done according to the *ground-truth* reaction center, not the *predicted* reaction center; the top-*k* accuracy measures the ratio of test products that have the ground-truth reactants correctly predicted among the top-*k* predictions. Please note that here ‘ground-truth’ reaction center means the reaction center as appears in the benchmark data per our reaction center definition.**”

Accordingly, we have revised the Table 2 and Table 3 captions as follows:

“Overall comparison **on retrosynthesis prediction** in top-*k* accuracy (%)”

“Module performance comparison **on reaction center identification and synthon completion** in top-*k* accuracy (%)”

Comment #1.16: “Figure 3a and Figure 3e are overlapping.”

Response: We thank the reviewer for the careful check. In this revision, we have addressed this issue by rearranging Figure 3a and Figure 3e. We also checked other figures to make sure there are not overlapped or have any format issues.

Comment #1.17: “In the case study, it seems that the G2Retro uses reaction type labeling. The description of the case study must be clear if it uses ‘a one-hot encoder as an additional feature (...) indicating the reaction type’.”

Response: We are very grateful for the detailed review. We clarify that the examples presented in the case study are from G²Retro with the reaction types unknown. In the revision, we have revised the paragraph below accordingly,

“we have highlighted two molecules and their predicted reactions by G2Retro **with reaction type unknown** in Figure 2 and 3, respectively.”

Comment #1.18: “The above examples (...) demonstrate that G2Retro performance is underestimated’. I cannot agree with this statement because there are no experimental evidence that the reactions proposed occurred in comparable conditions to those reported in the US patents. Therefore, the question if the performances are under- or overestimated is not decided.”

Response: We really appreciate the reviewer for this insightful comment. We agree with the reviewer that this statement is not accurate. Therefore, we have removed the statement, and revised this paragraph as below,

“The above examples indicate that the predicted reactions from G²Retro rather than the ground truth could be still possible and synthetically useful. Therefore, a more comprehensive evaluation strategy is needed not to miss those possible and potentially novel synthesis reactions. Discussions regarding *in vitro* validation are available in the ‘Discussions’ section.”

Comment #1.19: “To analyze the diversity of G2Retro results we identified products such that their third or fifth predicted reactions are ground truth (...) it is likely that their top-3 or top-5 predicted reactions are also possible’. The unknown predicted reactions cannot be ‘likely possible’. Therefore, the diversity may also be illustrative of

'errors' of the model if these hypothesis are not useful."

Response: We thank the reviewer for this great comment. We believe the word "likely possible" is confusing so in this revision, we included a paragraph as below explaining what we mean by "likelihood" and why we think the predicted reactions ranked above the ground truth might be likely to occur.

"To analyze the diversity of G²Retro results, we analyzed the reaction centers among the top-predicted reactions. We identified a set of products such that their third or fifth predicted reactions are the ground truth, referred to as having a hit at 3 or 5, respectively. Please note each predicted reaction was scored using the sum of the log-likelihoods of all the predictions along the transformation paths from the product to its reactants (please refer to Section 'Inference'), and then ranked based on the score. Thus, the predicted reactions ranked above the ground truth have a higher likelihood than the ground truth. Given that G²Retro has demonstrated strong performance as in Table 2 in scoring and prioritizing the ground-truth reactions, we assume that its likelihood calculation is reliable and therefore, the reactions ranked above the ground truth might also be likely to occur."

We agree with the reviewer that without experimental proof, we cannot ensure the top predictions are not errors. Thus, in the revision, we have included the following discussion.

"Meanwhile, we acknowledge that the diverse, top predictions may still be errors and thus, more reliable web-lab experimental validation is needed as will be discussed later in the 'Discussions' Section."

Comment #1.20: "the clustering description is not clear. The authors seem to have clustered reactants from diverse products. In that case, I don't understand what conclusions can be made."

Response: We thank the reviewer for raising this concern. We want to clarify that we did not cluster reactants from diverse products. "We clustered the products according to their reaction similarity distributions". That is, the instances that we clustered are products, not reactants. To cluster products, for each product, we used the *distribution* of the pairwise reaction similarities of its top-10 predicted reactions as the product's "feature vector." The similarity between two reactions of the *same* product was calculated using Equation (1), which maximizes the pairwise reactant similarities from the two reactions.

To make this clear, we have revised the manuscript as follows. We also included the **Algorithm A8 in the Supplementary Information**.

"We also analyzed the reaction diversity by comparing the number of reaction centers in products with high reaction diversity and low reaction diversity. For each product, the diversity of its predicted reactions is represented by the distribution of all pairwise similarities of its predicted reactions, that is, lower reaction similarities indicate higher reaction diversity. For two predicted reactions of a product M_p , for example, reaction $R_1: M_1 + M_2 \rightarrow M_p$ and reaction $R_2: M_3 + M_4 \rightarrow M_p$, the similarity between R_1 and R_2 was calculated as follows,

$$\text{sim}(R_1, R_2) = \max(\text{sim}_m(M_1, M_3) + \text{sim}_m(M_2, M_4), \text{sim}_m(M_1, M_4) + \text{sim}_m(M_2, M_3)), \quad (1)$$

where $\text{sim}_m()$ is a similarity function over molecules, calculated using Tanimoto coefficient over 2,048-bit Morgan fingerprints of the molecules. We clustered the products according to their reaction similarity distributions using the K-means clustering algorithm in Euclidean distances. The clustering algorithm is presented in **Algorithm A8 in Supplementary Information Section S5**."

The conclusion we made is that products with low reaction diversity tend to have fewer predicted reaction centers. We revised the corresponding analysis as follows:

"Comparing Figure 7a and Figure 7b, HRD **products** tend to have more reaction centers **in their predicted reactions** than those in LRD **products**, and the number of reaction centers correlates well with reaction diversity (-0.8486 between the average **reaction** similarities and the number of reaction centers)."

Comment #1.21: "The discussion is a bit weak because the authors bring criticisms on practices that they do not help to solve themselves. For instance, they mention that experimental validation should be done, but they did not perform it themselves. However, I don't request modifications."

Response: We thank the reviewer for this comment. Although the reviewer did not request modifications, we still wanted to strengthen the discussion accordingly. In this revision, we have significantly revised the Discussion Section. Please refer to the "**Discussions**" Section for more detail.

In the Discussions Section, we first identified the critical issue on difficult comparison among existing methods, including unavailability of source code, irreproducibility of reported results, different training data and test data used by different methods, and comparison among different amounts of results. To the best of our knowledge, these issues have never been identified in the literature on retrosynthesis prediction, and we were the

first to identify and discuss such issues. We also believe these issues “require dedicated research, implementation, and regulatory effort from the **entire research community**, for example, by following the Open Science Policy from the European Union and the Data Sharing Policy from the United States National Institute of Health.” For example, each method needs to ensure using the same training, validation, and test benchmark data, and has their source code available. This requires the researchers of each method to follow the same data use principles, which is out of the scope of our manuscript, or out of the capability of our own research team. The Open Science Policy from the European Union (https://research-and-innovation.ec.europa.eu/strategy/strategy-2020-2024/our-digital-future/open-science_en) and the Data Sharing Policy from the United States National Institute of Health (<https://sharing.nih.gov/data-management-and-sharing-policy/data-management>) have provided regulatory policies to enforce good practices of open science and open data. Clearly, this is an issue of substantial scale, scope, importance, and impact that our team cannot solve ourselves. Still, we raised the concerns and suggested resolutions, which we believe it is helpful.

We also discussed the needs of *in vitro* validation and the challenges it faces to experimentally validate all the predicted results. We believe this is an important issue to discuss, and we wanted to raise the awareness from the research community and to inspire potential solutions to move the research forward. In our discussion, we have provided some suggestions in order to address this issue, including the development of prioritization approaches to identifying predicted reactions to validate, literature search to identify same or similar reactivity patterns, evaluation based on domain knowledge (e.g., synthetic chemistry knowledge), and in the end *in vitro* experimental validation. Please refer to the “**In vitro validation**” Section for more details. We acknowledged that in terms of scope (e.g., 50,070 predicted reactions), costs (e.g., on average > \$100 per reaction), and time (e.g., about 30 reactions per month per lab staff including chemical purchase, chromatography, analysis, and documentation), *in vitro* validation of all the predicted reactions in order to fully validate the computational methods is a problem that our research team cannot solve ourselves. It is also out of the scope of our manuscript. Still, we believe our discussion, the raised concerns, and suggested solutions are helpful for the research community to develop solutions to address the issues.

Comment #1.22: “For the above reasons, I believe that the contribution is better suited to a more specialized journal. However, if the edit wish so, it is needed to bring to this manuscript some major modifications.”

Response: We really appreciate all the comments and suggestions from the reviewer. We have made substantial, major modifications as the reviewer requested. As responded to **Comment #1.2** and per our early discussion with the editor, we believe our manuscript fits the scope and audience of *Communication Chemistry*.

Comments from Reviewer 2:

Comment #2.1: “G2Retro presents a method for one-step retrosynthesis prediction consisting of two modules: a reaction center selection module proposing multiple candidates as well as a synthon completion block outputting the final reactants. The method falls into the category of semi-template-based approaches. The method shows strong performance and compares against current state-of-the-art methods.

G2Retro follows previous methods by dividing the one-step retrosynthesis prediction into a two-step procedure. The method is solid, and well introduced in the supplementing Method section. The paper is well written. The literature and related work section are complete and the authors are well informed on current developments as well as state-of-the-art methods. The chosen datasets are recognized benchmarks and the chosen comparison to other methods is fair. In particular, the authors point out current flaws in existing pipelines (i.e., RetroPrime and G2G) which suffer from an existing bias in the dataset. While one could argue that the method could be tested in a more large-scale setting, these are the usually chosen settings and a well-suited test environment to evaluate the method’s performance.”

Response: We appreciate the reviewer for this positive comment.

Comment #2.2: “As the proposed method is a semi-template-based method, it is always a stretch to assume diverse reactants. It would be great if the authors could emphasize and reiterate why they assume G2Retro to increase the diversity of reactants over previous methods.”

Response:

We are very grateful for this comment. The reason why G2Retro could increase the diversity of reactants is that G2Retro uses beam search to predict multiple reaction centers for each product (e.g., at different bonds), which further leads to multiple different synthons; each of the synthons will be completed into a reactant, and thus the different synthons lead to different reactants. In this revision, we have emphasized and reiterated such reasons in different contexts as follows:

- “G2Retro employs a new, effective beam search strategy that prioritizes the most possible reactants and the corresponding completion actions along the synthon completion paths. The beam search also allows multiple different reaction centers, enabling diversity in the completed reactants.” (in “Introduction” Section)
- “According to its authors, GraphRetro tends to bias its beam search to the most possible reaction center. Thus, it may prioritize the most possible reactants from the most possible reaction center at the very top of its predictions. However, if the most possible reaction centers are not the ground truth, GraphRetro would totally miss the ground truth in its beam search, resulting in poor performance on other top accuracy metrics. In addition, such focused beam search limits the diversity of identified synthons, and thus the completed reactants.” (In “Comparison with GraphRetro and RetroPrime” Section)
- “This could be due to that these TF methods with SMILES representations may fail to generate diverse or even many valid reactants with beam search, leading to limited variation in their predicted results, and thus low and similar top-3, top-5 and top-10 accuracies. This lack of diversity and richness in the predictions, in addition to the lack of interpretability during the chemical sequence transformation process, could hinder the application of TF methods in retrosynthesis prediction. However, the prediction diversity and richness in G2Retro is enabled by the multiple possible reaction centers predicted by G2Retro and the corresponding completed reactants.” (In “Comparison with template-free (TF) methods” Section)
- “G2Retro has the mechanisms to facilitate diverse predictions: The beam search strategy in G2Retro allows multiple reaction centers and multiple different attachments, and therefore potentially different scaffolds and structures in the predicted reactants.” (In “Diversity on predicted reactions” Section)
- “Then G2Retro converts the product graph \mathcal{G}_p into the top- K synthon graphs $\{\mathcal{G}_{s,i}\}_{i=1}^K$ accordingly. Different reaction centers lead to diverse synthons.” (In “Top- K Reaction Center Selection” Section)
- “Since during synthon completion, the attachment substructure type prediction (Equation 18) gives a distribution of all possible attachment substructures; by using top possible substructures, each synthon and its intermediate graphs can be extended to multiple different intermediate graphs, leading to exponentially many reactant graphs and diversity in the predicted reactions.” (In “Top- N Reactant Graph Generation” Section)

Comment #2.3: “While investigating the top-3 or top-5 predicted reactants if the ground truth is not contained, the analysis conducted is not clear. In particular, Figure 4 (‘Product (%)’) is not clear to me. Could the authors

clarify and potentially improve the description of that section? It is not clear how the conclusions the authors make can be drawn from the conducted analysis.”

Response: We thank the reviewer for this comment. We have revised the caption of Figure 4 as “Percentage of products (Product (%)) with the different number of predicted reaction centers. a, Products with hits at 3. b, Products with hits at 5.” In Figure 4, each bar represents the percentage of products with hits at 3 or 5 (*y*-axis) which have the corresponding number of reaction centers (*x*-axis) among top-3 or top-5 predictions. The analysis was done by calculating, among all the hits-at-3 products, the percentage of products that have 1, 2 and 3 reaction centers among top-3 predicted reactions (Figure 4a), and among all the hits-at-5 products that have 1, 2, 3, 4 and 5 reaction centers among top-5 predicted reactions (Figure 4b).

We also revised the description of the entire “Diversity on predicted reactions” section. Please refer to our response to **Comment #1.18**, and also the revised Section “Diversity on predicted reactions” in the marked-up version for details. The conclusion we made is that the top predicted reactions were diverse, demonstrated by the different reaction centers they were derived from.

Comment #2.4: “Further, the purpose of the section on ‘In vitro validation’ is unclear to me. Why did the authors not attempt further analysis to overcome the limitations of the top-*k* accuracy analysis? Would it be possible for the authors, to conduct further analysis in this direction? If not, why?”

Response: We are grateful that the reviewer pointed this out. The purpose of the Section “*In vitro* validation” within the “Discussions” Section is to discuss the current issue with top-*k* accuracy as the metric to evaluate predicted reactions, and to point out the need for *in vitro* validation to “truly translate the computational approaches into real impacts.” We also wanted to point out the challenge of *in vitro* validation, and propose potential solution directions.

In this revision, we have extended the discussion on *in vitro* validation challenges as follows:

“The use of top-*k* accuracy as the evaluation metric has been dominating in the current retrosynthesis prediction research. However, as we have demonstrated in our case study, top-*k* accuracy has serious limitations. It only compares the predicted reactions with those in the benchmark data, but does not consider novel predicted reactions that are not in the benchmark data but might be also possible. Some existing work utilized forward synthesis prediction to predict the outcome product of a novel predicted reaction, and then compare the predicted product with the target product. However, such methods may not predict the products well if the novel or similar reactions are not included in the forward-synthesis model training, which is typically the case as forward-synthesis and retrosynthesis models usually use the same or similar training data. Thus, novel predicted reactions should be assessed using existing data from very large reaction databases and evaluated from the perspective of a synthetic chemist, so as to determine the accuracy and likelihood that these predicted approaches could be employed. Finally, the predicted reactions should be prioritized for synthesis and executed in the laboratory to determine whether or not they proceed as predicted. Such *in vitro* testing and validation are very much needed ultimately to truly translate the computational approaches into real impacts.”

However, as mentioned in the revision, it is extremely costly and time-consuming to evaluate and validate all the predicted reactions in a wet-lab setting. There have not been any studies on how to select predicted reactions to validate. Therefore, further analysis to overcome the current limitation is really out of the scope of our current study. Please also refer to our response to **Comment #1.21** for additional reason in terms of cost on additional data for evaluation and validation purposes.

Comment #2.5: “In the first part of the Experimental Results section, when comparing to previous methods, it is necessary to also cite each method (despite being done before). This improves reading. Similarly, the table is easier to pass if citations are added here as well.”

Response: We thank the reviewer for this great suggestion. In this revision, we have included the citations for previous methods in all the tables wherever these methods are mentioned. We have also included the citations in “Experimental Results” Section wherever they are mentioned.

Comment #2.6: “You state that neither NeuralSym, LV-Trans, Dual, and Retroformer published their code. G2G should be added to this list as well, otherwise, I do not understand why the authors did not rerun G2G on the debiased dataset.”

Response: We sincerely thank the reviewer for this comment and great suggestion. Actually, we did run G2G on our debiased dataset. However, we got very poor results (top-1: 31.9%, top-3: 52.1%, top-5: 59.0%, top-

10: 67.4% without the reaction type) using the reported hyperparameters in the G2G paper compared with their reported results (top-1: 48.9%, top-3: 67.6%, top-5: 72.5%, top-10: 75.5%). Therefore, it seems that their reported results cannot be reproduced by their source code and reported hyperparameters. We noticed that others had the same issue as reported in this link <https://github.com/DeepGraphLearning/torchdrug/issues/131> and only when they used new hyperparameters could they get results close to what was reported (this new update was on Sept 27, 2022, far after the G2G manuscript was published and after we submitted our initial version on July 7, 2022). In our manuscript, however, we still cite their reported results.

In this revision, we have included the following discussion in the Section “Comparison issues among existing methods”:

“In addition, there have been some reproducibility issues with G2G, as we also observed in our study.”

Comment #2.7: “While the authors conduct a very thorough comparison to existing work, it might be worth pointing out that previous methods as well have divided retrosynthesis prediction into two modules (reaction center identification and synthon completion), i.e., as done in 26 and 7.”

Response: We are grateful that the reviewer pointed it out. At the beginning of “Semi-template-based methods” in the “Related Work” Section, we pointed out that

“Instead, semi-template-based methods follow a two-step workflow utilizing atom-mappings: (1) **they first** identify the reaction centers and transform the product into synthons (intermediate molecules) using the reaction centers; **and then** (2) **they** complete the synthons into the reactants.”

In this revision, we have reiterated the same in “Comparison among template-based, template-free and semi-template-based methods” in the “Discussions” Section as follows:

“Semi-template-based methods, typically over molecular graphs, represent the most recent and also in general the best performing retrosynthesis prediction methods. **They utilize the powerful graph representation learning paradigm to better capture molecule structures.** They also take advantage of graph (variational) auto-encoder frameworks or sequential predictions to empower the models with generative ability. More importantly, semi-template-based methods can extrapolate to novel reactions using the latent representations learned from data, **and also have the mechanism to enable diversity among predicted reactions, by allowing multiple samplings from the latent space.** Meanwhile, semi-template-based methods **have two steps: (1) reaction center identification, and (2) synthon completion,** better complying with how chemical reactions are understood **and enabling certain interpretability of predicted reaction centers and derived reactants.**”

We have also reiterated the same in the “Introduction” Section as follows:

“**G²Retro imitates the reversed logic of synthetic reactions: it first predicts the reaction centers in the target molecules (denoted herein by ‘products’), identifies the fragments (denoted herein as ‘synthons’) needed to assemble the final products, and transforms these synthons into reactants. Therefore, G²Retro follows the semi-template-based frame.**”

“**G²Retro follows a semi-template-based framework, predicts reaction centers of different types in products first, and then transforms the resulting synthons into reactants by adding substructures to the synthons.**”

We have also ensured that semi-template-based methods are cited.

Comment #2.8: “The presentation of the method is clear. Standard datasets are chosen. The hyperparameters are published in the Supplement. Additional algorithms are provided to improve the general understanding of the method. Code is released. The presented method is sound and, after addressing the concerns raised above, I recommend acceptance.”

Response: We appreciate the reviewer for this positive comment. We have addressed all the comments of the reviewer in this revision.

Comments from Reviewer 3:

Comment #3.1: “Chen and colleagues present a machine learning approach to one-step retrosynthesis prediction, named G2Retro. The method builds upon previous semi-template-based methods, in particular G2G and GraphRetro, and uses a two-step approach in which, first, multiple reaction centers are identified to break the product molecule into synthons, and then a generative process selects substructures that complete the synthons to yield the reactants. The authors compare their approach to several other computer-aided retrosynthesis methods, showing competitive performance. Overall, the work appears technically sound and the manuscript is fairly well written.”

Response: We thank the reviewer for the positive comment.

Comment #3.2: “However, I felt that the novelty and impact of the work was often overstated, and one key method comparison appeared to be missing from the discussion. My main suggestion would be to revise the manuscript to better and more fairly contextualize the work and reduce speculation.”

Response: We sincerely appreciate the reviewer for all the constructive comments and suggestions. We have substantially improved our manuscript according to the reviewer’s suggestions. We have also included additional experiments and corresponding discussions. Please refer to our response to **Comment #3.3**, **Comment #3.4** and **Comment #3.5** below.

Comment #3.3: “This work mostly builds on top of G2Gs and GraphRetro, but the main novelty seems to be a more crafted definition of reaction centers which can cover a greater portion of test molecules. The author claimed the deep neural networks are ‘novel, customized’ (page 2) but the GMPN and FMPN seem to be simple versions of MPNNs (e.g., unlike RetroXpert which considers edge update, or GraphRetro which uses gated and directed updates). The reactant recovery stage makes use of a similar graph generation process as G2Gs, albeit with fragments. However, also the novelty of this aspect is not too clear (see next comment). The work might be publishable even if the technical novelty is limited, but what the novel aspects are should be stated clearly and without exaggeration.”

Response: We appreciate the reviewer for this comment. We agree with the reviewer that it is important to make our novel aspects clear. In this revision, we have revised the novelty statement in the “Introduction” Section. Please refer to the marked-up version of the revision.

In terms of the difference between G²Retro and G2G, we have revised the comparison in the “Semi-template-based methods” in the “Related Work” Section as follows:

“However, G²Retro is different from G2G. G²Retro can cover multiple types of reaction centers while G2G takes only the newly formed bonds as the reaction center, which leads to lower coverage of G2G on the dataset. During synthon completion, G²Retro attaches substructures (e.g., rings and bonds) instead of single atoms as in G2G, into synthons to simplify the completion process. In addition and more importantly, G²Retro uses other synthons of the same reaction and also the product to complete a synthon, and thus the synthon completion is more contextualized for the product, while G2G does not consider other synthons.”

Comment #3.4: “The contextualization of fragment-based molecular generation methods is missing. In particular, the synthon-to-reactants graph is formulated as sequential attachment of fragments starting from the synthons. This approach seems very similar to that of junction-tree VAEs. However, this and related generative methods are not mentioned in the text, which makes it hard to understand whether the approach used by the authors is per se novel or whether it is an adaptation/application of existing techniques.”

Response: We thank the reviewer for pointing this out. We agree with the reviewer that it is important to add the contextualization of fragment-based molecular generation methods. In this revision, we have included a new subsection “Fragment-based Molecule Generation” in the Section “Related Work” as follows, in which we compared G²Retro with JT-VAE in their generative process.

“Following the idea of fragment-based drug design, fragment-based molecule generation methods have been developed. For example, Jin et al first decomposed a molecular graph into a junction tree of chemical substructures, and then used a variational autoencoder over the junction trees and its chemical substructures to generate and assemble new molecules. Podda et al encoded and decoded a sequence of fragments via a variational autoencoder, and generated new molecules by connecting fragments generated from the autoencoder. Chen et al optimized a molecule by removing and attaching substructures in a starting molecule. G²Retro generates reactants from synthons also by attaching new substructures. However, the generation strategy in G²Retro is

fundamentally different from that in the previous fragment-based molecule generation methods. During synthon completion, G²Retro does not encode the synthons using their substructures as what JT-VAE and Modof do. It does not either encode or decode the substructures that are to be attached to the synthons. Instead, G²Retro attaches the substructures to a specific, identified atom in the molecular graph of the synthons. Therefore, G²Retro can directly attach a substructure to the predicted reaction centers.”

In addition, in response to **Comment #1.6**, we have moved the description of G²Retro-B, the extended G²Retro with fragments encoding, into the **Supplementary Information Section S1**, with additional details.

Comment #3.5: “The paper ‘Root-aligned SMILES: a tight representation for chemical reaction prediction’ previously published on arXiv reports a template-free method that seems to provide higher performance than G2Retro on USPTO-50K across the board with reaction type unknown. The results from this study should be included and discussed too, and the text adapted to reflect how G2Retro is not necessarily the top performing approach.”

Response: We sincerely thank the reviewer for pointing us to the paper. In this revision, we have included R-SMILES in the comparison. We noticed that R-SMILES augments its training and test data: for each product, it uses multiple non-canonical SMILES representations. Particularly during testing, it predicts reactions for each of the SMILES representations of each product, and then selects the best. Thus, R-SMILES explores a much larger reaction space and selects the best predictions out of a much larger pool of predicted reactions, compared to G²Retro. To enable selection from a pool of the size same as that of R-SMILES, we developed an ensemble of G²Retro models, referred to as G²Retro-ens, to test each molecule multiple times. We included a new Section “**Performance of Ensemble-based Methods**” to compare G²Retro-ens with baselines R-SMILES and AT that test each molecule multiple times. We also included a new **Section S3 in the Supplementary Information** to describe G²Retro-ens. Our experimental results demonstrate that G²Retro-ens achieves competitive or even better performance compared to R-SMILES.

Comment #3.6: “The fact the model proposes non-ground truth reactants with known transformation is claimed across the text as evidence that the approach can discover novel reactions. This claim seems unwarranted and highly speculative without any prospective validation, and as such I would remove it from the manuscript. Perhaps, the claim may be rephrased as to say that the method can propose novel synthetic routes for known compounds, which is different from discovering new reactions.”

Response: We are very grateful for this great suggestion. In this revision, as the reviewer suggested, we have revised the proposition of “discover new reactions” to “propose novel synthetic routes” across the entire manuscript as follows.

“Here we show that G²Retro is able to better predict the reactants for given products in the benchmark dataset than the state-of-the-art methods, and it can propose novel synthesis routes.”

“Case studies show that G²Retro could propose novel synthesis routes with high predicted likelihoods that are not included in the benchmark data (in ‘Case Study’ Section).”

“This is because high top- k ($k > 1$) accuracy implies that there might be a few reactions different from the ground truth but are very possible and thus are ranked on top. Such results may propose novel synthetic routes.”

“Diversity in predicted reactions is always desired, as it has the potential to propose novel synthetic routes.”

Comment #3.7: “Table 3 and comparison with GraphRetro: it appears that for synthon completion, GraphRetro still outperforms G2Retro. The reason for G2Retro’s overall better performance was mentioned as being that ‘GraphRetro does not do well in reaction center identification’. But if that’s the case, what’s the advantage of G2Retro’s synthon completion process over that of GraphRetro since it looks more complicated, and worse?”

Response: We thank the reviewer for raising this question. We want to point out that although the synthon completion module of GraphRetro performs better than that of G²Retro with respect to the ground-truth reaction centers, the overall performance of GraphRetro combining the reaction center prediction and synthon completion is still worse than that of G²Retro as in Table 2. This could be because the generative process of G²Retro in its synthon completion module allows it to better consider all possible completion actions and their likelihoods, whereas the classification process of GraphRetro in its synthon completion module may not generalize well, particularly when GraphRetro’s reaction center identification does not perform well with respect to the ground-truth reaction centers (i.e., in the top panel of Table 3), but its synthon completion module is trained using the ground-truth reaction centers (i.e., in the bottom panel of Table 3).

In this revision, we have included the following discussion in the “Comparison on synthon completion” Section

to clarify our analysis:

“In addition, the synthon completion module of GraphRetro may fail to accurately estimate the likelihoods of leaving groups, due to the ignorance of overall structures of predicted reactants. Such inaccurate likelihood estimation may aggravate the bias of beam search and reduce the diversity of predicted reactants as discussed in GraphRetro.”

“Although G²Retro does not outperform GraphRetro in the synthon completion module alone, its generative process allows G²Retro to consider all the intermediate molecular structures and more accurately estimate the likelihood of each completion action, conditioned on the reaction centers and the corresponding synthons from its reaction center identification module (i.e., not the ground-truth reaction centers), whereas GraphRetro may not generalize well, particularly given that GraphRetro’s reaction center identification does not perform well with respect to the ground-truth reaction centers (i.e., in the top panel of Table 3), but its synthon completion module is trained using the ground-truth reaction centers (i.e., in the bottom panel of Table 3).”

Comment #3.8: “As the author stated on page 3, MEGAN ‘does not follow the two-step workflow’, does not make use of the half-template, and is essentially template-free. It thus seems odd to categorize MEGAN as a semi-template method. Unless the authors have a strong argument for classifying MEGAN as semi-template-based, the text should probably be adjusted accordingly.”

Response: We really appreciate the reviewer for this great suggestion. We agree with the reviewer that MEGAN should be categorized as a template-free method. In the revision, we have changed the category of MEGAN (e.g., in “Related Work”) and updated the discussion on MEGAN (e.g., in “Comparison with template-free (TF) methods”) accordingly throughout the entire manuscript.

Comment #3.9: “Some statements do not seem to be entirely correct, in particular:

- On page 2, the authors discuss template-free methods and mention that ‘SMILES strings cannot fully leverage molecular structures, which ultimately determine molecule synthesizability and reaction types’. However, there are models that make use of graph information and the full molecular structures of the targets with their graph encoders (e.g., Retroformer, GTA, GET, Graph2SMILES). It might be better to be a bit more nuanced here.
- At page 12, the author mentioned that semi-template-based methods ‘utilize the most advanced graph representation learning paradigm’, however, it is unclear in what way these are the most advanced. None of these works made use of the most expressive graph encoder (e.g., GIN), nor did they use pretraining/contrastive learning techniques that are standard for representation learning (e.g., GROVER, DMP). Therefore, if no additional clarification is provided, it would be better to avoid this statement.”

Response: We sincerely thank the reviewer for the provided materials and for the great suggestions. For the first statement, we agreed with the reviewer that we missed the methods utilizing both SMILES and molecular graph representations. In this revision, we have included additional discussion as follows:

“To mitigate this issue, some template-free methods either enrich the product SMILES representation with molecular graph information or decode reactant SMILES strings from product molecular graphs, which, however, require additionally learning of the mapping from molecular graphs to SMILES and thus increase the learning complexity.”

For the second statement, we agree with the reviewer that it was overstated. Therefore, we have revised the statement as follows:

“They utilize the powerful graph representation learning paradigm to better capture molecule structures.”

We also checked other statements to make sure they are accurate.

Comment #3.10: “I’d suggest removing claims about how the approach would speed up drug discovery....so unless these claims are backed up with some data/reference, it seems to me they are unjustified.”

Response: We appreciate the reviewer for this suggestion. In this revision, we have removed this claim and revised the “Introduction” Section substantially.

Comment #3.11: “‘In general, template-based methods underperform template-free and semi-template-based methods.’ Is this statement correct? Table 2 would suggest that LocalRetro is in fact very competitive.”

Response: We are really grateful for the careful checking. We acknowledge that this statement is not accurate. In this revision, we have removed this statement.

Comment #3.12: “In the Conclusions: ‘G2Retro also enables diverse predictions and novel reactions that were accessed as very possible by synthetic chemists’. In addition to the issue of claiming the prediction of novel reactions, it is also unclear who the synthetic chemists were, how many, and how the study was designed. It seems like the reactions were simply analyzed by the authors. As such, I would remove this claim as it makes it sound as if an external panel of chemists evaluated the reactions in some rigorous fashion.”

Response: We are grateful that the reviewer pointed this important aspect out. We totally agree with the reviewer that it is important to make it clear that we did not ask an external panel of chemists to evaluate the predicted reactions in a rigorous fashion. Therefore, we have removed this statement in the revision.

Comment #3.13: “Table 2: G2Retro-B does not seem to be introduced when presenting the method but is encountered only in the Results. It might be good to anticipate how it is different from G2Retro before its results are presented in Table 2 and discussed at page 6.”

Response: We appreciate the reviewer for this careful review. In this revision, we have added the following description in the “Baselines” Section to introduce G²Retro-B before the Section Results:

“Inspired by the recent success of using fragments in other tasks, we further extended G²Retro into G²Retro-B by incorporating the fragments generated from the breaking retrosynthetically interesting chemical substructures (BRICS) fragmentation algorithm. Details of G²Retro-B are available in Supplementary Information Section S1. The experimental setting for G²Retro-B is identical to that of G²Retro.”

Please also refer to our response to **Comment #1.6** regarding more details on G²Retro-B in the Supplementary Information Section S1.

Comment #3.14: “My understanding is that the performance comparison to other methods (Table 2) is based on what is reported in the literature, as opposed to being based on re-runs of all the experiments with the various methods. It might be good to make this very explicit at some point in the text.”

Response: We thank the reviewer for this great comment. In this revision, we have revised the following paragraph to improve clarity:

“Note that the top-*k* accuracies of all the baseline methods are the reported results in their original papers (issues related to the comparison among methods are discussed later).”

We also included the following description in the footnote of table to make this more explicit:

“All the baseline results are reported in their original papers, where ‘-’ represents that the corresponding results are not reported.”

Comment #3.15: “On page 1, ‘auto-encoders’ are mentioned but the associated ref. 10 is about ‘variational autoencoders’, which are distinct from general auto-encoders. It might be good to specify whether the authors are referring to variational auto-encoders specifically.”

Response: We really appreciate the reviewer for this very careful review. We acknowledge that reference 10 (corresponding to reference 35 in the revision) should be for variational auto-encoder, not autoencoder. We have revised that as follows.

“These deep-learning methods learn from string-based representations (SMILES) or graph representations of given molecules, and generate possible reactant structures that can be used to synthesize these molecules, leveraging the advancement of natural language processing, graph neural networks, **variational auto-encoders** and other techniques in deep learning.”

We have also carefully checked our references to make sure they are accurate.

Comment #3.16: “Contextualization of Retroformer: The way it’s described on page 3 (reaction center detection followed by decoding) makes it sound like a semi-template-based method. Some elaboration is needed to explain why it’s categorized as template-free.”

Response: We sincerely appreciate the reviewer for this great suggestion. In this revision, we have revised the description of Retroformer in “Template-free baseline methods” to explain why Retroformer is categorized as template free:

“Retroformer predicts the reaction center region using a reaction center detection module, and uses the embedding of predicted centers as a condition to transform via Transformer the product into the reactants in SMILES. Although Retroformer predicts the reaction center, it does not split products into synthons using the reaction center, and thus does not follow a two-step, semi-template-based framework.”

Comment #3.17: “The reference for BART on Page 12 is missing.”

Response: We thank the reviewer for the careful check. In this revision, we have included the reference 56 for BART below on Page 12.

“Lewis, M. *et al.* BART: Denoising sequence-to-sequence pre-training for natural language generation, translation, and comprehension. In *Proceedings of the 58th Annual Meeting of the Association for Computational Linguistics (Association for Computational Linguistics, 2020)*.”

Comment #3.18: “Figure 3: the header of panel ‘e’ is on top of the molecular structure of panel ‘a’.”

Response: We thank the reviewer for the careful check. In this revision, we have rearranged Figure 3a and Figure 3e to address this issue. We also checked other figures to make sure there are not overlapped or have format issues.

Comment #3.19: “‘Discussion and conclusions’ should probably be renamed just ‘Discussion’ since there is a section called ‘Conclusion’ later on.”

Response: We appreciate the reviewer for this great suggestion. In this revision, we have renamed the Section “Discussion and conclusions” to “**Discussions**”.

Comment #3.20: “The header ‘Materials’ might not be too apt for a computational study, and could perhaps be called ‘Methods’ (with then ‘Methods’ becoming ‘Extended Methods’), or something else that does not imply wet-lab experiments, like ‘Dataset and baselines’?”

Response: We really appreciate the reviewer for this comment. We have renamed the Section “Materials” to “**Dataset and Baselines**”.

Comment #3.21: “At page 11, but also earlier in the text, it is discussed how the fact that the test set used by various methods are not the exactly the same results in ‘unfair’ comparison. ‘Unfair’ might not be the most appropriate word, as it implies some specific methods are at a disadvantage. It might be better to replace it with something else that implies it is hard to compare methods tested in slightly different ways, but not necessarily ‘unfair’.”

Response: We sincerely thank the reviewer for this comment. We agree with the reviewer that “unfair” is not the most appropriate word. Therefore, we have revised it as follows.

“In our study of the baseline methods, several issues were identified among existing methods **that make comparison across different methods hard**.”

“Even though all the methods adopted the same ratio (i.e., 80%/10%/10% for training/validation/test set) to split the benchmark dataset, their splits, particularly their test sets, are not identical, **making it hard to compare** these methods.”

We modified the title of the section that discussed the comparison issues as follows,

“**Comparison issues** among existing methods”

Reviewers' comments:

Reviewer #1 (Remarks to the Author):

Comment #1.2. The perception of novelty in a contribution is something that is very subjective. However, the manuscript claims to present a "new deep-learning-based generative model for one-step retrosynthesis prediction" which is in fact a "a new semi-template-based method for one-step retrosynthesis prediction". The formulation of the later is more accurate than the one in the manuscript.

However, I appreciate the modifications to put in perspective one-step retrosynthesis and more specifically their contribution. The authors attempted to improve the manuscript to better frame the manuscript. Yet, there are still an inconsistencies: "G2Retro could propose novel synthesis routes with high predicted likelihoods". The truth is there are no evidence in the manuscript that these routes are reasonable: no novel chemistry has been demonstrated here.

Also, the authors claim: "G2Retro develops a new fragment-based generation strategy". It should be "new" compared to something. Otherwise, this is only a position statement. I have the same remark with the claim of having implemented a "new, effective beam search strategy". Actually, the situation is correctly described on page 4: "However, G2Retro is different (...)".

Therefore, these previously mentioned problems are not completely solved.

The authors confirm in Comment #1.3 that they provide no new data in any form and no new understanding of existing data. I acknowledge that they provide the source code, but data are just duplicated. The source code itself is a Python code requiring expertise to be downloaded, installed and used. The code has incomplete specifications, limited documentation and no test case. This definitely limits the audience of such material.

For the comment #1.4, unfortunately I cannot accept the modification: "To the best of our knowledge, there have not been any studies in which synthetic chemists, based on their chemistry knowledge, evaluate all the 50,070 predicted reactions for the 5,007 products in the benchmark dataset used by the current computational methods, or any in vitro experiments in a web-lab setting that validate all of them. An in-house estimation we conducted shows that to in vitro validate one reaction, it takes on average more than \$100 cost on chemical reagents, solvents, chromatography supplies and lab supplies, excluding costs on lab staff. It also takes a lab staff at least 125 years to finish the experiments, all the related analyses and documentation. Therefore, it is extremely costly and time-consuming to conduct in vitro validation for all the predicted reactions." This is just a provocation. The authors are expected to identify interesting cases that could be supported bibliographically or experimentally for a reasonable costs. They even proposed such cases in figure 2 and 3.

The general advertising tone of the manuscript has been attenuated in response to the comment #1.5.

The modifications concerning G2Retro-B in comment #1.6 have been made, improving the readability of the manuscript without sacrificing important information.

The authors did not correct the manuscript following their answer in the comment #1.7. The current formulation in the conclusion is insufficient because the authors consider it is covered by a general remark about operational limitations of G2Retro and other tools.

The authors accepted some corrections in the comment #1.8, #1.9 and #1.10 that are indeed useful.

The authors partially accepted some correction in #1.11. Their answer is wrong and possibly problematic. It is wrong because SciFinder is not an editor and does not own reaction data. They own CAS and SciFinder database and they sell a service. I suppose that the SciFinder representative named by the authors have been informed of their letter and gave his consent, this discussion being public. Yet, the SciFinder representative cannot be blamed by the authors for not having validated their results.

For the comment #1.13, I cannot accept the correction "Please note that while there is always one ground-truth reaction for each product in the benchmark data, there may exist actually numerous feasible reactions for each product that are not included in the benchmark data. Therefore, reactants that are considered incorrect based on the benchmark data might still be plausible and included in other larger databases." If the reactants are not the defined ground truth, from the data mining point of view they are erroneous predictions. I can accept that the authors term them as hypothetical, requiring a validation: a proposed reaction can be searched in another source (such as a publication), or it can be experimentally challenged. I insist that hypothetical reactions cannot be included in "other large databases".

The answer to the comment #1.14 cannot be accepted neither because of poor formulation. The correction propose "Compared to these TF methods, though without the full coverage on the test set, G2Retro and G2Retro-B better imitate the reversed logic of synthetic reactions with two steps: reaction center identification and synthon completion." The problem here is that G2Retro follows its own "reversed logic of synthetic reactions" and TF has no such thing to "imitate".

In the comment #1.19 the correction "more reliable web-lab experimental validation" must be corrected by "wet-lab".

For comment #1.20, the situation is insufficiently improved after the correction. The authors are now precise about what is the measure presumably used for clustering. The measure maybe meaningful for reactions that have the same product. But the authors state that they "clustered the products according to their reaction similarity". Here, there is no description of reaction similarity for reactions in general, including those that do not share the same product. The Figure 6 is also improperly documented and labeled: the x- and y- are not described in the caption, the color scale is not described, the acronyms LRD and HRD are not developed in the caption.

Considering the modifications in the discussion mentioned in comment #1.21, they cannot be accepted for two reasons:

- About forward synthesis, the authors state that "such methods may not predict the products well if the (...) reactions are not included in the (...) training set". This is a personal opinion unsupported by evidences or bibliography. It must be presented as such if not better grounded.
- "To the best of our knowledge, there have not been any studies (...) [evaluating] all the 50,070 predicted reactions (...). Therefore, it is extremely costly and time-consuming to conduct in vitro validation for all the predicted reactions." As mentioned before, in my opinion, this is unacceptable. The authors did not validated their results and made no efforts to do so.

About comment #2.1, the authors answered the GraphRetro "may prioritize the most possible reactants from the most possible reaction center at the very top of its predictions. However, if the most possible reaction centers are not the ground truth, GraphRetro would totally miss (...)". In the manuscript, I read "Instead, G2Retro applies a greedy beam search strategy (...) to only explore the most possible top reactant graph completion paths." Which seems very similar to me: G2Retro may suffer the same diseases. However, the authors globally conformed to the request of the reviewer.

The correction in comment #2.3 is insufficient. The authors accurately state "We identified a set of products such that their third or fifth predicted reactions are the ground truth, referred to as having a

hit at 3 or 5, respectively." However the caption of Figure 4 mention the term "hits" without any precision. The situation did not improve.

The answer of the authors to the comment #2.4 is not sufficient. The section is termed "In vitro validation" and there are no in vitro validation. The proposed modification is off-topic and are not scientifically grounded.

The modification listed in comment #2.5 and #2.6 are satisfactory.

The comment #2.7 gave clear directions about how to cite 26 and 7. This was not accepted by the authors.

Reviewer #3 (Remarks to the Author):

The authors substantially revised the manuscript and addressed all the major concerns raised. The clarity of the text is improved and the work better contextualized with respect to other retrosynthetic methods. What I felt were overstatements have also been removed or adapted accordingly. Therefore, my opinion is that the manuscript is now suitable for publication.

Summary of revisions:

We sincerely thank again the reviewers for their constructive comments. We believe these comments are greatly helpful for us to improve the quality of our manuscript further. We have incorporated the comments in the revised manuscript.

In this document, we are providing point-to-point responses to all of the reviewers' comments, and providing our corresponding revised materials in the revised manuscript: the revised materials will be quoted and highlighted in red; otherwise, only the Section titles will be quoted and highlighted, and please refer to the revised main text and Supplementary Information for details.

Together with this document, we are submitting a marked manuscript in which all the revisions have been highlighted in red. In the revised submission, we have made changes and improvements, summarized as follows (sections, figures and tables with references starting with 'S' indicate those in the Supplementary Information):

- (1) We extended the "Case Study" section in the main text, and included more discussions on the predicted reactants for two newly approved drugs. The analyses demonstrated that G²Retro could predict diverse and reasonable synthesis routes for these drugs.
- (2) We added a new section Section S4 in the Supplementary Information, in which we discussed the predicted reactants for the other two newly approved drugs.
- (3) We extended the "Performance on Different Reaction Types" section in the main text. We included more discussions on the limited coverage of G²Retro on reaction types.
- (4) We removed the "In vitro validation" section from the main text, as it does not contain any *in vitro* experimental results, and the discussion within this section may not be accurate.

Comments from Reviewer 1:

Comment #1.1: “(Comment #1.2). The perception of novelty in a contribution is something that is very subjective. However, the manuscript claims to present a ‘new deep-learning-based generative model for one-step retrosynthesis prediction’ which is in fact a ‘a new semi-template-based method for one-step retrosynthesis prediction’. The formulation of the later is more accurate than the one in the manuscript.”

Response: We thank the reviewer for this comment. We agree with the reviewer that it is important to accurately clarify our claims. In this revision, we have reformulated our claim below as the reviewer suggested.

“We developed a new **semi-template-based method via deep learning** for one-step retrosynthesis prediction, denoted as G²Retro.”

Comment #1.2: “(Comment #1.2). However, I appreciate the modifications to put in perspective one-step retrosynthesis and more specifically their contribution. The authors attempted to improve the manuscript to better frame the manuscript. Yet, there are still an inconsistencies: ‘G²Retro could propose novel synthesis routes with high predicted likelihoods’. The truth is there are no evidence in the manuscript that these routes are reasonable: no novel chemistry has been demonstrated here.”

Response: We sincerely thank the reviewer for the positive feedback on our modifications. In this revision, we have removed the statements regarding the “novel synthesis routes” from the abstract and Introduction. However, we would like to clarify that according to the analyses, most of the synthesis routes in the “Case Study” Section are reasonable with bibliographical support; the predicted synthesis routes can be diverse. Please refer to our response to **Comment #1.5** later. Therefore, we reformulated our claim for the case studies as below in the Introduction,

“Case studies show that G²Retro could propose **diverse and reasonable** synthesis routes with high predicted likelihoods that are not included in the benchmark data (in ‘Case Study’ Section).”

We also reformulated our claims regarding novel synthesis routes in the main text as follows,

“High top-*k* accuracies with *k* > 1 may signify plausible reactions **not included in the dataset**, as will be examined later in Section ‘Case Study’.”

“**Such results may enable the exploration of multiple synthesis routes and may be of synthetic value if specific coupling methods fail or if specific starting materials are unavailable.**”

“Diversity in predicted reactions is always desired, as it has the potential to **enable the exploration of multiple synthesis routes.**”

Comment #1.3: “(Comment #1.2). Also, the authors claim: ‘G²Retro develops a new fragment-based generation strategy’. It should be ‘new’ compared to something. Otherwise, this is only a position statement. I have the same remark with the claim of having implemented a ‘new, effective beam search strategy’. Actually, the situation is correctly described on page 4: ‘However, G²Retro is different (...)’.

Therefore, these previously mentioned problems are not completely solved.”

Response: We appreciate the reviewer’s comment on rigor in wording. Following the comment, in this revision, we reformulated our claims as below, with references:

“G²Retro develops a new fragment-based generation strategy **compared to the previous semi-template-based methods**,^{27–30} to complete synthons into reactants by sequentially attaching substructures (i.e., bonds and rings) starting from the predicted reaction centers (in ‘Synthon Completion’ Section).”

“G²Retro employs a new, effective beam search strategy **compared to the previous semi-template-based methods**,^{27–30} that prioritizes the most possible reactants and the corresponding completion actions along the synthon completion paths. The beam search also allows multiple different reaction centers, enabling diversity in the completed reactants.”

Comment #1.4: “The authors confirm in Comment #1.3 that they provide no new data in any form and no new understanding of existing data. I acknowledge that they provide the source code, but data are just duplicated. The source code itself is a Python code requiring expertise to be downloaded, installed and used. The code has incomplete specifications, limited documentation and no test case. This definitely limits the audience of such material.”

Response: We appreciate the reviewer for this comment.

New data: We wanted to clarify that in our previous response to Comment #1.3, we confirmed that **we did provide new data**. Based on the multiple definitions of “scientific data” in the United States and in Europe,

"our source code, the data, and the hyper-parameters for the model and result reproducibility. [...] together with the analysis results and discussions in this manuscript, should be considered as 'scientific data.' ". "Since the scientific data produced in our study have not been produced by anyone else in the literature, they should be considered as 'new data.' So we believe **our manuscript brings new data.**" We attached our response to the previous Comment #1.3 for your reference.

Particularly, we believe that all of our published source code, experimental results from the G²Retro model, predicted reactants and their analyses, and case studies should be considered as **new data**. Our manuscript also brings a new computational method to predict reactions better than the existing 21 methods, which should be considered as new data. In addition, G²Retro model could provide a new understanding of existing data, as it can predict synthesis routes that do not exist in the training data, by learning from existing data.

Python code: We want to clarify that using Python code for developing tools to solve computational chemistry problems has been very popular. For example, all the 21 baseline methods in our manuscript provided Python source code that requires to be downloaded, installed, and used (e.g., GraphRetro provided Python code in <https://github.com/vsomnath/graphretro>). Python also has been extensively used in various other chemistry problems, such as protein-ligand docking (e.g., <https://github.com/HannesStark/EquiBind>) and molecule property prediction (e.g., <https://github.com/chemprop/chemprop>). Below we provided a short list of recent work published by Nature, which used python to address various chemical problems:

- (1) Grünewald, F., Alessandri, R., Kroon, P.C. et al. Polyply; a python suite for facilitating simulations of macromolecules and nanomaterials. *Nat Commun* **13**, 68 (2022).
- (2) Uchino, H., Tsugawa, H., Takahashi, H. et al. Computational mass spectrometry accelerates C=C position-resolved untargeted lipidomics using oxygen attachment dissociation. *Commun Chem* **5**, 162 (2022)
- (3) Sato, H., Ishikawa, A., Saito, H. et al. Critical impacts of interfacial water on C–H activation in photocatalytic methane conversion. *Commun Chem* **6**, 8 (2023).
- (4) Bieniek, M.K., Cree, B., Pirie, R. et al. An open-source molecular builder and free energy preparation workflow. *Commun Chem* **5**, 136 (2022).

More information can be found here: <https://www.nature.com/search?q=python&order=relevance&journal=commschem> for manuscripts published in Communications Chemistry and here <https://www.nature.com/search?q=python&order=relevance> for manuscripts published in Nature journals. Particularly, in Nature publications, 17,728 publications use Python as their programming language or mention Python in their studies. As a matter of fact, Python has been one of the most popular programming languages for scientific computing and the most popular programming language in general. Please find several references below:

- (1) Oliphant, T. E. (2007). Python for scientific computing. *Computing in Science & Engineering*, **9**(3), 10-20.
- (2) Python is becoming the world's most popular coding language ¹

Using github is extremely common for source code sharing. Please refer to <https://en.wikipedia.org/wiki/GitHub> for more information. As it states, "as of January 2023, GitHub reported having over 100 million developers and more than 372 million repositories, including at least 28 million public repositories. It is the largest source code host as of November 2021." All the baseline methods studied in our manuscript shared their Python code in github. The documentation in our Python code repository follows the standard templates commonly used by previous Python-based tools for chemistry problems, and follows the common practice in open source code sharing. Therefore, given the wide use of Python and github, we believe that providing Python source code would not substantially limit the audience of our work.

We recognize the importance of providing more detailed documentation and test cases to ensure that our model is easily accessible to a wide audience. Therefore, following the reviewer's suggestions, we have improved our code repository as outlined below.

- (1) We have included detailed instructions on how to set up a Python conda environment for G²Retro model, to ensure that even non-experts can easily use G²Retro model.
- (2) We have provided example commands for users to conveniently train and test G²Retro model using either the provided benchmark dataset or other new datasets.
- (3) We have included well-trained G²Retro models and sample test cases, allowing users to test G²Retro model

¹https://www.economist.com/graphic-detail/2018/07/26/python-is-becoming-the-worlds-most-popular-coding-language?utm_medium=cpc.adword.pd&utm_source=google&ppccampaignID=17210591673&ppcadID=&utm_campaign=a.22brand_pmax&utm_content=conversion.direct-response.anonymous&gclid=EATaIQobChMIxbDx30Th_QIVOTtBh2DFw9iEAMYASAAEgKCRfD_BwE&gclidsrc=aw.ds

on custom examples without needing to train a new G²Retro model.

Comment #1.5: “For the comment #1.4, unfortunately I cannot accept the modification: ‘To the best of our knowledge, there have not been any studies in which synthetic chemists, based on their chemistry knowledge, evaluate all the 50,070 predicted reactions for the 5,007 products in the benchmark dataset used by the current computational methods, or any in vitro experiments in a web-lab setting that validate all of them. An in-house estimation we conducted shows that to in vitro validate one reaction, it takes on average more than \$100 cost on chemical reagents, solvents, chromatography supplies and lab supplies, excluding costs on lab staff. It also takes a lab staff at least 125 years to finish the experiments, all the related analyses and documentation. Therefore, it is extremely costly and time-consuming to conduct in vitro validation for all the predicted reactions.’ This is just a provocation. The authors are expected to identify interesting cases that could be supported bibliographically or experimentally for a reasonable costs. They even proposed such cases in figure 2 and 3.”

Response: We appreciate the reviewer’s comment. First of all, we want to apologize for any offense our statement may have caused to the reviewer. We never intended to offend the reviewer or anyone. Our discussion was never meant to be provocative to the reviewer or anyone. We extremely value the comments from the reviewer and have been very carefully addressing all the comments. In this revision, we have removed the Section “In Vitro Validation” with the above discussion.

We agree with the reviewer that it is crucial to provide more interesting cases that could be supported bibliographically. To address this, we have selected four recently approved drugs for further analysis, including “mitapivat”, “tapinorf”, “mavacamten” and “oteseconazole”². In the revised main manuscript, we have presented the predicted reactants by G²Retro model for “oteseconazole” and “tapinorf” in Figure 2 and 3. Furthermore, we have extended the discussion on the predicted reactants in the Section “Case Study” in the main manuscript as below.

“G²Retro can predict multiple reactions for each product due to multiple predicted reaction centers. This variability could be useful for chemical synthesis in order to consider all possible reaction strategies. In order to illustrate the predictive power of G²Retro, we have highlighted the top-10 predicted reactants by G²Retro with reaction types unknown for four newly approved drug molecules in 2022, including Mitapivat, Tapinorf, Mavacamten, and Oteseconazole. Among them, the predicted reactants for Mitapivat and Tapinorf are presented in Figure 2 and 3 which will be discussed later; the results and the discussions for Mavacamten and Oteseconazole are available in Supplementary Information Section S4. Note that these drugs are not included in our training, validation, or testing data. Therefore, how G²Retro works on these drugs truly indicates its predictive power for new molecules.

Mitapivat as in Figure 2a is a drug approved for hereditary hemolytic anemias in 2022. The synthetic route within the patent reporting the discovery of Mitapivat utilizes an amide coupling reaction to form the C:2-N:23 bond (Figure 2b). This is correctly predicted by G²Retro as the top-1 reaction (Figure 2c). As indicated by the top-5 reaction (Figure 2g), G²Retro also predicts that the amide coupling reaction could be performed with the carboxylate salt of one of the reactants, a useful reactant under the right pH conditions. G²Retro also predicts that the acyl chloride as the substrate in this transformation would also react with the amine group and produce the desired molecule (Figure 2j). In addition, G²Retro identifies the N:7-S:8 bond of sulfonamide linkage as the reaction center (e.g., Figure 2d, 2e, 2f, 2k, 2l). Most impressively, G²Retro predicts various S:8 sulfonyl groups reacting with the N:7 amine group, such as sulfonyl chloride (Figure 2d), sulfonyl fluoride (Figure 2e) and sulfonic acid (Figure 2f), which are theoretically feasible for the formation of the N:7-S:8 bond. G²Retro also predicts that the N:26-C:27 bond could be the reaction center and formed by the N:26 amine group reacting through a reductive amination with ketone in Figure 2h or through a nucleophilic substitution with the chloride in Figure 2i.

Tapinarof as in Figure 3a is a drug approved for plaque psoriasis and atopic dermatitis. The reported synthesis in patent constructs this drug by removing the protecting groups on O:5 and O:10 (Figure 3b). G²Retro correctly predicts the deprotection of the methyl groups on O:5 (Figure 3c) or O:10 (Figure 3d), which would work to produce the desired molecule, although the ground truth failed to be predicted due to the limitation of reaction centers. Similarly, G²Retro generates possible reactants that contain different types of protected alcohols, as seen with the methoxymethyl groups on O:5 and O:10 in Figure 3f and Figure 3i and the benzyl-protected O:5 in Figure 3j. Most impressively, G²Retro also identifies the alkene linkage between C:11 and C:12 (Figure 3e and 3l) and the C-C bond between C:7 and C:11 (Figure 3g, 3h, and 3k) as reaction centers with various coupling

²<https://www.fda.gov/drugs/new-drugs-fda-cders-new-molecular-entities-and-new-therapeutic-biological-products/novel-drug-approvals-2022>

reactions. These coupling reactions include McMurry coupling (Figure 3e), Wittig coupling (Figure 3l) and Suzuki coupling (Figure 3g and 3h). ”

In the supplementary information, we included the predicted reactants for “mavacamten” and “oteseconazole” in Figure S1 and S2, respectively. Additionally, we have added a new Supplementary Section S4 to discuss these predicted reactants as below.

“For Mavacamten as in Figure S1a, which was approved by FDA in 2022 to treat hypertrophic cardiomyopathy, the patent literature reports the utilization of a nucleophilic aromatic substitution for the formation of the C:8-N:9 bond (ground truth in Figure S1b). G²Retro correctly predicts this coupling as the top-1 reaction (Figure S1c), and also identifies its additional permutations by replacing the aryl chloride with the aryl bromide and the aryl fluoride, respectively (Figure S1d and S1g). Aryl fluorides in Figure S1g are not as typical as aryl chlorides and bromides, and G²Retro ranks the substitution reaction involving the aryl fluoride low. In addition to the amine coupling strategy with aryl halides, G²Retro also identifies the reaction of the amine with trifluoro methyl sulfate to make the same bond (Figure S1h), which would be expected to work as with aryl halides in Figure S1c and S1d. However, the alcohol in Figure S1i is not a good enough leaving group to make the bond (i.e., C:8-N:9). Interestingly, G²Retro also identifies other amine linkages (e.g., in Figure S1e between C:2 and N:4; in Figure S1j between N:9 and C:10) as potential reaction centers. However, the proposed synthesis in Figure S1e and S1f would most likely lead to the formation of undesired products as these reactant pairs would likely result in the alkylation of both N:4 and N:9. Therefore, the use of the aryl halides in Figure S1c and S1d would be the more efficient way of obtaining the desired product.

Oteseconazole as in Figure S2a is a drug approved for recurrent vulvovaginal candidiasis⁷. In the patent literature⁸, this drug is constructed by the C-C bond forming reaction between C:6 and C:7 and is assembled with Suzuki coupling¹³ between an aryl bromide group and a boronic ester (Figure S2b). G²Retro correctly predicts this coupling as the top-1 with the boronic acid (Figure S2c), the top-3 which is the same as the patented reaction (Figure S2e), and the top-9 reaction with a relatively uncommon boronic ester (Figure S2k). Boronic acids in Figure S2c would typically be considered by synthetic chemists as interchangeable with boronic esters, and thus should be considered a feasible reaction; while Boronic ester in Figure S2k should react in the same way with the patented reaction, and thus could deliver the desired compound. Interestingly, G²Retro also predicts the Ullmann-type coupling⁵⁹ with different aryl halides to construct the C:6-C:7 bond in Figure S2d, S2f and S2g, all of which would be expected as feasible reactions. Although the reaction center is correctly identified in Figure S2i, the proposed coupling of two boronic acids would not be effective. G²Retro also identifies another C-N coupling of various aryl halides with imidazoles (C:15-N:16 - Figure S2h and S2j), which hypothetically would also work as expected.”

Comment #1.6: “The general advertising tone of the manuscript has been attenuated in response to the comment #1.5.”

Response: We thank the reviewer for this positive comment.

Comment #1.7: “The modifications concerning G2Retro-B in comment #1.6 have been made, improving the readability of the manuscript without sacrificing important information.”

Response: We appreciate the reviewer for this positive comment.

Comment #1.8: “The authors did not correct the manuscript following their answer in the comment #1.7. The current formulation in the conclusion is insufficient because the authors consider it is covered by a general remark about operational limitations of G2Retro and other tools.”

Response: We appreciate the reviewer for this comment. We acknowledge the importance of extending the discussion on the limited coverage of G²Retro on reaction types. In this revision, we have added the following discussion on limited coverage in the Section “Performance on Different Reaction Types”:

“Please note that G²Retro is designed to predict reactions that involve three types of reaction centers: 1) a single newly formed bond with induced changes in bond types; 2) a single changed bond; 3) a single atom with a fragment removed. As a result, G²Retro could not fully cover reaction types such as rearrangement, isomerization, cyclization and click reactions, which involve multiple changes in bond formation or atom detachment. This illustrates G²Retro’s limitation in handling all possible reaction types. It is worth noting that other semi-template-based methods such as G2G, GraphRetro, also share this limitation. Therefore, developing an effective semi-template-based method that overcomes this limitation could be an interesting future research direction.”

Comment #1.9: “The authors accepted some corrections in the comment #1.8, #1.9 and #1.10 that are indeed useful.”

Response: We thank the reviewer for this positive comment.

Comment #1.10: “The authors partially accepted some correction in #1.11. Their answer is wrong and possibly problematic. It is wrong because SciFinder is not an editor and does not own reaction data. They own CAS and SciFinder database and they sell a service. I suppose that the SciFinder representative named by the authors have been informed of their letter and gave his consent, this discussion being public. Yet, the SciFinder representative cannot be blamed by the authors for not having validated their results.”

Response: We appreciate the reviewer for the clarification regarding SciFinder’s ownership of reaction data. We understand that SciFinder is a service provider that sells access to their CAS and SciFinder Databases. Regarding our previous statement about the SciFinder representative not validating our results, we apologize for any confusion that may have caused. We did not mean to imply that the SciFinder or that representative was responsible for validating our predicted reactants. Instead, we were trying to explain that we did not choose to validate our results using SciFinder at this point due to the high cost of the service. This is, of course, not the fault of SciFinder nor the representative. We have been negotiating with CAS to buy their reaction information and service for future validation if some funding is available. As our manuscript does not include this statement regarding the validation with SciFinder, we do not make any revisions to our manuscript regarding this comment.

Comment #1.11: “The answer to the comment #1.14 cannot be accepted neither because of poor formulation. The correction propose ‘Compared to these TF methods, though without the full coverage on the test set, G2Retro and G2Retro-B better imitate the reversed logic of synthetic reactions with two steps: reaction center identification and synthon completion.’ The problem here is that G2Retro follows its own ‘reversed logic of synthetic reactions’ and TF has no such thing to ‘imitate’.”

Response: We sincerely appreciate the reviewer for pointing it out. In this revision, we have revised the statement as below following the reviewer’s suggestion.

“Compared to these TF methods, though without the full coverage on the test set, G²Retro and G²Retro-B model reactions through a two-step process of reaction center identification and synthon completion, allowing for the interpretability of reaction centers in the predicted reactants.”

Comment #1.12: “In the comment #1.19 the correction ‘more reliable web-lab experimental validation’ must be corrected by ‘wet-lab’.”

Response: We thank the reviewer for this careful review. In this revision, we have corrected this typo.

Comment #1.13: “For comment #1.20, the situation is insufficiently improved after the correction. The authors are now precise about what is the measure presumably used for clustering. The measure maybe meaningful for reactions that have the same product. But the authors state that they ‘clustered the products according to their reaction similarity’. Here, there is no description of reaction similarity for reactions in general, including those that do not share the same product. The Figure 6 is also improperly documented and labeled: the x- and y- are not described in the caption, the color scale is not described, the acronyms LRD and HRD are not developed in the caption.”

Response: We thank the reviewer for this comment. We would like to clarify that the term “reaction similarity” in the Section “Diversity on predicted reactions” refers to the similarity between the top-predicted reactions for the **same product**, and is not applicable to reactions that do not share the same product. For each product, we looked at their top-10 predicted reactions, and used Equation (1) in the manuscript to calculate the similarity between two reactions. As described in the manuscript, for two reactions $R_1: M_1 + M_2 \rightarrow M_p$ and $R_2: M_3 + M_4 \rightarrow M_p$, the similarity between them is calculated as the half of the maximum of two possible molecule-wise similarities: $\text{sim}_m(M_1, M_3) + \text{sim}_m(M_2, M_4)$ and $\text{sim}_m(M_1, M_4) + \text{sim}_m(M_2, M_3)$ (i.e., the reactants in the two reactions are matched in different ways), where sim_m is the Tanimoto coefficient over 2,048-bit Morgan fingerprints of the molecules. For reaction $R_1: M_1 \rightarrow M_p$ and reaction $R_2: M_2 + M_3 \rightarrow M_p$, the similarity between them is calculated as the similarity between M_1 and the composite molecule consisting of two disconnected components M_2 and M_3 : $\text{sim}_m(M_1, M_2 + M_3)$. For reaction $R_1: M_1 \rightarrow M_p$ and reaction $R_2: M_2 \rightarrow M_p$, the similarity between them is calculated by $\text{sim}_m(M_1, M_2)$.

We clustered the products according to their **reaction similarity distributions**. For the top-10 predicted reactions of a product, we first calculated its 45 ($= 10 \times (10 - 1)/2$) pair-wise reaction similarities. We then conducted

a histogram to bin these 45 similarity values (all in range [0, 1] based on the definition) into 10 bins, and counted how many similarity values in each bin (i.e., frequency). We used the frequency distribution over the 10 bins to represent reaction diversity – if the reactions are very different, intuitively, we expect more frequencies in low-value bins. Such distributions are what were presented in Figure 8.

We recognize the importance of clarifying that the term “reaction similarity” is only applicable to reactions that share the same product. In this revision, we have included the following discussion in Section “Diversity on predicted reactions”:

“Please note that the reaction similarity is only applicable to two reactions that share the same product. Therefore, the product is not considered in the similarity calculation.”

We also recognize the importance of clearly describing how to calculate the reaction similarity. In this revision, we have expanded the description regarding the calculation of reaction similarity as follows,

“For reaction $R_1: M_1 \rightarrow M_p$ and reaction $R_2: M_2 + M_3 \rightarrow M_p$, the similarity between them was calculated as follows,

$$\text{sim}(R_1, R_2) = \text{sim}_m(M_1, M_2 + M_3),$$

where $M_2 + M_3$ denotes the composite molecule consisting of two disconnected components M_2 and M_3 . For reaction $R_1: M_1 \rightarrow M_p$ and reaction $R_2: M_2 \rightarrow M_p$, the similarity between them was calculated as follows,

$$\text{sim}(R_1, R_2) = \text{sim}_m(M_1, M_2).$$

”

In terms of the Figure 8 (Figure 6 in the reviewer’s comment), we have extended its caption as below to clearly describe the meaning of x- and y-axis, the color scale, and the “LRD and HRD”.

“Clustering on test products based on similarities of their predicted reactions. The x-axis indicates the range of reaction similarities (e.g., the column between 0.1 and 0.2 indicates the range (0.1, 0.2]); the y-axis shows the cluster ID and the cluster size. Each row in the heatmap corresponds to the reaction similarity distribution of a product belonging to a specific cluster; each block in the row corresponds to the frequency of reaction similarities within each similarity range, and the block color represents the scale of the frequency (e.g., a darker color indicates a higher frequency value). The clusters are labeled as ‘HRD’ for high-reaction-diversity clusters with average low reaction similarities, and ‘LRD’ for low-reaction-diversity clusters with average high reaction similarities.”

Comment #1.14: “Considering the modifications in the discussion mentioned in comment #1.21, they cannot be accepted for two reasons:

- About forward synthesis, the authors state that ‘such methods may not predict the products well if the (...) reactions are not included in the (...) training set’. This is a personal opinion unsupported by evidences or bibliography. It must be presented as such if not better grounded.

- ‘To the best of our knowledge, there have not been any studies (...) [evaluating] all the 50,070 predicted reactions (...). Therefore, it is extremely costly and time-consuming to conduct in vitro validation for all the predicted reactions.’ As mentioned before, in my opinion, this is unacceptable. The authors did not validated their results and made no efforts to do so.”

Response: We sincerely thank the reviewer for this comment. We agree with the reviewer that the discussion in Section ‘In Vitro Validation’ may not be accurate and that it does not include any *In vitro* validation results. Therefore, we have removed this section from the manuscript. Additionally, we have performed further analysis to bibliographically validate the predicted reactants generated by G²Retro for four recently approved drugs. Please refer to our response to **Comment #1.5** for more details about our analysis.

Comment #1.15: “About comment #2.1, the authors answered the GraphRetro ‘may prioritize the most possible reactants from the most possible reaction center at the very top of its predictions. However, if the most possible reaction centers are not the ground truth, GraphRetro would totally miss (...)’. In the manuscript, I read ‘Instead, G2Retro applies a greedy beam search strategy (...) to only explore the most possible top reactant graph completion paths.’ Which seems very similar to me: G2Retro may suffer the same diseases. However, the authors globally conformed to the request of the reviewer.”

Response: We thank the reviewer for the careful review and for this comment. We acknowledge that G²Retro may suffer from the same issue as GraphRetro. However, it is worth noting that the synthon completion module of G²Retro could alleviate the issue by considering all the intermediate molecular structures and accurately es-

timating the likelihoods for completed reactants. For example, if the predicted reactants with the most possible reaction center are less likely, the synthon completion module of G²Retro could effectively reduce the likelihood of these reactants by estimating a low likelihood value for each completion action, taking into account the predicted molecular structures. The analyses in Section “Diversity on predicted reactions” also demonstrates that G²Retro can predict reactants with diverse reaction centers. In contrast, the synthon completion module of GraphRetro may aggravate the issue by classifying the most possible leaving groups without considering the overall structures of predicted reactants. This could result in the inaccurate likelihood estimation of the leaving groups and may cause the less likely reactants with the most possible reaction centers to be mistakenly ranked at the top, thereby reducing the diversity of predicted reactants. In the last revision, we added the following discussion regarding the comparison between G²Retro and GraphRetro.

“GraphRetro formulates the synthon completion as a classification problem over all the subgraphs that can realize the difference between the synthons and reactants. (...) In addition, the synthon completion module of GraphRetro may fail to accurately estimate the likelihoods of leaving groups, due to the ignorance of overall structures of predicted reactants. Such inaccurate likelihood estimation may aggravate the bias of beam search and reduce the diversity of predicted reactants as discussed in GraphRetro.”

In this revision, we extended the comparison between G²Retro and GraphRetro as below.

“Although G²Retro does not outperform GraphRetro in the synthon completion module alone, its generative process allows G²Retro to consider all the intermediate molecular structures and more accurately estimate the likelihood of each completion action, conditioned on the reaction centers and the corresponding synthons from its reaction center identification module (i.e., not the ground-truth reaction centers). **Consequently, despite employing a beam search strategy similar to that of GraphRetro, the generative process of G²Retro could alleviate the bias of beam search on most possible reaction centers by accurately estimating the likelihood of the completed reactants. In contrast,** GraphRetro may not generalize well, particularly given that GraphRetro’s reaction center identification does not perform well with respect to the ground-truth reaction centers (i.e., in the top panel of Table 3), but its synthon completion module is trained using the ground-truth reaction centers (i.e., in the bottom panel of Table 3).”

Comment #1.16: “The correction in comment #2.3 is insufficient. The authors accurately state ‘We identified a set of products such that their third or fifth predicted reactions are the ground truth, referred to as having a hit at 3 or 5, respectively.’ However the caption of Figure 4 mention the term ‘hits’ without any precision. The situation did not improve.”

Response: We appreciate the reviewer for this comment. In this revision, we have extended the caption of Figure 4 as below,

“Percentage of products (Product (%)) with the different number of predicted reaction centers. a, Products with **the third predicted reaction as the ground-truth reaction (i.e., hits at 3); b,** Products with **the fifth predicted reaction as the ground-truth reaction (i.e., hits at 5).”**

Comment #1.17: “The answer of the authors to the comment #2.4 is not sufficient. The section is termed ‘In vitro validation’ and there are no in vitro validation. The proposed modification is off-topic and are not scientifically grounded.”

Response: We sincerely thank the reviewer for this comment. We agreed with the reviewer that the Section “In vitro validation” do not include the results of “in vitro” experiments and may not be accurate. Therefore, in this revision, we have removed the Section “In vitro validation” from the manuscript to improve clarity. In addition, following the reviewer’s suggestions, we also provided more predicted reactants from G²Retro and validated them bibliographically. Please refer to our response to **Comment #1.5** for the details about our prediction results and the corresponding analyses.

Comment #1.18: “The modification listed in comment #2.5 and #2.6 are satisfactory.”

Response: We thank the reviewer for this positive comment.

Comment #1.19: “The comment #2.7 gave clear directions about how to cite 26 and 7. This was not accepted by the authors.”

Response: We thank the reviewer for this comment. We would like to clarify that, in the last revision, we **did have included the citations of previous methods** including reference 26 and 7 as the reviewer suggested (in the last response, we removed the references when we quoted the paragraphs from the main manuscript, as we explained at the beginning of the last response), and acknowledged that these methods had divided

retrosynthesis prediction into two modules, and G²Retro followed this semi-template-based framework with two modules. However, we recognize that it would be beneficial to improve the clarity of our statement regarding the semi-template-based framework. In this revision, we have reformulated our statement in the Introduction as below,

“Therefore, G²Retro follows the semi-template-based frame, **as in the previous methods^{27–30}**.”

The citations for reference numbers 30 and 29 (previously referred to as 26 and 7) mentioned in the previous comment #2.7 are included in the above statement.

Comments from Reviewer 3:

Comment #3.1: “The authors substantially revised the manuscript and addressed all the major concerns raised. The clarity of the text is improved and the work better contextualized with respect to other retrosynthetic methods. What I felt were overstatements have also been removed or adapted accordingly. Therefore, my opinion is that the manuscript is now suitable for publication.”

Response: We greatly appreciate the reviewer for this positive comment. We sincerely thank the reviewer for supporting the publication of our manuscript.

REVIEWERS' COMMENTS:

Reviewer #1 (Remarks to the Author):

It is obvious that on many aspects I will never agree with the authors. However, I think that the manuscript has now achieved a consensus and there is no reason to delay its publication.

Summary of revisions:

We sincerely thank again the reviewer for comments and the editor for editorial requests. We have edited the manuscript according to the editorial requests. We summarized our edits as follows,

- (1) We redrew all the chemical structures with Nature Chemistry Template using ChemDraw.
- (2) We moved the content related to G^2 Retro in the Introduction after the "Problem Definition" section, and added a brief summary about the results and conclusions in the final paragraph of the Introduction.
- (3) We added headings for abstract and introduction as suggested, and modified the headings of results and discussions.

Comments from Reviewer 1:

Comment #1.1: “It is obvious that on many aspects I will never agree with the authors. However, I think that the manuscript has now achieved a consensus and there is no reason to delay its publication.”

Response: We greatly appreciate the reviewer for this valuable comment. We understand that there may be points where our perspectives differ. Nonetheless, we are grateful for the consensus reached. We sincerely thank you for your time and effort in reviewing our manuscript.